# Burning RED: Unlocking Subtask-Driven Reinforcement Learning and Risk-Awareness in Average-Reward Markov Decision Processes

## Abstract

Average-reward Markov decision processes (MDPs) provide a foundational framework for sequential decision-making under uncertainty. However, average-reward MDPs have remained largely unexplored in reinforcement learning (RL) settings, with the majority of RL-based efforts having been allocated to episodic and discounted MDPs. In this work, we study a unique structural property of average-reward MDPs and utilize it to introduce *Reward-Extended Differential* (or *RED*) reinforcement learning: a novel RL framework that can be used to effectively and efficiently solve various subtasks simultaneously in the average-reward setting. We introduce a family of RED learning algorithms for prediction and control, including proven-convergent algorithms for the tabular case. We then showcase the power of these algorithms by demonstrating how they can be used to learn a policy that optimizes, for the first time, the well-known conditional value-at-risk (CVaR) risk measure in a fully-online manner, *without* the use of an explicit bi-level optimization scheme or an augmented state-space.

## 1 Introduction

Markov decision processes (MDPs) (Puterman, 1994) are a long-established framework for sequential decision-making under uncertainty. Episodic and discounted MDPs, which aim to optimize a sum of rewards over time, have enjoyed success in recent years when utilizing reinforcement learning (RL) solution methods (Sutton and Barto, 2018) to tackle certain problems of interest in various domains. Despite this success however, these MDP-based methods have yet to be fully embraced in real-world applications due to the various intricacies and implications of real-world operation that often trump the ability of current state-of-the-art methods (Dulac-Arnold et al., 2021). We therefore turn to the less-explored average-reward MDP, which aims to optimize the reward received per time-step, to see how its unique structural properties can be leveraged to tackle challenging problems that have evaded its episodic and discounted counterparts.

In particular, we present results that show how the average-reward MDP's unique structural properties can be leveraged to enable a more *subtask-driven* approach to reinforcement learning, where various learning problems, or *subtasks*, are solved simultaneously (and in a fully-online manner) to help solve a larger, central learning problem. Importantly, we find compelling case-study in the realm of risk-aware decision-making that demonstrates how this subtask-driven approach can greatly simplify problems that have proven to be challenging to solve in episodic and discounted MDPs.

More formally, we introduce *Reward-Extended Differential* (or *RED*) reinforcement learning: a first-of-its-kind RL framework that makes it possible to solve various subtasks simultaneously in the average-reward setting. At the heart of this framework is the novel concept of the reward-extended temporal-difference (TD) error, an extension of the celebrated TD error (Sutton, 1988), which we leverage in combination with a unique structural property of the average-reward MDP to solve various subtasks simultaneously. We first present the RED RL framework in a generalized way, then adopt it to successfully tackle a problem that has exceeded the capabilities of current state-of-the-art methods in risk-aware decision-making: learning a policy that optimizes the well-known conditional value-at-risk (CVaR) risk measure (Rockafellar and Uryasev, 2000) in a fully-online manner *without* the use of an explicit bi-level optimization scheme or an augmented state-space.

Our work is organized as follows: in Section 2 we provide a brief overview of relevant work done on average-reward RL as well as risk-aware learning and optimization in MDP-based settings. In Section 3 we give an overview of the fundamental concepts related to average-reward RL and CVaR. In Section 4, we motivate the need and opportunity for a subtask-driven approach to RL through the lens of CVaR optimization. In Section 5, we introduce the RED RL framework, including the concept of the reward-extended TD error. We also introduce a family of RED RL algorithms for prediction and control, and highlight their convergence properties (with full convergence proofs in Appendix C). In Section 6, we empirically show how RED RL can be used to successfully learn a policy that optimizes the CVaR risk measure. Finally, in Section 7 we emphasize our framework's potential usefulness towards tackling other challenging problems outside the realm of risk-awareness, highlight some of its limitations, and suggest some directions for future research.

## 2 RELATED WORK

**Average-Reward Reinforcement Learning:** Average-reward (or average-cost) MDPs, despite being one of the most well-studied frameworks for sequential decision-making under uncertainty (Puterman, 1994), have remained relatively unexplored in reinforcement learning (RL) settings. To date, notable works on the subject (in the context of RL) include Mahadevan (1996), Tsitsiklis and Van Roy (1999), Abounadi et al. (2001), Bhatnagar et al. (2009), and Wan et al. (2021). Most relevant to our work is Wan et al. (2021), which provided a rigorous theoretical treatment of average-reward MDPs in the context of RL, and proposed the proven-convergent 'Differential Q-learning' and (off-policy) 'Differential TD-learning' algorithms for the tabular case. Our work primarily builds on Wan et al. (2021), and we utilize their proof technique when formulating the convergence proofs for our algorithms. We note that the notion of a 'subtask', as explored in our work, is different to that of hierarchical RL (e.g. Sutton et al. (1999)), where the focus is on using temporally-abstracted actions, known as 'options' or 'skills', such that the agent learns a policy for each option, as well as an inter-option policy. By contrast, in our work we learn a single policy, and the subtasks are not part of the action-space. In the episodic and discounted settings, the notion of solving multiple objectives in parallel has been explored in various works (e.g. McLeod et al. (2021)), although much of this work focuses on learning multiple features, options, policies, and/or value functions. By contrast, in our work we learn a single policy and value function, and the subtasks are not part of the state or action-spaces. To the best of our knowledge, our work is the first to explore solving subtasks simultaneously in the average-reward setting.

**Risk-Aware Learning and Optimization in MDPs:** The notion of risk-aware learning and optimization in MDP-based settings has been long-studied, from the well-established expected utility framework (Howard and Matheson, 1972), to the more contemporary framework of coherent risk measures (Artzner et al., 1999). To date, these risk-based efforts have almost exclusively focused on the episodic and discounted settings. Critically, optimizing the CVaR risk measure in these settings typically requires augmenting the state-space and/or having to utilize an explicit bi-level optimization scheme, which can, for example, involve solving multiple MDPs. Seminal works that have looked at CVaR optimization in the standard discounted and episodic settings include Bäuerle and Ott (2011) and Chow et al. (2015); Hau et al. (2023a). In the distributional setting, works such as Dabney et al. (2018) have proposed a CVaR optimization approach that does not require an augmented state-space or an explicit bi-level optimization, however it was later shown by Lim and Malik (2022) that such an approach converges to neither the optimal dynamic-CVaR nor the optimal static-CVaR policies (Lim and Malik (2022) then proposed a valid approach that utilizes an augmented state-space). Some works have looked at optimizing a time-consistent (Ruszczyński, 2010) interpretation of CVaR, however this only approximates CVaR, as CVaR is not a time-consistent risk measure (Boda and Filar, 2006). Other works have looked at optimizing similar objectives to CVaR that are more computationally tractable, such as the entropic value-at-risk (Hau et al., 2023b).

Most similar to our work (in non average-reward settings) are Stanko and Macek (2019) and Miller and Yang (2017). In Stanko and Macek (2019), the authors use a vaguely similar update to the one derived in our work, however all of the methods proposed in Stanko and Macek (2019) require either an augmented state-space or an explicit bi-level optimization. Similarly, while the approach presented in Miller and Yang (2017) does not require an augmented state-space, it requires an explicit bi-level optimization. In the average-reward setting, Xia et al. (2023) recently proposed a set of algorithms for optimizing the CVaR risk measure, however their methods require the use of an

augmented state-space and a sensitivity-based bi-level optimization. By contrast, our work, to the best of our knowledge, is the first to optimize the CVaR risk measure in an MDP-based setting without the use of an explicit bi-level optimization scheme or an augmented state-space. We note that other works have looked at optimizing other risk measures in the average-reward setting, such as the exponential cost (Murthy et al., 2023), and variance (Prashanth and Ghavamzadeh, 2016).

# 3 PRELIMINARIES

## 3.1 AVERAGE-REWARD REINFORCEMENT LEARNING

A finite average-reward MDP is the tuple $\mathcal{M} \doteq \langle \mathcal{S}, \mathcal{A}, \mathcal{R}, p \rangle$, where $\mathcal{S}$ is a finite set of states, $\mathcal{A}$ is a finite set of actions, $\mathcal{R} \subset \mathbb{R}$ is a finite set of rewards, and $p : \mathcal{S} \times \mathcal{A} \times \mathcal{R} \times \mathcal{S} \to [0, 1]$ is a probabilistic transition function that describes the dynamics of the environment. At each discrete time step, $t = 0, 1, 2, \ldots$, an agent chooses an action, $A_t \in \mathcal{A}$, based on its current state, $S_t \in \mathcal{S}$, and receives a reward, $R_{t+1} \in \mathcal{R}$, while transitioning to a (potentially) new state, $S_{t+1}$, such that $p(s', r \mid s, a) = \mathbb{P}(S_{t+1} = s', R_{t+1} = r \mid S_t = s, A_t = a)$. In an average-reward MDP, an agent aims to find a policy, $\pi : \mathcal{S} \to \mathcal{A}$, that optimizes the long-run (or limiting) average-reward, $\bar{r}$, which is defined as follows for a given policy, $\pi$:

$$\bar{r}_\pi(s) \doteq \lim_{n \to \infty} \frac{1}{n} \sum_{t=1}^{n} \mathbb{E}[R_t \mid S_0 = s, A_{0:t-1} \sim \pi]. \tag{1}$$

In this work, we limit our discussion to *stationary Markov* policies, which are time-independent policies that satisfy the Markov property. The underlying process by which average-reward MDPs operate is depicted in Fig. A.1 (in Appendix A).

Equation 1 can be simplified into a more workable form by making certain assumptions about the Markov chain, $\{S_t\}$, induced by following policy $\pi$. To this end, a *unichain* assumption is typically used when doing prediction (learning) because it ensures the existence of a unique limiting distribution of states, $\mu_\pi(s) \doteq \lim_{t \to \infty} \mathbb{P}(S_t = s \mid A_{0:t-1} \sim \pi)$, that is independent of the initial state, thereby simplifying Equation 1 to the following:

$$\bar{r}_\pi = \sum_{s \in \mathcal{S}} \mu_\pi(s) \sum_{a \in \mathcal{A}} \pi(a \mid s) \sum_{s' \in \mathcal{S}} \sum_{r \in \mathcal{R}} p(s', r \mid s, a) r. \tag{2}$$

Similarly, a *communicating* assumption is typically used for control (optimization) because it ensures the existence of a unique optimal average-reward, $\bar{r}*$, that is independent of the initial state.

To solve an average-reward MDP, solution methods such as dynamic programming or RL can be used, in conjunction with the following *Bellman* (or *Poisson*) equations:

$$v_\pi(s) = \sum_{a} \pi(a \mid s) \sum_{s'} \sum_{r} p(s', r \mid s, a)[r - \bar{r}_\pi + v_\pi(s')], \tag{3}$$

$$q_\pi(s, a) = \sum_{s'} \sum_{r} p(s', r \mid s, a)[r - \bar{r}_\pi + \max_{a'} q_\pi(s', a')], \tag{4}$$

where, $v_\pi(s)$ is the state-value function and $q_\pi(s, a)$ is the state-action value function for a given policy, $\pi$. Solution methods for average-reward MDPs are typically referred to as *differential* methods because of the reward difference (i.e., $r - \bar{r}_\pi$) operation that occurs in Equations 3 and 4. Note that solution methods typically find solutions to Equations 3 and 4 up to a constant, $c$. This is typically not a concern, given that the relative ordering of policies is usually what is of interest.

In the context of RL, Wan et al. (2021) proposed the tabular 'Differential TD-learning' and 'Differential Q-learning' algorithms, which are able to learn and/or optimize the value function and average-reward simultaneously using only the TD error. The 'Differential TD-learning' algorithm is shown below:

$$V_{t+1}(S_t) \doteq V_t(S_t) + \alpha_t \rho_t \delta_t \tag{5a}$$

$$V_{t+1}(s) \doteq V_t(s), \quad \forall s \neq S_t \tag{5b}$$

$$\delta_t \doteq R_{t+1} - \bar{R}_t + V_t(S_{t+1}) - V_t(S_t) \tag{5c}$$

$$\bar{R}_{t+1} \doteq \bar{R}_t + \eta \alpha_t \rho_t \delta_t \tag{5d}$$

where, $V_t : \mathcal{S} \to \mathbb{R}$ is a table of state-value function estimates, $\alpha_t$ is the step size, $\delta_t$ is the TD error, $\rho_t \doteq \pi(A_t|S_t)/B(A_t|S_t)$ is the importance sampling ratio (with behavior policy, $B$), $\bar{R}_t$ is an estimate of the average-reward, $\bar{r}_\pi$, and $\eta$ is a positive scalar.

We end by noting that the average-reward criterion (Equation 1) can also be optimized using discounted MDPs (e.g. Grand-Clément and Petrik (2023)). In such cases, the solution is said to be *Blackwell-optimal* because it takes into account the limiting *and* transient behavior of the system. In this work however, we employ methods that utilize the standard average-reward MDP formulation, because, as we will see in Sections 5 and 6, it enables a subtask-driven approach to RL that can alleviate computational challenges and non-trivialities that arise in discounted MDPs.

## 3.2 Conditional Value-at-Risk (CVaR)

Consider a random variable $X$ with a finite mean on a probability space $(\Omega, \mathcal{F}, \mathbb{P})$, and with a cumulative distribution function $F(x) = \mathbb{P}(X \leq x)$. The (left-tail) *value-at-risk (VaR)* of $X$ with parameter $\tau \in (0, 1)$ represents the $\tau$-quantile of $X$, such that $\mathrm{VaR}_\tau(X) = \sup\{x \mid F(x) \leq \tau\}$. The (left-tail) *conditional value-at-risk (CVaR)* of $X$ with parameter $\tau$ is defined as follows:

$$\mathrm{CVaR}_\tau(X) = \frac{1}{\tau} \int_0^\tau \mathrm{VaR}_u(X) du. \tag{6}$$

When $F(X)$ is continuous at $x = \mathrm{VaR}_\tau(X)$, the conditional value-at-risk can be interpreted as the expected value of the $\tau$ left quantile of the distribution of $X$, such that $\mathrm{CVaR}_\tau(X) = \mathbb{E}[X \mid X \leq \mathrm{VaR}_\tau(X)]$. Fig. A.2 (in Appendix A) depicts this interpretation of CVaR.

Importantly, CVaR can be formulated as follows (Rockafellar and Uryasev, 2000):

$$\mathrm{CVaR}_\tau(X) = \sup_{b \in \mathbb{R}} \mathbb{E}[b - \frac{1}{\tau}(b - X)^+] = \mathbb{E}[\mathrm{VaR}_\tau(X) - \frac{1}{\tau}(\mathrm{VaR}_\tau(X) - X)^+], \tag{7}$$

where, $(y)^+ = \max(y, 0)$. Existing MDP-based methods typically leverage the above formulation when optimizing for CVaR, by augmenting the state-space with an estimate of $\mathrm{VaR}_\tau(X)$ (in this case, $b$), and solving the following bi-level optimization:

$$\sup_\pi \mathrm{CVaR}_\tau(X) = \sup_\pi \sup_{b \in \mathbb{R}} \mathbb{E}[b - \frac{1}{\tau}(b - X)^+] = \sup_{b \in \mathbb{R}}(b - \frac{1}{\tau}\sup_\pi \mathbb{E}[(b - X)^+]), \tag{8}$$

where the 'inner' optimization problem can be solved using standard MDP solution methods.

In discounted and episodic MDPs, the random variable $X$ corresponds to a (potentially-discounted) sum of rewards. In average-reward MDPs, $X$ corresponds to the (limiting) per-step reward. In other words, the natural interpretation of CVaR in the average-reward setting is that of the CVaR of the limiting reward distribution, as shown below (for a given policy, $\pi$) (Xia et al., 2023):

$$\mathrm{CVaR}_{\tau,\pi}(s) \doteq \lim_{n \to \infty} \frac{1}{n} \sum_{t=1}^n \mathrm{CVaR}_\tau[R_t \mid S_0 = s, A_{0:t-1} \sim \pi]. \tag{9}$$

As with the average-reward (Equation 1), a unichain assumption (or similar) makes this CVaR objective independent of the initial state. In recent years, CVaR has emerged as a popular risk measure, in-part because it is a 'coherent' risk measure (Artzner et al., 1999), meaning that it satisfies key mathematical properties which can be meaningful in safety-critical and risk-related applications.

## 4 A Subtask-Driven Approach

In this section, we motivate the need and opportunity for a subtask-driven approach to RL through the lens of CVaR optimization. Let us begin by considering the standard approach used by existing MDP-based methods for optimizing CVaR, which requires an explicit bi-level optimization, as described in Equation 8. In words, Equation 8 says that to optimize CVaR, we need to pick a wide range of guesses for VaR, and for each guess, $b$, we need to solve an MDP. Then, out of all of the MDP solutions, we pick the best one as our final solution. To further compound the computational costs, this approach typically requires that the state-space be augmented with a state that corresponds

to the VaR guess, $b$. Importantly, this computationally-expensive process would not be needed if we somehow knew what the optimal value for $b$ (i.e., VaR) was. In fact, in the average-reward setting, if we knew VaR, then optimizing for CVaR ultimately amounts to optimizing an average (as per Equation 7), which can be done trivially using the standard average-reward MDP.

As such, it would appear that, to optimize CVaR, we are stuck between two extremes: a significantly computationally-expensive process if we don't know VaR, and a trivial process if we do. But what if we could estimate VaR along the way? That is, keep some sort of running estimate of VaR that we optimize simultaneously as we optimize CVaR. Indeed, such an approach has been proposed in the discounted and episodic settings (e.g. Stanko and Macek (2019)), however, no approach has been able to successfully remove both the augmented state-space and the explicit bi-level optimization requirements. The primary difficulty lies in how one updates the estimate of VaR along the way.

Critically, this is where the findings from Wan et al. (2021) come into play. In particular, Wan et al. (2021) proposed proven-convergent algorithms for the average-reward setting that can learn and/or optimize the value function and average-reward simultaneously using only the TD error. In other words, these algorithms are able to solve two learning objectives simultaneously using only the TD error. Yet, the focus in Wan et al. (2021) was on proving the convergence of such algorithms, without exploring the underlying structural properties of the average-reward MDP that made such a process possible to begin with. In this work, we formalize these underlying properties, and utilize them to show that if one modifies, or *extends*, the reward from the MDP with various learning objectives, then these objectives, or *subtasks*, can be solved simultaneously using a modified version of the TD error. Consequently, in terms of CVaR optimization, this allows us to develop appropriate learning updates for the VaR and CVaR estimates based solely on the TD error, such that we no longer need to augment the state-space or perform an explicit bi-level optimization.

In Section 5, we present the theoretical framework that enables the aforementioned subtask-driven approach. Then, in Section 6, we adapt this general-purpose framework for CVaR optimization.

## 5 REWARD-EXTENDED DIFFERENTIAL (RED) REINFORCEMENT LEARNING

In this section, we present our primary contribution: a framework for solving various learning objectives, or *subtasks*, simultaneously in the average-reward setting. We call this framework *reward-extended differential* (or *RED*) reinforcement learning. The 'differential' part of the name comes from the use of the differential algorithms from average-reward MDPs. The 'reward-extended' part of the name comes from the use of the *reward-extended TD error*, a novel concept that we will introduce shortly. Through this framework, we show how the average-reward MDP's unique structural properties can be leveraged to solve various subtasks simultaneously in a fully-online manner. We first provide a formal definition for a (generic) subtask, then proceed to derive a learning framework that allows us to simultaneously solve any given subtask that satisfies this definition. In the subsequent section, we utilize this framework to tackle the CVaR optimization problem.

**Definition 5.1** (Subtask). *A subtask, $z_i$, is any scalar prediction or control objective belonging to a corresponding finite set $\mathcal{Z}_i \subset \mathbb{R}$, such that there exists a linear (or piecewise linear) subtask function, $f : \mathcal{R} \times \mathcal{Z}_1 \times \mathcal{Z}_2 \times \cdots \times \mathcal{Z}_i \times \cdots \times \mathcal{Z}_n \to \tilde{\mathcal{R}}$, where $\mathcal{R}$ is the finite set of observed per-step rewards from the MDP $\mathcal{M}$, $\tilde{\mathcal{R}} \subset \mathbb{R}$ is a finite set of 'extended' per-step rewards whose long-run average is the primary prediction or control objective of the MDP, $\tilde{\mathcal{M}} \doteq \langle \mathcal{S}, \mathcal{A}, \tilde{\mathcal{R}}, p \rangle$, and $\mathcal{Z} = \{z_1 \in \mathcal{Z}_1, z_2 \in \mathcal{Z}_2, \ldots, z_n \in \mathcal{Z}_n\}$ is the set of $n$ subtasks that we wish to solve, such that:*

*i) $f$ is invertible with respect to each input given all other inputs; and*

*ii) each subtask $z_i \in \mathcal{Z}$ in $f$ is independent of the states and actions, and hence independent of the observed per-step reward, $R_t \in \mathcal{R}$, such that $\mathbb{E}[f(R_t, z_1, z_2, \ldots, z_n)] = f(\mathbb{E}[R_t], z_1, z_2, \ldots, z_n)$, where $\mathbb{E}$ denotes any expectation taken with respect to the states and actions.*

With this definition in mind, we now proceed by providing the basic intuition behind our framework by using the average-reward itself, $\bar{r}_\pi$, as a blueprint of sorts for how we will derive the update rules in our learning algorithms for our subtasks. In particular, we will show how the process for deriving the update rule for the average-reward estimate, $\bar{R}_t$, in Equation 5 can be adapted to derive equivalent update rules for estimates corresponding to any subtask that satisfies Definition 5.1.

Consider the Bellman equation 3. We begin by pointing out that the average-reward satisfies many of the key properties of a subtask. In particular, we can see that $\bar{r}_\pi$ satisfies $\sum[r - \bar{r}_\pi + v_\pi(s')] = \sum[r + v_\pi(s')] - \bar{r}_\pi$, where we use $\sum$ as shorthand for the sums in the Bellman equation 3. This allows us to rewrite the Bellman equation 3 as follows:

$$\bar{r}_\pi = \sum_a \pi(a \mid s) \sum_{s',r} p(s', r \mid s, a)[r + v_\pi(s') - v_\pi(s)]. \tag{10}$$

Now, if we wanted to learn $\bar{r}_\pi$ from experience, we can utilize the common RL update rule of the form: *NewEstimate $\leftarrow$ OldEstimate + StepSize [Target $-$ OldEstimate]* (Sutton and Barto, 2018) to do so. In this case, the 'target' is the term inside the expectation (i.e., the sums) in Equation 10. This yields the update in Equation 5d: $\bar{R}_{t+1} = \bar{R}_t + \eta\alpha_t\delta_t$. Hence, we are able to learn $\bar{r}_\pi$ using the TD error, $\delta$. This highlights a unique structural property of average-reward MDPs: we are able to *simultaneously* predict (learn) the value function and the average-reward using the TD error. Similarly, in the control case we are able to *simultaneously* control (optimize) these same two objectives using the TD error. We will now show, through the RED RL framework, how this structural property can be utilized to simultaneously predict or control any subtask that satisfies Definition 5.1. More specifically, we will show how we can replicate what we just did for the average-reward for any arbitrary subtask:

**Theorem 5.1** (The RED Theorem). *An average-reward MDP can simultaneously predict or control any arbitrary number of subtasks (within a single subtask function that satisfies Definition 5.1) using the TD error.*

*Proof.* Let $\tilde{R}_t = f(R_t, z_1, z_2, \ldots, z_n) = f(\cdot)$ be a linear subtask function (as per Definition 5.1) corresponding to $n$ subtasks, where $R_t \in \mathcal{R}$ is the observed per-step reward, and $\tilde{R}_t \in \tilde{\mathcal{R}}$ is the extended per-step reward whose long-run average, $\bar{r}_\pi$, is the primary prediction or control objective.

We first note that without a loss in generality, the subtask function can be written as follows:

$$\tilde{R}_t = R_t + a_0 + a_1 z_1 + a_2 z_2 + \ldots + a_n z_n, \tag{11}$$

for some constant $a_0 \in \mathbb{R}$ and $a_1, a_2, \ldots, a_n \in \mathbb{R} \setminus \{0\}$.

We can then write the TD error for the prediction case as follows:

$$\delta_t = \tilde{R}_t - \bar{R}_t + V_t(S_{t+1}) - V_t(S_t) \tag{12a}$$

$$= R_t + a_0 + a_1 Z_{1,t} + a_2 Z_{2,t} + \ldots + a_n Z_{n,t} - \bar{R}_t + V_t(S_{t+1}) - V_t(S_t), \tag{12b}$$

where $V_t : \mathcal{S} \to \mathbb{R}$ denotes a table of state-value function estimates, $\bar{R}_t$ denotes an estimate of the average-reward, $\bar{r}_\pi$, and $Z_{i,t}$ denotes an estimate of subtask $z_i$ $\forall i = 1, 2, \ldots, n$.

Similarly, we can write the Bellman equation 3 for the MDP $\tilde{\mathcal{M}}$ and solve for an arbitrary subtask, $z_i$, as follows:

$$v_\pi(s) = \mathbb{E}_\pi[\tilde{R}_t - \bar{r}_\pi + v_\pi(S_{t+1}) \mid S_t = s] \tag{13a}$$

$$0 = \mathbb{E}_\pi[R_t + a_0 + a_1 z_1 + a_2 z_2 + \ldots + a_n z_n - \bar{r}_\pi + v_\pi(S_{t+1}) - v_\pi(s) \mid S_t = s] \tag{13b}$$

$$0 = \mathbb{E}_\pi[R_t + a_0 + \ldots + a_{i-1} z_{i-1} + a_{i+1} z_{i+1} + \ldots$$
$$\ldots + a_n z_n - \bar{r}_\pi + v_\pi(S_{t+1}) - v_\pi(s) \mid S_t = s] + a_i z_i \tag{13c}$$

$$\implies z_i = \mathbb{E}_\pi[-\frac{1}{a_i}(R_t + a_0 + \ldots + a_{i-1} z_{i-1} + a_{i+1} z_{i+1} + \ldots$$
$$\ldots + a_n z_n - \bar{r}_\pi + v_\pi(S_{t+1}) - v_\pi(s)) \mid S_t = s] \tag{13d}$$

$$\doteq \mathbb{E}_\pi[\phi_{i,t} \mid S_t = s], \tag{13e}$$

where we used the fact that $z_i$ is independent of the states and actions to pull it out of the expectation. Here, we use $\phi_{i,t}$ to denote the expression inside the expectation in Equation 13d.

Hence, to learn $z_i$ from experience, we can utilize the common RL update rule (in a similar fashion to what we did with Equation 10 for the average-reward), using the term inside the expectation in Equation 13d, $\phi_{i,t}$, as the target, which yields the update:

$$Z_{i,t+1} = Z_{i,t} + \eta\alpha_t[\phi_{i,t} - Z_{i,t}] \tag{14a}$$

$$= Z_{i,t} + \eta\alpha_t(-1/a_i)\delta_t \quad \text{(when combining Equations 12 and 13d)} \tag{14b}$$

$$\doteq Z_{i,t} + \eta\alpha_t\beta_{i,t}, \tag{14c}$$

where, $Z_{i,t}$ is the estimate of subtask $z_i$ at time $t$, and $\eta\alpha_t$ is the step size.

Here, we define $\beta_{i,t} \doteq (-1/a_i)\delta_t$ as the *reward-extended TD error* for subtask $z_i$. Importantly, this term satisfies a TD error-dependent property: it goes to zero as the TD error, $\delta_t$, goes to zero. This implies that, like the average-reward update in Equation 5d, the arbitrary subtask update is dependent on the TD error, such that the subtask estimate will only cease to update once the TD error is zero. Hence, minimizing the TD error allows us to solve the arbitrary subtask simultaneously.

As such, we have derived an update rule based on the TD error for our arbitrary subtask, $z_i$. Finally, because we picked $z_i$ arbitrarily, it follows that we can derive an update rule for every subtask in $f(\cdot)$ based on the TD error. This means that we can perform prediction for all our subtasks simultaneously by minimizing the (regular) TD error. The same logic can be applied in the control case to derive equivalent updates, where we note that it directly follows from Definition 5.1 that the existence of an optimal average-reward, $\bar{r}*$, implies the existence of corresponding optimal subtask values, $z_i^* \ \forall z_i \in \mathcal{Z}$. In a similar fashion, these results can trivially be extended for piecewise linear subtask functions by applying the above logic for each linear segment separately, such that the resulting subtask updates are also piecewise linear. This completes the proof of Theorem 5.1. $\qquad\square$

Having derived the update rules for the subtasks, we now present our family of RED RL algorithms. The full set of algorithms, including algorithms that utilize function approximation, are included in Appendix B. We provide full convergence proofs for the tabular algorithms in Appendix C.

**RED TD-learning algorithm (tabular):** We update a table of estimates, $V_t : \mathcal{S} \to \mathbb{R}$ as follows:

$$\tilde{R}_{t+1} = f(R_{t+1}, Z_{1,t}, Z_{2,t}, \ldots, Z_{n,t}) \tag{15a}$$

$$\delta_t = \tilde{R}_{t+1} - \bar{R}_t + V_t(S_{t+1}) - V_t(S_t) \tag{15b}$$

$$V_{t+1}(S_t) = V_t(S_t) + \alpha_t\rho_t\delta_t \tag{15c}$$

$$V_{t+1}(s) = V_t(s), \quad \forall s \neq S_t \tag{15d}$$

$$\bar{R}_{t+1} = \bar{R}_t + \eta_r\alpha_t\rho_t\delta_t \tag{15e}$$

$$Z_{i,t+1} = Z_{i,t} + \eta_{z_i}\alpha_t\rho_t\beta_{i,t}, \quad \forall z_i \in \mathcal{Z} \tag{15f}$$

where, $R_t$ is the observed reward, $Z_{i,t}$ is an estimate of subtask $z_i$, $\beta_{i,t}$ is the reward-extended TD error for subtask $z_i$, $\alpha_t$ is the step size, $\delta_t$ is the TD error, $\rho_t$ is the importance sampling ratio, $\bar{R}_t$ is an estimate of the long-run average-reward of $\tilde{R}_t$, $\bar{r}_\pi$, and $\eta_r, \eta_{z_i}$ are positive scalars. Wan et al. (2021) showed for their Differential TD-learning algorithm that $R_t$ converges to $\bar{r}_\pi$, and $V_t$ converges to a solution of $v$ in Equation 3 for a given policy, $\pi$. We now provide an equivalent theorem for our RED TD-learning algorithm, which also shows that $Z_{i,t}$ converges to $z_{i,\pi} \ \forall z_i \in \mathcal{Z}$, where $z_{i,\pi}$ denotes the subtask value when following policy $\pi$:

**Theorem 5.2** (informal). *The RED TD-learning algorithm 15 converges, almost surely, $\bar{R}_t$ to $\bar{r}_\pi$, $Z_{i,t}$ to $z_{i,\pi} \ \forall z_i \in \mathcal{Z}$, and $V_t$ to a solution of $v$ in the Bellman Equation 3, up to an additive constant, $c$, if the following assumptions hold: 1) the Markov chain induced by the target policy, $\pi$, is unichain, 2) every state–action pair for which $\pi(a \mid s) > 0$ occurs an infinite number of times under the behavior policy, 3) the step sizes are decreased appropriately, 4) the ratio of the update frequency of the most-updated state to the least-updated state is finite, and 5) the subtasks are in accordance with Definition 5.1.*

*Proof.* See Appendix C for the full proof. $\qquad\square$

**RED Q-learning algorithm (tabular):** We update a table of estimates, $Q_t : \mathcal{S} \times \mathcal{A} \to \mathbb{R}$ as follows:

$$\tilde{R}_{t+1} = f(R_{t+1}, Z_{1,t}, Z_{2,t}, \ldots, Z_{n,t}) \tag{16a}$$

$$\delta_t = \tilde{R}_{t+1} - \bar{R}_t + \max_a Q_t(S_{t+1}, a) - Q_t(S_t, A_t) \tag{16b}$$

$$Q_{t+1}(S_t, A_t) = Q_t(S_t, A_t) + \alpha_t\delta_t \tag{16c}$$

$$Q_{t+1}(s, a) = Q_t(s, a), \quad \forall s, a \neq S_t, A_t \tag{16d}$$

$$\bar{R}_{t+1} = \bar{R}_t + \eta_r\alpha_t\delta_t \tag{16e}$$

$$Z_{i,t+1} = Z_{i,t} + \eta_{z_i}\alpha_t\beta_{i,t}, \quad \forall z_i \in \mathcal{Z} \tag{16f}$$

where, $R_t$ is the observed reward, $Z_{i,t}$ is an estimate of subtask $z_i$, $\beta_{i,t}$ is the reward-extended TD error for subtask $z_i$, $\alpha_t$ is the step size, $\delta_t$ is the TD error, $\bar{R}_t$ is an estimate of the long-run average-reward of $\tilde{R}_t$, $\bar{r}_\pi$, and $\eta_r, \eta_{z_i}$ are positive scalars. Wan et al. (2021) showed for their Differential Q-learning algorithm that $R_t$ converges to $\bar{r}*$, and $Q_t$ converges to a solution of $q$ in Equation 4. We now provide an equivalent theorem for our RED Q-learning algorithm, which also shows that $Z_{i,t}$ converges to the corresponding optimal subtask value $z_i^* \ \forall z_i \in \mathcal{Z}$:

**Theorem 5.3** (informal). *The RED Q-learning algorithm 16 converges, almost surely, $\bar{R}_t$ to $\bar{r}*$, $Z_{i,t}$ to $z_i^* \ \forall z_i \in \mathcal{Z}$, $\bar{r}_{\pi_t}$ to $\bar{r}*$, $z_{i,\pi_t}$ to $z_i^* \ \forall z_i \in \mathcal{Z}$, and $Q_t$ to a solution of $q$ in the Bellman Equation 4, up to an additive constant, $c$, where $\pi_t$ is any greedy policy with respect to $Q_t$, if the following assumptions hold: 1) the MDP is communicating, 2) the solution of $q$ in 4 is unique up to a constant, 3) the step sizes are decreased appropriately, 4) all the state–action pairs are updated an infinite number of times, 5) the ratio of the update frequency of the most-updated state–action pair to the least-updated state–action pair is finite, and 6) the subtasks are in accordance with Definition 5.1.*

*Proof.* See Appendix C for the full proof. □

# 6 CASE STUDY: RED RL FOR CVaR OPTIMIZATION

In the previous section, we derived a general-purpose framework and a corresponding set of algorithms that enable a more *subtask-driven* approach to reinforcement learning, where various learning problems, or subtasks, are solved simultaneously to help solve a larger, central learning problem. In this section, we provide a compelling case-study which illustrates how this subtask-driven approach can be used to successfully tackle the CVaR optimization problem *without* the use of an explicit bi-level optimization scheme (as in Equation 8), or an augmented state-space.

First, in order to leverage the RED RL framework for CVaR optimization, we need to derive a valid subtask function for CVaR that satisfies the requirements of Definition 5.1. It turns out that we can use a modified version of Equation 7 as the subtask function. The details of the adaptation of Equation 7 into a subtask function are presented in Appendix D. Critically, as discussed in Appendix D, optimizing the long-run average of the *extended* reward ($\tilde{R}_t$) from this subtask function corresponds to optimizing the long-run CVaR of the *observed* reward ($R_t$). Hence, we can utilize CVaR-specific versions of the RED algorithms presented in Equations 15 and 16 (or their non-tabular equivalents) to optimize VaR and CVaR, such that CVaR corresponds to the primary control objective (i.e., the $\bar{r}_\pi$ that we want to optimize), and VaR is the (single) subtask. We call the resulting algorithms, the *RED CVaR algorithms*. These algorithms, which are shown in full in Appendix D, update CVaR in an analogous way to the average-reward (i.e., CVaR corresponds to $\bar{R}_t$ in Equations 15 or 16), and update VaR using a VaR-specific version of Equation 15f or 16f as follows:

$$\text{VaR}_{t+1} = \begin{cases} \text{VaR}_t - \eta\alpha_t\delta_t, & R_t \geq \text{VaR}_t \\ \text{VaR}_t + \eta\alpha_t(\frac{\tau}{1-\tau})\delta_t, & R_t < \text{VaR}_t \end{cases}, \tag{17}$$

where, $\text{VaR}_t$ is an estimate of VaR, $\eta\alpha_t$ is the step size, $\tau$ is the CVaR parameter, and $\delta_t$ is the regular TD error. As such, we are able to optimize our subtask, VaR, and our primary objective, CVaR, without the use of an explicit bi-level optimization scheme or an augmented state-space.

We now present empirical results when applying the RED CVaR algorithms on two learning tasks. The first task is a two-state environment that we created for the purposes of testing our algorithms. It is called the *red-pill blue-pill* environment (see Appendix F), where at every time step an agent can take either a red pill, which takes them to the 'red world' state, or a blue pill, which takes them to the 'blue world' state. Each state has its own characteristic reward distribution, and in this case, for a sufficiently low CVaR parameter, $\tau$, the red world state has a reward distribution with a lower (worse) mean but higher (better) CVaR compared to the blue world state. Hence, we would expect that the Differential Q-learning algorithm (from Wan et al. (2021)) learns a policy that prefers to stay in the blue world, and that the RED CVaR Q-learning algorithm learns a policy that prefers to stay in the red world. This task is illustrated in Fig. 1a).

The second learning task is the well-known *inverted pendulum* task, where an agent learns how to optimally balance an inverted pendulum. We chose this task because it provides us with opportunity to test our algorithm in an environment where: 1) we must use function approximation (given the

large state and action spaces), and 2) where the policy for the optimal average-reward and the policy for the optimal reward CVaR is the same policy (i.e., the policy that best balances the pendulum will yield a limiting reward distribution with both the optimal average-reward and reward CVaR). This hence allows us to directly compare the performance of our RED algorithms to the regular Differential learning algorithms, as well as to gauge how function approximation affects the performance of our algorithms. For this task, we utilized a simple actor-critic architecture (Barto et al., 1983; Sutton and Barto, 2018) as this allowed us to compare the performance of the (non-tabular) RED TD-learning algorithm with a (non-tabular) Differential TD-learning algorithm. This task is illustrated in Fig. 1b). The full set of experimental details, including additional experiments performed, can be found in Appendix E.

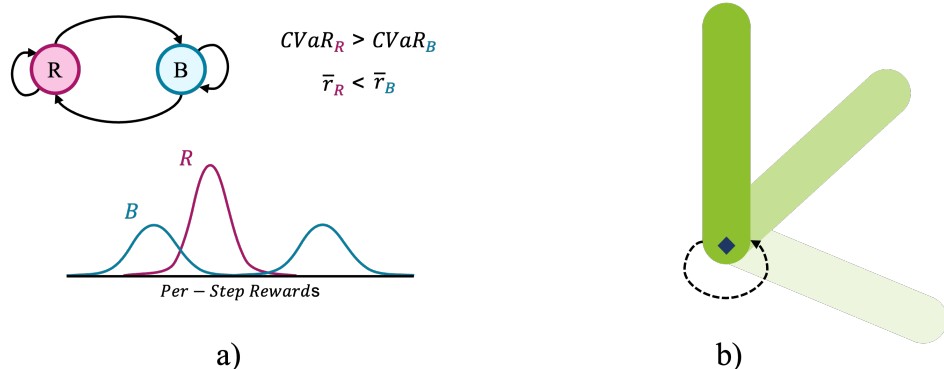

Figure 1: An illustration of the a) red-pill blue-pill, and b) inverted pendulum environments.

In terms of empirical results, Fig. 2 shows rolling averages of the average-reward and reward CVaR as learning progresses in both tasks when using the regular Differential learning algorithms (to optimize the average-reward) vs. the RED CVaR algorithms (to optimize the reward CVaR). As shown in the figure, in the red-pill blue-pill task the RED CVaR algorithm is able to successfully learn a policy that prioritizes maximizing the reward CVaR over the average-reward, thereby achieving a sort of risk-awareness. In the inverted pendulum task, both methods converge to the same policy, as expected. Fig. 3 shows typical convergence plots of the agent's VaR and CVaR estimates as learning progresses on the red-pill blue-pill task for various combinations of initial VaR and CVaR guesses. We see that regardless of the initial guess, the estimates still converge. These estimates converge to the correct VaR and CVaR values, up to a constant, thereby yielding the optimal CVaR policy, as in Fig. 2a). See Appendix E for a more detailed discussion of the empirical results.

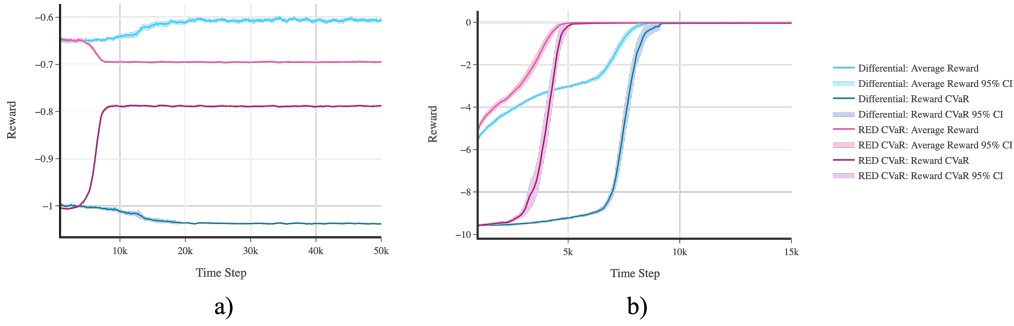

Figure 2: Rolling average-reward and reward CVaR as learning progresses when using the (risk-neutral) Differential algorithms vs. the (risk-aware) RED CVaR algorithms in the a) red-pill blue-pill, and b) inverted pendulum tasks. A solid line denotes the mean average-reward or reward CVaR, and the corresponding shaded region denotes the 95% confidence interval over 50 runs.

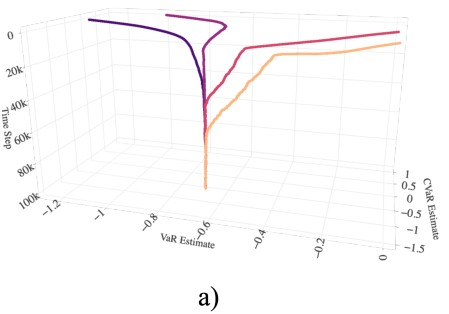
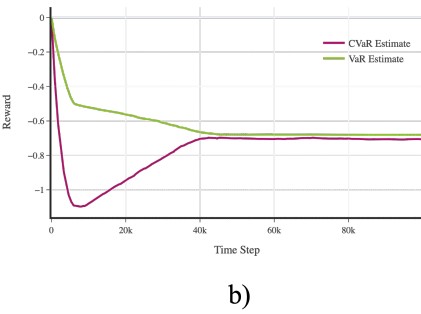

a)                                                                b)

Figure 3: Convergence plots of the agent's VaR and CVaR estimates as learning progresses when using the RED CVaR Q-learning algorithm on the red-pill blue-pill task with a) various combinations of initial VaR and CVaR guesses, and b) an initial guess of 0.0 for both the VaR and CVaR estimates.

## 7    DISCUSSION, LIMITATIONS, AND FUTURE WORK

In this work, we introduced *reward-extended differential* (or *RED*) reinforcement learning: a novel reinforcement learning framework that can be used to solve various subtasks simultaneously in the average-reward setting. We introduced a family of RED RL algorithms for prediction and control, and then showcased how these algorithms could be adopted to effectively and efficiently tackle the CVaR optimization problem. More specifically, we were able to use the RED RL framework to successfully learn a policy that optimized the CVaR risk measure without using an explicit bi-level optimization scheme or an augmented state-space, thereby alleviating some of the computational challenges and non-trivialities that arise when performing risk-based optimization in the episodic and discounted settings. Empirically, we showed that the RED-based CVaR algorithms fared well both in tabular and linear function approximation settings. Moreover, our experiments suggest that these algorithms are robust to the initial guesses for the subtasks and primary learning objective.

More broadly, our work has introduced a theoretically-sound framework that allows for a subtask-driven approach to reinforcement learning, where various learning problems (or subtasks) are solved simultaneously to help solve a larger, central learning problem. In this work, we showed (both theoretically and empirically) how this framework can be utilized to predict and/or optimize any arbitrary number of subtasks simultaneously in the average-reward setting. Central to this result is the novel concept of the reward-extended TD error, which is utilized in our framework to develop learning rules for the subtasks, and satisfies key theoretical properties that make it possible to solve any given subtask in a fully-online manner by minimizing the regular TD error. Moreover, we built-upon existing results from Wan et al. (2021) to show the almost sure convergence of tabular algorithms derived from our framework. While we have only begun to grasp the implications of our framework, we have already seen some promising indications in the CVaR case study: the ability to turn explicit bi-level optimization problems into implicit bi-level optimizations that can be solved in a fully-online manner, as well as the potential to turn certain states (that meet certain conditions) into subtasks, thereby reducing the size of the state-space.

Nonetheless, while these results are encouraging, they are subject to a number of limitations. Firstly, by nature of operating in the average-reward setting, we are subject to the somewhat-strict assumptions made about the Markov chain induced by the policy (e.g. unichain or communicating). These assumptions could restrict the applicability of our framework, as they may not always hold in practice. Similarly, our definition for a subtask requires that the associated subtask function be linear, which may also limit the applicability of our framework to simpler functions. Finally, it remains to be seen empirically how our framework performs when dealing with multiple subtasks, when taking on more complex tasks, and/or when utilizing nonlinear function approximation.

In future work, we hope to address many of these limitations, as well as explore how these promising results can be extended to other domains, beyond the risk-awareness problem. In particular, we believe that the ability to optimize various subtasks simultaneously, as well as the potential to reduce the size of the state-space, by converting certain states to subtasks (where appropriate), could help alleviate significant computational challenges in other areas moving forward.

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

# A ADDITIONAL FIGURES

In this appendix, we provide figures that present average-reward MDPs and the CVaR risk measure in a more visual manner.

## A.1 AVERAGE-REWARD MDPS

Fig. A.1 depicts the underlying process by which average-reward MDPs operate, where a given policy, $\pi$, induces a Markov chain, $\{S_t\}_\pi$, that yields a stationary reward distribution, whose mean corresponds to the long-run average-reward $\bar{r}_\pi$. Different policies can then be compared based on their $\bar{r}_\pi$ values to find the policy that yields the optimal average-reward.

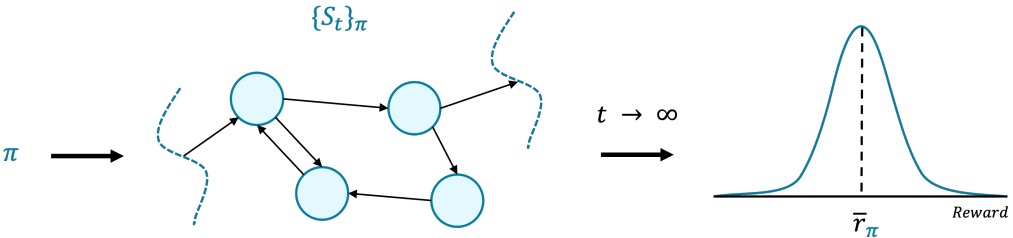

Figure A.1: Visual depiction of the underlying process by which average-reward MDPs operate. Here, following policy $\pi$ induces a Markov chain, $\{S_t\}_\pi$. As $t \to \infty$, this yields a stationary (or steady-state) reward distribution with an average reward, $\bar{r}_\pi$. It is this long-run (or steady-state) average-reward that the standard average-reward MDP formulation aims to optimize.

## A.2 CVAR

Fig. A.2a) depicts the interpretation of CVaR as the expected value of the $\tau$ left quantile of the distribution corresponding to a random variable. Fig. A.2b) depicts two limiting reward distributions that have the same long-run average-reward, but different CVaR values (assuming a sufficiently low CVaR parameter, $\tau$).

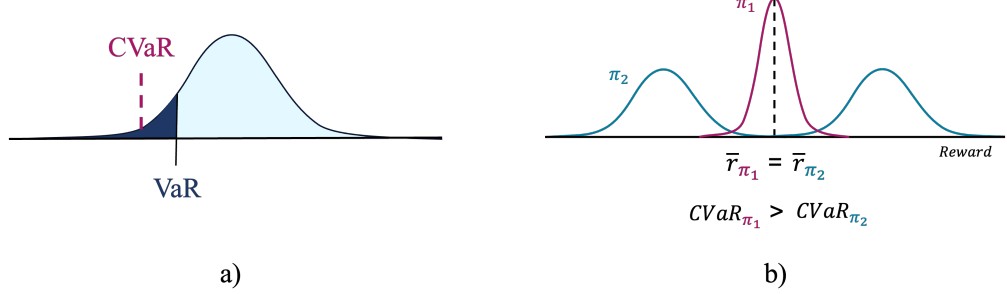

Figure A.2: a) The left-tail conditional value-at-risk (CVaR) of a probability distribution; b) The limiting reward distributions induced by two policies, $\pi_1$ and $\pi_2$. Both distributions have the same long-run average-reward, but different CVaR values (assuming a sufficiently low CVaR $\tau$).

## B  RED RL ALGORITHMS

In this appendix, we provide pseudocode for our RED RL algorithms. We first present tabular algorithms, whose convergence proofs are included in Appendix C, and then provide equivalent algorithms that utilize function approximation.

---

**Algorithm 1** RED TD-Learning (Tabular)

---

**Input:** the policy $\pi$ to be evaluated, policy $B$ to be used, subtask function $f$ with constants $a_1, a_2, \ldots, a_n$
**Algorithm parameters:** step size parameters $\alpha, \eta_r, \eta_{z_1}, \eta_{z_2}, \ldots, \eta_{z_n}$
Initialize $V(s) \; \forall s$; $\bar{R}$ arbitrarily (e.g. to zero)
Initialize subtasks $Z_1, Z_2, \ldots, Z_n$ arbitrarily (e.g. to zero)
Obtain initial $S$
**while** still time to train **do**
  $A \leftarrow$ action given by $B$ for $S$
  Take action $A$, observe $R, S'$
  $\tilde{R} = f(R, Z_1, Z_2, \ldots, Z_n)$
  $\delta = \tilde{R} - \bar{R} + V(S') - V(S)$
  $\rho = \pi(A \mid S)/B(A \mid S)$
  $V(S) = V(S) + \alpha\rho\delta$
  $\bar{R} = \bar{R} + \eta_r \alpha\rho\delta$
  $\beta_i = (-1/a_i)\delta, \quad \forall i = 1, 2, \ldots, n$
  $Z_i = Z_i + \eta_{z_i}\alpha\rho\beta_i, \quad \forall i = 1, 2, \ldots, n$
  $S = S'$
**end while**
return V

---

**Algorithm 2** RED Q-Learning (Tabular)

---

**Input:** the policy $\pi$ to be used (e.g., $\epsilon$-greedy), subtask function $f$ with constants $a_1, a_2, \ldots, a_n$
**Algorithm parameters:** step size parameters $\alpha, \eta_r, \eta_{z_1}, \eta_{z_2}, \ldots, \eta_{z_n}$
Initialize $Q(s, a) \; \forall s, a$; $\bar{R}$ arbitrarily (e.g. to zero)
Initialize subtasks $Z_1, Z_2, \ldots, Z_n$ arbitrarily (e.g. to zero)
Obtain initial $S$
**while** still time to train **do**
  $A \leftarrow$ action given by $\pi$ for $S$
  Take action $A$, observe $R, S'$
  $\tilde{R} = f(R, Z_1, Z_2, \ldots, Z_n)$
  $\delta = \tilde{R} - \bar{R} + \max_a Q(S', a) - Q(S, A)$
  $Q(S, A) = Q(S, A) + \alpha\delta$
  $\bar{R} = \bar{R} + \eta_r \alpha\delta$
  $\beta_i = (-1/a_i)\delta, \quad \forall i = 1, 2, \ldots, n$
  $Z_i = Z_i + \eta_{z_i}\alpha\beta_i, \quad \forall i = 1, 2, \ldots, n$
  $S = S'$
**end while**
return Q

---

---

**Algorithm 3** RED TD-Learning (Function Approximation)

---

**Input:** the policy $\pi$ to be evaluated, policy $B$ to be used, a differentiable state-value function parameterization: $\hat{v}(s, \boldsymbol{w})$, subtask function $f$ with constants $a_1, a_2, \ldots, a_n$
**Algorithm parameters:** step size parameters $\alpha, \eta_r, \eta_{z_1}, \eta_{z_2}, \ldots, \eta_{z_n}$
Initialize state-value weights $\boldsymbol{w} \in \mathbb{R}^d$ arbitrarily (e.g. to $\boldsymbol{0}$)
Initialize subtasks $Z_1, Z_2, \ldots, Z_n$ arbitrarily (e.g. to zero)
Obtain initial $S$
**while** still time to train **do**
    $A \leftarrow$ action given by $B$ for $S$
    Take action $A$, observe $R, S'$
    $\tilde{R} = f(R, Z_1, Z_2, \ldots, Z_n)$
    $\delta = \tilde{R} - \bar{R} + \hat{v}(S', \boldsymbol{w}) - \hat{v}(S, \boldsymbol{w})$
    $\rho = \pi(A \mid S)/B(A \mid S)$
    $\boldsymbol{w} = \boldsymbol{w} + \alpha\rho\delta\nabla\hat{v}(S, \boldsymbol{w})$
    $\bar{R} = \bar{R} + \eta_r\alpha\rho\delta$
    $\beta_i = (-1/a_i)\delta, \quad \forall i = 1, 2, \ldots, n$
    $Z_i = Z_i + \eta_{z_i}\alpha\rho\beta_i, \quad \forall i = 1, 2, \ldots, n$
    $S = S'$
**end while**
return $\boldsymbol{w}$

---

**Algorithm 4** RED Q-Learning (Function Approximation)

---

**Input:** the policy $\pi$ to be used (e.g., $\epsilon$-greedy), a differentiable state-action value function parameterization: $\hat{q}(s, a, \boldsymbol{w})$, subtask function $f$ with constants $a_1, a_2, \ldots, a_n$
**Algorithm parameters:** step size parameters $\alpha, \eta_r, \eta_{z_1}, \eta_{z_2}, \ldots, \eta_{z_n}$
Initialize state-action value weights $\boldsymbol{w} \in \mathbb{R}^d$ arbitrarily (e.g. to $\boldsymbol{0}$)
Initialize subtasks $Z_1, Z_2, \ldots, Z_n$ arbitrarily (e.g. to zero)
Obtain initial $S$
**while** still time to train **do**
    $A \leftarrow$ action given by $\pi$ for $S$
    Take action $A$, observe $R, S'$
    $\tilde{R} = f(R, Z_1, Z_2, \ldots, Z_n)$
    $\delta = \tilde{R} - \bar{R} + \max_a \hat{q}(S', a, \boldsymbol{w}) - \hat{q}(S, A, \boldsymbol{w})$
    $\boldsymbol{w} = \boldsymbol{w} + \alpha\delta\nabla\hat{q}(S, A, \boldsymbol{w})$
    $\bar{R} = \bar{R} + \eta_r\alpha\delta$
    $\beta_i = (-1/a_i)\delta, \quad \forall i = 1, 2, \ldots, n$
    $Z_i = Z_i + \eta_{z_i}\alpha\beta_i, \quad \forall i = 1, 2, \ldots, n$
    $S = S'$
**end while**
return $\boldsymbol{w}$

---

## C  CONVERGENCE PROOFS

In this appendix, we present the full convergence proofs for the tabular RED TD-learning and tabular RED Q-learning algorithms. Our general strategy is as follows: we first show that the results from Wan et al. (2021), which show the a.s. convergence of the value function and average-reward estimates of differential algorithms, are applicable to our algorithms. We then build upon these results to show that the subtask estimates of our algorithms converge as well.

For consistency, we adopt similar notation as Wan et al. (2021) for our proofs:

- For a given vector $x$, let $\sum x$ denote the sum of all elements in $x$, such that $\sum x \doteq \sum_i x(i)$.
- Let $\bar{r}_*$ denote the optimal average-reward.
- Let $z_{i_*}$ denote the corresponding optimal subtask value for subtask $z_i \in \mathcal{Z}$.

### C.1  CONVERGENCE PROOF FOR THE TABULAR RED TD-LEARNING ALGORITHM

In this section, we present the proof for the convergence of the value function, average-reward, and subtask estimates of the RED TD-learning algorithm. Similar to what was done in Wan et al. (2021), we will begin by considering a general algorithm, called *General RED TD*. We will first define General RED TD, then show how the RED TD-learning algorithm is a special case of this algorithm. We will then provide necessary assumptions, state the convergence theorem of General RED TD, and then provide a proof for the theorem, where we show that the value function, average-reward, and subtask estimates converge, thereby showing that the RED TD-learning algorithm converges. We begin by introducing the General RED TD algorithm:

Consider an MDP $\mathcal{M} \doteq \langle \mathcal{S}, \mathcal{A}, \mathcal{R}, p \rangle$, a behavior policy, $B$, and a target policy, $\pi$. Given a state $s \in \mathcal{S}$ and discrete step $n \geq 0$, let $A_n(s) \sim B(\cdot \mid s)$ denote the action selected using the behavior policy, let $R_n(s, A_n(s)) \in \mathcal{R}$ denote a sample of the resulting reward, and let $S'_n(s, A_n(s)) \sim p(\cdot, \cdot \mid s, a)$ denote a sample of the resulting state. Let $\{Y_n\}$ be a set-valued process taking values in the set of nonempty subsets of $\mathcal{S}$, such that: $Y_n = \{s : s \text{ component of the } |\mathcal{S}|\text{-sized table of state-value}$ estimates, $V$, that was updated at step $n\}$. Let $\nu(n, s) \doteq \sum_{j=0}^{n} I\{s \in Y_j\}$, where $I$ is the indicator function, such that $\nu(n, s)$ represents the number of times $V(s)$ was updated up to step $n$.

Now, let $f$ be a valid subtask function (see Definition 5.1), such that $\tilde{R}_n(s, A_n(s)) \doteq f(R_n(s, A_n(s)), Z_{1,n}, Z_{2,n}, \ldots, Z_{k,n})$ for $k$ subtasks $\in \mathcal{Z}$, where $\tilde{R}_n(s, A_n(s))$ is the extended reward, $\mathcal{Z}$ is the set of subtasks, and $Z_{i,n}$ denotes the estimate of subtask $z_i \in \mathcal{Z}$ at step $n$. Consider an MDP with the extended reward: $\tilde{\mathcal{M}} \doteq \langle \mathcal{S}, \mathcal{A}, \tilde{\mathcal{R}}, p \rangle$, such that $\tilde{R}_n(s, A_n(s)) \in \tilde{\mathcal{R}}$. The update rules of General RED TD for this MDP are as follows, for $n \geq 0$:

$$V_{n+1}(s) \doteq V_n(s) + \alpha_{\nu(n,s)} \rho_n(s) \delta_n(s) I\{s \in Y_n\}, \quad \forall s \in \mathcal{S}, \tag{C.1}$$

$$\bar{R}_{n+1} \doteq \bar{R}_n + \eta_r \sum_s \alpha_{\nu(n,s)} \rho_n(s) \delta_n(s) I\{s \in Y_n\}, \tag{C.2}$$

$$Z_{i,n+1} \doteq Z_{i,n} + \eta_{z_i} \sum_s \alpha_{\nu(n,s)} \rho_n(s) \beta_{i,n}(s) I\{s \in Y_n\}, \quad \forall z_i \in \mathcal{Z}, \tag{C.3}$$

where,

$$\begin{aligned} \delta_n(s) &\doteq \tilde{R}_n(s, A_n(s)) - \bar{R}_n + V_n(S'_n(s, A_n(s))) - V_n(s) \\ &= f(R_n(s, A_n(s)), Z_{1,n}, Z_{2,n}, \ldots, Z_{k,n}) - \bar{R}_n + V_n(S'_n(s, A_n(s))) - V_n(s), \end{aligned} \tag{C.4}$$

and,

$$\beta_{i,n}(s) \doteq \phi_{z_i,n} - Z_{i,n}, \quad \forall z_i \in \mathcal{Z}. \tag{C.5}$$

Here, $\rho_n(s) \doteq \pi(A_n(s) \mid s) / B(A_n(s) \mid s)$ denotes the importance sampling ratio (with behavior policy, $B$), $\bar{R}_n$ denotes the estimate of the average-reward (see Equation 2), $\delta_n(s)$ denotes the TD error, $\eta_r$ and $\eta_{z_i}$ are positive scalars, $\phi_{z_i,n}$ denotes the inverse of the TD error (i.e., Equation C.4) with respect to subtask estimate $Z_{i,n}$ given all other inputs when $\delta_n(s) = 0$, and $\alpha_{\nu(n,s)}$ denotes the step size at time step $n$ for state $s$. In this case, the step size depends on the sequence $\{\alpha_n\}$, as well as the number of visitations to state $s$, which is denoted by $\nu(n, s)$.

We now show that the RED TD-learning algorithm is a special case of the General RED TD algorithm. Consider a sequence of experience from our MDP $\tilde{\mathcal{M}}$: $S_t, A_t(S_t), \tilde{R}_{t+1}, S_{t+1}, \ldots$ . Now recall the set-valued process $\{Y_n\}$. If we let $n =$ time step $t$, we have:

$$Y_t(s) = \begin{cases} 1, s = S_t, \\ 0, \text{ otherwise}, \end{cases}$$

as well as $S'_n(S_t, A_t(S_t)) = S_{t+1}$, $R_n(S_t, A_t) = R_{t+1}$, $\tilde{R}_n(S_t, A_t(S_t)) = \tilde{R}_{t+1}$.

Hence, update rules C.1, C.2, C.3, C.4, and C.5 become:

$$V_{t+1}(S_t) \doteq V_t(S_t) + \alpha_{\nu(t,S_t)}\rho_t(S_t)\delta_t \text{ , and } V_{t+1}(s) \doteq V_t(s), \forall s \neq S_t, \tag{C.6}$$

$$\bar{R}_{t+1} \doteq \bar{R}_t + \eta_r\alpha_{\nu(t,S_t)}\rho_t(S_t)\delta_t, \tag{C.7}$$

$$Z_{i,t+1} \doteq Z_{i,t} + \eta_{z_i}\alpha_{\nu(t,S_t)}\rho_t(S_t)\beta_{i,t}, \quad \forall z_i \in \mathcal{Z}, \tag{C.8}$$

$$\delta_t \doteq \tilde{R}_{t+1} - \bar{R}_t + V_t(S_{t+1}) - V_t(S_t),$$
$$= f(R_{t+1}, Z_{1,t}, Z_{2,t}, \ldots, Z_{k,t}) - \bar{R}_t + V_t(S_{t+1}) - V_t(S_t), \tag{C.9}$$

$$\beta_{i,t} \doteq \phi_{z_i,t} - Z_{i,t}, \quad \forall z_i \in \mathcal{Z}, \tag{C.10}$$

which are RED TD-learning's update rules with $\alpha_{\nu(t,S_t)}$ denoting the step size at time $t$.

We now specify the assumptions on General RED TD that are needed to ensure convergence. Please refer to Wan et al. (2021) for an in-depth discussion on Assumptions C.1 – C.5:

**Assumption C.1** (Unichain Assumption). *The Markov chain induced by the target policy is unichain.*

**Assumption C.2** (Coverage Assumption). $B(a \mid s) > 0$ *if* $\pi(a \mid s) > 0$ *for all* $s \in \mathcal{S}$, $a \in \mathcal{A}$.

**Assumption C.3** (Step Size Assumption). $\alpha_n > 0$, $\sum_{n=0}^{\infty} \alpha_n = \infty$, $\sum_{n=0}^{\infty} \alpha_n^2 < \infty$.

**Assumption C.4** (Asynchronous Step Size Assumption 1). *Let* $[\cdot]$ *denote the integer part of* $(\cdot)$. *For* $x \in (0, 1)$,

$$\sup_i \frac{\alpha_{[xi]}}{\alpha_i} < \infty$$

*and*

$$\frac{\sum_{j=0}^{[yi]} \alpha_j}{\sum_{j=0}^{i} \alpha_j} \to 1$$

*uniformly in* $y \in [x, 1]$.

**Assumption C.5** (Asynchronous step size Assumption 2). *There exists* $\Delta > 0$ *such that*

$$\liminf_{n \to \infty} \frac{\nu(n, s)}{n + 1} \geq \Delta,$$

*a.s., for all* $s \in \mathcal{S}$. *Furthermore, for all* $x > 0$, *and*

$$N(n, x) = \min \left\{ m \geq n : \sum_{i=n+1}^{m} \alpha_i \geq x \right\},$$

*the limit*

$$\lim_{n \to \infty} \frac{\sum_{i=\nu(n,s)}^{\nu(N(n,x),s)} \alpha_i}{\sum_{i=\nu(n,s')}^{\nu(N(n,x),s')} \alpha_i}$$

*exists a.s. for all* $s, s'$.

Assumptions C.3, C.4, and C.5, which originate from Borkar (1998), outline the step size requirements needed to show the convergence of stochastic approximation algorithms. Assumptions C.3 and C.4 can be satisfied with step size sequences that decrease to 0 appropriately, including $1/n$, $1/(n \log n)$, and $\log n/n$ (Abounadi et al., 2001). Assumption C.5 first requires that the limiting ratio of visits to any given state, compared to the total number of visits to all states, is greater than or equal to some fixed positive value. The assumption then requires that the relative update frequency between any two states is finite. For instance, Assumption C.5 can be satisfied with $\alpha_n = 1/n$ (see page 403 of Bertsekas and Tsitsiklis (1996) for more information).

**Assumption C.6** (Subtask Function Assumption). *The subtask function, $f$, is 1) linear or piecewise linear, and 2) is invertible with respect to each input given all other inputs.*

**Assumption C.7** (Subtask Independence Assumption). *Each subtask $z_i \in \mathcal{Z}$ in $f$ is independent of the states and actions, and hence independent of the observed reward, $R_n(s, a)$, such that $\mathbb{E}[f(R_n(s, a), Z_{1,n}, Z_{2,n}, \ldots, Z_{k,n})] = f(\mathbb{E}[R_n(s, a)], Z_{1,n}, Z_{2,n}, \ldots, Z_{k,n})$, where $\mathbb{E}$ denotes any expectation taken with respect to the states and actions.*

Assumptions C.6 and C.7 outline the subtask-related requirements. Assumption C.6 ensures that we can explicitly write out the update C.3, and Assumption C.7 ensures that we do not break the Markov property in the process (i.e., we preserve the Markov property by ensuring that the subtasks are independent of the states and actions, and thereby also independent of the observed reward).

We next point out that it is easy to verify that under Assumption C.1, the following system of equations:

$$v_\pi(s) = \sum_a \pi(a \mid s) \sum_{s', \tilde{r}} p(s', \tilde{r} \mid s, a)(\tilde{r} - \bar{r}_\pi + v_\pi(s')), \text{ for all } s \in \mathcal{S},$$
$$= \sum_a \pi(a \mid s) \sum_{s', r} p(s', r \mid s, a)(f(r, z_1, z_2, \ldots, z_k) - \bar{r}_\pi + v_\pi(s')), \quad \text{(C.11)}$$

and,

$$\bar{r}_\pi - \bar{R}_0 = \eta_r \left( \sum v_\pi - \sum V_0 \right), \quad \text{(C.12)}$$
$$z_{i,\pi} - Z_{i,0} = \eta_i \left( \sum v_\pi - \sum V_0 \right), \text{ for all } z_i \in \mathcal{Z}, \quad \text{(C.13)}$$

has a unique solution of $v_\pi$, where $\bar{r}_\pi$ denotes the average-reward induced by following a given policy, $\pi$, and $z_{i,\pi}$ denotes the corresponding subtask value for subtask $z_i \in \mathcal{Z}$. Denote this unique solution of $v_\pi$ as $v_\infty$.

We are now ready to state the convergence theorem:

**Theorem C.1.1** (Convergence of General RED TD). *If Assumptions C.1 – C.7 hold, then General RED TD (Equations C.1 – C.5) converges a.s., $\bar{R}_n$ to $\bar{r}_\pi$, $Z_{i,n}$ to $z_{i,\pi}$ $\forall z_i \in \mathcal{Z}$, and $V_n$ to $v_\infty$.*

We prove this theorem in the following section. To do so, we first show that General RED TD is of the same form as *General Differential TD* from Wan et al. (2021), which consequently allows us to apply their convergence results for the value function and average-reward estimates of General Differential TD to General RED TD. We then build upon these results, using similar techniques as Wan et al. (2021), to show that the subtask estimates converge as well.

### C.1.1   PROOF OF THEOREM C.1.1

**Convergence of the average-reward and state-value function estimates:**

Consider the increment to $\bar{R}_n$ at each step. We can see from Equation C.2 that the increment is $\eta_r$ times the increment to $V_n$. As such, as was done in Wan et al. (2021), we can write the cumulative increment as follows:

$$\bar{R}_n - \bar{R}_0 = \eta_r \sum_{j=0}^{n-1} \sum_s \alpha_{\nu(j,s)} \rho_j(s) \delta_j(s) I\{s \in Y_j\}$$

$$= \eta_r \left( \sum V_n - \sum V_0 \right)$$

$$\implies \bar{R}_n = \eta_r \sum V_n - \eta_r \sum V_0 + \bar{R}_0 = \eta_r \sum V_n - c, \tag{C.14}$$

$$\text{where } c \doteq \eta_r \sum V_0 - \bar{R}_0. \tag{C.15}$$

We can then substitute $\bar{R}_n$ in C.1 with C.14 $\forall s \in \mathcal{S}$, which yields:

$$V_{n+1}(s) = V_n(s) + \dots$$
$$\alpha_{\nu(n,s)} \rho_n(s) \left( \tilde{R}_n(s, A_n(s)) + V_n(S'_n(s, A_n(s))) - V_n(s) - \eta \sum V_n + c \right) I\{s \in Y_n\}$$

$$V_{n+1}(s) = V_n(s) + \dots$$
$$\alpha_{\nu(n,s)} \rho_n(s) \left( \hat{R}_n(s, A_n(s)) + V_n(S'_n(s, A_n(s))) - V_n(s) - \eta \sum V_n \right) I\{s \in Y_n\}, \tag{C.16}$$

where $\hat{R}_n(s, A_n(s)) \doteq \tilde{R}_n(s, A_n(s)) + c = f(R_n(s, A_n(s)), Z_{1,n}, Z_{2,n}, \dots, Z_{k,n}) + c$.

Equation C.16 is now in the same form as Equation (B.37) (i.e., General Differential TD) from Wan et al. (2021), who showed that the equation converges a.s. $V_n$ to $v_\infty$ as $n \to \infty$. Moreover, from this result, Wan et al. (2021) showed that $\bar{R}_n$ converges a.s. to $\bar{r}_\pi$ as $n \to \infty$. Given that General RED TD adheres to all the assumptions listed for General Differential TD in Wan et al. (2021), these convergence results apply to General RED TD.

**Convergence of the subtask estimates:**

Consider the increment to $Z_{i,n}$ (for an arbitrary subtask $z_i \in \mathcal{Z}$) at each step. Given Equation 14, we can write the increment in Equation C.3 as some constant, subtask-specific fraction of the increment to $V_n$. Consequently, we can write the cumulative increment as follows:

$$Z_{i,n} - Z_{i,0} = \eta_{z_i} \sum_{j=0}^{n-1} \sum_s \alpha_{\nu(j,s)} \rho_j(s) \beta_{i,j}(s) I\{s \in Y_j\}$$

$$= \eta_{z_i} \sum_{j=0}^{n-1} \sum_s \alpha_{\nu(j,s)} \rho_j(s) b_i \delta_j(s) I\{s \in Y_j\}$$

$$= \eta_i \left( \sum V_n - \sum V_0 \right)$$

$$\implies Z_{i,n} = \eta_i \sum V_n - \eta_i \sum V_0 + Z_{i,0} = \eta_i \sum V_n - c, \tag{C.17}$$

where,

$$c \doteq \eta_i \sum V_0 - Z_{i,0}, \text{ and} \tag{C.18}$$

$$\eta_i \doteq \eta_{z_i} b_i. \tag{C.19}$$

Now, let $f(Z_{i,n})$ be shorthand for the subtask function (i.e., $\tilde{R}_n(s, A_n(s))$). We can substitute $Z_{i,n}$ in C.1 with C.17 $\forall s \in \mathcal{S}$ as follows:

$$V_{n+1}(s) = V_n(s) + \dots$$
$$\alpha_{\nu(n,s)}\rho_n(s)\left(\tilde{R}_n(s, A_n(s)) - \bar{R} + V_n(S'_n(s, A_n(s))) - V_n(s)\right)I\{s \in Y_n\}$$

$$\implies V_{n+1}(s) = V_n(s) + \dots$$
$$\alpha_{\nu(n,s)}\rho_n(s)\left(f(Z_{i,n}) - \bar{R} + V_n(S'_n(s, A_n(s))) - V_n(s)\right)I\{s \in Y_n\}$$

$$\implies V_{n+1}(s) = V_n(s) + \dots$$
$$\alpha_{\nu(n,s)}\rho_n(s)\left(f(\underbrace{\eta_i\sum V_n}_{\hat{Z}_{i,n}} - c) - \bar{R} + V_n(S'_n(s, A_n(s))) - V_n(s)\right)I\{s \in Y_n\}$$

$$\implies V_{n+1}(s) = V_n(s) + \dots$$
$$\alpha_{\nu(n,s)}\rho_n(s)\left(\hat{f}(\hat{Z}_{i,n}) - \bar{R} + V_n(S'_n(s, A_n(s))) - V_n(s)\right)I\{s \in Y_n\}$$

$$\implies V_{n+1}(s) = V_n(s) + \dots$$
$$\alpha_{\nu(n,s)}\rho_n(s)\left(\hat{R}_n - \bar{R} + V_n(S'_n(s, A_n(s))) - V_n(s)\right)I\{s \in Y_n\},$$
$$\tag{C.20}$$

where, $\hat{R}_n \doteq \hat{f}(\hat{Z}_{i,n}) = f(Z_{i,n}+c) = h(\tilde{R}_n)$. Here, $h(\tilde{R}_n)$ corresponds to the change in $\tilde{R}_n$ due to shifting subtask $Z_{i,n}$ by $c$. Denote the inverse of $h(\tilde{R}_n)$ (which exists given Assumption C.6) as $h^{-1}$.

Now consider an MDP, $\hat{\mathcal{M}}$, which has rewards, $\hat{\mathcal{R}}$, corresponding to rewards modified by $h$ from the MDP $\tilde{\mathcal{M}}$, has the same state and action spaces as $\tilde{\mathcal{M}}$, and has the transition probabilities defined as:
$$\hat{p}(s', h(\hat{r}) \mid s, a) \doteq p(s', \tilde{r} \mid s, a), \tag{C.21}$$
such that $\hat{\mathcal{M}} \doteq \langle \mathcal{S}, \mathcal{A}, \hat{\mathcal{R}}, \hat{p}\rangle$. It is easy to check that the unichain assumption holds for the transformed MDP $\hat{\mathcal{M}}$. As such, and given Assumptions C.6 and C.7, the average-reward induced by following policy $\pi$ for the MDP $\hat{\mathcal{M}}$, $\hat{\bar{r}}_\pi$, can be written as follows:
$$\hat{\bar{r}}_\pi = h(\bar{r}_\pi). \tag{C.22}$$

Now, because
$$v_\infty(s) = \sum_a \pi(a \mid s)\sum_{s',\tilde{r}} p(s', \tilde{r} \mid s, a)(\tilde{r} + v_\infty(s') - \bar{r}_\pi) \quad \text{(from C.11)}$$
$$= \sum_a \pi(a \mid s)\sum_{s',\tilde{r}} p(s', \tilde{r} \mid s, a)(\tilde{r} + v_\infty(s') - h^{-1}(\hat{\bar{r}}_\pi)) \quad \text{(from C.22)}$$
$$= \sum_a \pi(a \mid s)\sum_{s',\tilde{r}} p(s', \tilde{r} \mid s, a)(h(\tilde{r}) + v_\infty(s') - \hat{\bar{r}}_\pi) \quad \text{(by linearity of } h)$$
$$= \sum_a \pi(a \mid s)\sum_{s',\tilde{r}} \hat{p}(s', \tilde{r} \mid s, a)(\tilde{r} + v_\infty(s') - \hat{\bar{r}}_\pi) \quad \text{(from C.21)},$$

we can see that $v_\infty$ is a solution of not just the state-value Bellman equation for the MDP $\tilde{\mathcal{M}}$, but also the state-value Bellman equation for the transformed MDP $\hat{\mathcal{M}}$.

Next, we can write the subtask value induced by following policy $\pi$ for the MDP $\hat{\mathcal{M}}$, $\hat{z}_{i,\pi}$, as follows:
$$\hat{z}_{i,\pi} = z_{i,\pi} + c. \tag{C.23}$$

We can then combine Equations C.13, C.18, and C.23, which yields:

$$\hat{z}_{i,\pi} = \eta_i \sum v_\infty. \tag{C.24}$$

Next, we can combine Equation C.17 with the result from Wan et al. (2021) which shows that $V_n \to v_\infty$, which yields:

$$Z_{i,n} \to \eta_i \sum v_\infty - c. \tag{C.25}$$

Moreover, because $\hat{z}_{i,\pi} = \eta_i \sum v_\infty$ (Equation C.24), we have:

$$Z_{i,n} \to \hat{z}_{i,\pi} - c. \tag{C.26}$$

Finally, because $\hat{z}_{i,\pi} = z_{i,\pi} + c$ (Equation C.23), we have:

$$Z_{i,n} \to z_{i,\pi} \quad \text{a.s. as} \quad n \to \infty. \tag{C.27}$$

This completes the proof of Theorem C.1.1.

## C.2    Convergence Proof for the Tabular RED Q-learning Algorithm

In this section, we present the proof for the convergence of the value function, average-reward, and subtask estimates of the RED Q-learning algorithm. Similar to what was done in Wan et al. (2021), we will begin by considering a general algorithm, called *General RED Q*. We will first define General RED Q, then show how the RED Q-learning algorithm is a special case of this algorithm. We will then provide necessary assumptions, state the convergence theorem of General RED Q, and then provide a proof for the theorem, where we show that the value function, average-reward, and subtask estimates converge, thereby showing that the RED Q-learning algorithm converges. We begin by introducing the General RED Q algorithm:

First consider an MDP $\mathcal{M} \doteq \langle \mathcal{S}, \mathcal{A}, \mathcal{R}, p \rangle$. Given a state $s \in \mathcal{S}$, action $a \in \mathcal{A}$, and discrete step $n \geq 0$, let $R_n(s,a) \in \mathcal{R}$ denote a sample of the resulting reward, and let $S'_n(s,a) \sim p(\cdot \mid s,a)$ denote a sample of the resulting state. Let $\{Y_n\}$ be a set-valued process taking values in the set of nonempty subsets of $\mathcal{S} \times \mathcal{A}$, such that: $Y_n = \{(s,a) : (s,a) \text{ component of the } |\mathcal{S} \times \mathcal{A}|\text{-sized table of state-action value estimates, } Q, \text{ that was updated at step } n\}$. Let $\nu(n,s,a) \doteq \sum_{j=0}^{n} I\{(s,a) \in Y_j\}$, where $I$ is the indicator function, such that $\nu(n,s,a)$ represents the number of times the $(s,a)$ component of $Q$ was updated up to step $n$.

Now, let $f$ be a valid subtask function (see Definition 5.1), such that $\tilde{R}_n(s,a) \doteq f(R_n(s,a), Z_{1,n}, Z_{2,n}, \dots, Z_{n,k})$ for $k$ subtasks $\in \mathcal{Z}$, where $\tilde{R}_n(s,a)$ is the extended reward, $\mathcal{Z}$ is the set of subtasks, and $Z_{i,n}$ denotes the estimate of subtask $z_i \in \mathcal{Z}$ at step $n$. Consider an MDP with the extended reward: $\tilde{\mathcal{M}} \doteq \langle \mathcal{S}, \mathcal{A}, \tilde{\mathcal{R}}, p \rangle$, such that $\tilde{R}_n(s,a) \in \tilde{\mathcal{R}}$. The update rules of General RED Q for this MDP are as follows:

$$Q_{n+1}(s,a) \doteq Q_n(s,a) + \alpha_{\nu(n,s,a)} \delta_n(s,a) I\{(s,a) \in Y_n\}, \quad \forall s \in \mathcal{S}, a \in \mathcal{A}, \tag{C.28}$$

$$\bar{R}_{n+1} \doteq \bar{R}_n + \eta_r \sum_{s,a} \alpha_{\nu(n,s,a)} \delta_n(s,a) I\{(s,a) \in Y_n\}, \tag{C.29}$$

$$Z_{i,n+1} \doteq Z_{i,n} + \eta_{z_i} \sum_{s,a} \alpha_{\nu(n,s,a)} \beta_{i,n}(s,a) I\{(s,a) \in Y_n\}, \quad \forall z_i \in \mathcal{Z} \tag{C.30}$$

where,

$$\delta_n(s,a) \doteq \tilde{R}_n(s,a) - \bar{R}_n + \max_{a'} Q_n(S'_n(s,a), a') - Q_n(s,a)$$
$$= f(R_n(s,a), Z_{1,n}, Z_{2,n}, \dots, Z_{k,n}) - \bar{R}_n + \max_{a'} Q_n(S'_n(s,a), a') - Q_n(s,a), \tag{C.31}$$

and,

$$\beta_{i,n}(s,a) \doteq \phi_{z_i,n} - Z_{i,n}, \quad \forall z_i \in \mathcal{Z}. \tag{C.32}$$

Here, $\bar{R}_n$ denotes the estimate of the average-reward (see Equation 2), $\delta_n(s,a)$ denotes the TD error, $\eta_r$ and $\eta_{z_i}$ are positive scalars, $\phi_{z_i,n}$ denotes the inverse of the TD error (i.e., Equation C.31) with respect to subtask estimate $Z_{i,n}$ given all other inputs when $\delta_n(s,a) = 0$, and $\alpha_{\nu(n,s,a)}$ denotes the step size at time step $n$ for state-action pair $(s,a)$. In this case, the step size depends on the sequence $\{\alpha_n\}$, as well as the number of visitations to the state-action pair $(s,a)$, which is denoted by $\nu(n,s,a)$.

We now show that the RED Q-learning algorithm is a special case of the General RED Q algorithm. Consider a sequence of experience from our MDP $\tilde{\mathcal{M}}$: $S_t, A_t, \tilde{R}_{t+1}, S_{t+1}, \ldots$ . Now recall the set-valued process $\{Y_n\}$. If we let $n$ = time step $t$, we have:

$$Y_t(s,a) = \begin{cases} 1, s = S_t \text{ and } a = A_t, \\ 0, \text{ otherwise,} \end{cases}$$

as well as $S'_n(S_t, A_t) = S_{t+1}$, $R_n(S_t, A_t) = R_{t+1}$, and $\tilde{R}_n(S_t, A_t) = \tilde{R}_{t+1}$.

Hence, update rules C.28, C.29, C.30, C.31, and C.32 become:

$$Q_{t+1}(S_t, A_t) \doteq Q_t(S_t, A_t) + \alpha_{\nu(t,S_t,A_t)}\delta_t \tag{C.33}$$

$$Q_{t+1}(s,a) \doteq Q_t(s,a), \forall s \neq S_t, a \neq A_t, \tag{C.34}$$

$$\bar{R}_{t+1} \doteq \bar{R}_t + \eta_r \alpha_{\nu(t,S_t,A_t)}\delta_t, \tag{C.35}$$

$$Z_{i,t+1} \doteq Z_{i,t} + \eta_{z_i} \alpha_{\nu(t,S_t,A_t)}\beta_{i,t}, \quad \forall z_i \in \mathcal{Z}, \tag{C.36}$$

$$\delta_t \doteq \tilde{R}_{t+1} - \bar{R}_t + \max_{a'} Q_t(S_{t+1}, a') - Q_t(S_t, A_t),$$
$$= f(R_{t+1}, Z_{1,t}, Z_{2,t}, \ldots, Z_{k,t}) - \bar{R}_t + \max_{a'} Q_t(S_{t+1}, a') - Q_t(S_t, A_t), \tag{C.37}$$

$$\beta_{i,t} \doteq \phi_{z_i,t} - Z_{i,t}, \quad \forall z_i \in \mathcal{Z}, \tag{C.38}$$

which are RED Q-learning's update rules with $\alpha_{\nu(t,S_t,A_t)}$ denoting the step size at time $t$.

We now specify the assumptions on General RED Q that are needed to ensure convergence. Please refer to Wan et al. (2021) for an in-depth discussion on these assumptions:

**Assumption C.8** (Communicating Assumption). *The MDP has a single communicating class. That is, each state in the MDP is accessible from every other state under some deterministic stationary policy.*

**Assumption C.9** (Action-Value Function Uniqueness). *There exists a unique solution of $q$ only up to a constant in the Bellman equation 4.*

**Assumption C.10** (Asynchronous Step Size Assumption 3). *There exists $\Delta > 0$ such that*

$$\liminf_{n\to\infty} \frac{\nu(n,s,a)}{n+1} \geq \Delta,$$

*a.s., for all $s \in \mathcal{S}, a \in \mathcal{A}$.*

*Furthermore, for all $x > 0$, and*

$$N(n,x) = \min\left\{m > n : \sum_{i=n+1}^{m} \alpha_i \geq x\right\},$$

*the limit*

$$\lim_{n\to\infty} \frac{\sum_{i=\nu(n,s,a)}^{\nu(N(n,x),s,a)} \alpha_i}{\sum_{i=\nu(n,s',a')}^{\nu(N(n,x),s',a')} \alpha_i}$$

*exists a.s. for all $s, s', a, a'$.*

We next point out that it is easy to verify that under Assumption C.8, the following system of equations:

$$q_\pi(s, a) = \sum_{s', \tilde{r}} p(s', \tilde{r} \mid s, a)(\tilde{r} - \bar{r}_\pi + \max_{a'} q_\pi(s, a)), \quad \forall s \in \mathcal{S}, a \in \mathcal{A},$$

$$= \sum_{s', r} p(s', r \mid s, a)(f(r, z_1, z_2, \dots, z_k) - \bar{r}_\pi + \max_{a'} q_\pi(s, a)), \tag{C.39}$$

and,

$$\bar{r}_* - \bar{R}_0 = \eta_r \left( \sum q_\pi - \sum Q_0 \right), \tag{C.40}$$

$$z_{i_*} - Z_{i,0} = \eta_i \left( \sum q_\pi - \sum Q_0 \right), \quad \forall z_i \in \mathcal{Z}, \tag{C.41}$$

has a unique solution for $q_\pi$, where $\bar{r}_*$ denotes the optimal average-reward, and $z_{i_*}$ denotes the corresponding optimal subtask value for subtask $z_i \in \mathcal{Z}$. Denote this unique solution for $q_\pi$ as $q_*$.

We are now ready to state the convergence theorem:

**Theorem C.2.1** (Convergence of General RED Q). *If Assumptions C.3, C.4, C.6, C.7, C.8, C.9, and C.10 hold, then the General RED Q algorithm (Equations C.28–C.32) converges a.s. $\bar{R}_n$ to $\bar{r}_*$, $Z_{i,n}$ to $z_{i_*}$ $\forall z_i \in \mathcal{Z}$, $Q_n$ to $q_*$, $\bar{r}_{\pi_t}$ to $\bar{r}_*$, and $z_{i,\pi_t}$ to $z_{i_*}$ $\forall z_i \in \mathcal{Z}$, where $\pi_t$ is any greedy policy with respect to $Q_t$, and $z_{i,\pi_t}$ denotes the subtask value induced by following policy $\pi_t$.*

We prove this theorem in the following section. To do so, we first show that General RED Q is of the same form as *General Differential Q* from Wan et al. (2021), which consequently allows us to apply their convergence results for the value function and average-reward estimates of General Differential Q to General RED Q. We then build upon these results, using similar techniques as Wan et al. (2021), to show that the subtask estimates converge as well.

### C.2.1 PROOF OF THEOREM C.2.1

**Convergence of the average-reward and state-action value function estimates:**

Consider the increment to $\bar{R}_n$ at each step. We can see from Equation C.29 that the increment is $\eta_r$ times the increment to $Q_n$. As such, as was done in Wan et al. (2021), we can write the cumulative increment as follows:

$$\bar{R}_n - \bar{R}_0 = \eta_r \sum_{j=0}^{n-1} \sum_{s,a} \alpha_{\nu(j,s,a)} \delta_j(s, a) I\{(s, a) \in Y_j\}$$

$$= \eta_r \left( \sum Q_n - \sum Q_0 \right)$$

$$\implies \bar{R}_n = \eta_r \sum Q_n - \eta_r \sum Q_0 + \bar{R}_0 = \eta_r \sum Q_n - c, \tag{C.42}$$

$$\text{where } c \doteq \eta_r \sum Q_0 - \bar{R}_0. \tag{C.43}$$

We can then substitute $\bar{R}_n$ in C.28 with C.42 $\forall s \in \mathcal{S}, a \in \mathcal{A}$, which yields:

$$Q_{n+1}(s, a) = Q_n(s, a) + \dots$$

$$\alpha_{\nu(n,s,a)} \left( \tilde{R}_n(s, a) + \max_{a'} Q_n(S'_n(s, a), a') - Q_n(s, a) - \eta_r \sum Q_n + c \right) I\{(s, a) \in Y_n\}$$

$$Q_{n+1}(s, a) = Q_n(s, a) + \dots$$

$$\alpha_{\nu(n,s,a)} \left( \hat{R}_n(s, a) + \max_{a'} Q_n(S'_n(s, a), a') - Q_n(s, a) - \eta_r \sum Q_n \right) I\{(s, a) \in Y_n\}, \tag{C.44}$$

where, $\hat{R}_n(s,a) \doteq \tilde{R}_n(s,a) + c = f(R_n(s,a), Z_{1,n}, Z_{2,n}, \ldots, Z_{k,n}) + c$.

Equation C.44 is now in the same form as Equation (B.14) (i.e., General Differential Q) from Wan et al. (2021), who showed that the equation converges a.s. $Q_n$ to $q_*$ as $n \to \infty$. Moreover, from this result, Wan et al. (2021) showed that $\bar{R}_n$ converges a.s. to $\bar{r}_*$ as $n \to \infty$, and that $\bar{r}_{\pi_t}$ converges a.s. to $\bar{r}_*$, where $\pi_t$ is a greedy policy with respect to $Q_t$. Given that General RED Q adheres to all the assumptions listed for General Differential Q in Wan et al. (2021), these convergence results apply to General RED Q.

**Convergence of the subtask estimates:**

Consider the increment to $Z_{i,n}$ (for an arbitrary subtask $z_i \in \mathcal{Z}$) at each step. Given Equation 14, we can write the increment in Equation C.30 as some constant, subtask-specific fraction of the increment to $Q_n$. Consequently, we can write the cumulative increment as follows:

$$Z_{i,n} - Z_{i,0} = \eta_{z_i} \sum_{j=0}^{n-1} \sum_{s,a} \alpha_{\nu(j,s,a)} \beta_{i,j}(s,a) I\{(s,a) \in Y_j\}$$

$$= \eta_{z_i} \sum_{j=0}^{n-1} \sum_{s,a} \alpha_{\nu(j,s,a)} b_i \delta_j(s,a) I\{(s,a) \in Y_j\}$$

$$= \eta_i \left( \sum Q_n - \sum Q_0 \right)$$

$$\implies Z_{i,n} = \eta_i \sum Q_n - \eta_i \sum Q_0 + Z_{i,0} = \eta_i \sum Q_n - c, \qquad (C.45)$$

where,

$$c \doteq \eta_i \sum Q_0 - Z_{i,0}, \text{ and} \qquad (C.46)$$

$$\eta_i \doteq \eta_{z_i} b_i. \qquad (C.47)$$

Now, let $f(Z_{i,n})$ be shorthand for the subtask function (i.e., $\tilde{R}_n(s,a)$). We can substitute $Z_{i,n}$ in C.28 with C.45 $\forall s \in \mathcal{S}, a \in \mathcal{A}$ as follows:

$$Q_{n+1}(s,a) = Q_n(s,a) + \ldots$$
$$\alpha_{\nu(n,s,a)} \left( \tilde{R}_n(s,a) - \bar{R} + \max_{a'} Q_n(S'_n(s,a), a') - Q_n(s,a) \right) I\{(s,a) \in Y_n\}$$

$$\implies Q_{n+1}(s,a) = Q_n(s,a) + \ldots$$
$$\alpha_{\nu(n,s,a)} \left( f(Z_{i,n}) - \bar{R} + \max_{a'} Q_n(S'_n(s,a), a') - Q_n(s,a) \right) I\{(s,a) \in Y_n\}$$

$$\implies Q_{n+1}(s,a) = Q_n(s,a) + \ldots$$
$$\alpha_{\nu(n,s,a)} \left( f(\underbrace{\eta_i \sum Q_n - c}_{\hat{Z}_{i,n}}) - \bar{R} + \max_{a'} Q_n(S'_n(s,a), a') - Q_n(s,a) \right) I\{(s,a) \in Y_n\}$$

$$\implies Q_{n+1}(s,a) = Q_n(s,a) + \ldots$$
$$\alpha_{\nu(n,s,a)} \left( \hat{f}(\hat{Z}_{i,n}) - \bar{R} + \max_{a'} Q_n(S'_n(s,a), a') - Q_n(s,a) \right) I\{(s,a) \in Y_n\}$$

$$\implies Q_{n+1}(s,a) = Q_n(s,a) + \ldots$$
$$\alpha_{\nu(n,s,a)} \left( \hat{R}_n - \bar{R} + \max_{a'} Q_n(S'_n(s,a),a') - Q_n(s,a) \right) I\{(s,a) \in Y_n\}, \tag{C.48}$$

where, $\hat{R}_n \doteq \hat{f}(\hat{Z}_{i,n}) = f(Z_{i,n}+c) = h(\tilde{R}_n)$. Here, $h(\tilde{R}_n)$ corresponds to the change in $\tilde{R}_n$ due to shifting subtask $Z_{i,n}$ by $c$. Denote the inverse of $h(\tilde{R}_n)$ (which exists given Assumption C.6) as $h^{-1}$.

Now consider an MDP, $\hat{\mathcal{M}}$, which has rewards, $\hat{\mathcal{R}}$, corresponding to rewards modified by $h$ from the MDP $\tilde{\mathcal{M}}$, has the same state and action spaces as $\tilde{\mathcal{M}}$, and has the transition probabilities defined as:

$$\hat{p}\left(s', h(\tilde{r}) \mid s, a\right) \doteq p(s', \tilde{r} \mid s, a), \tag{C.49}$$

such that $\hat{\mathcal{M}} \doteq \langle \mathcal{S}, \mathcal{A}, \hat{\mathcal{R}}, \hat{p} \rangle$. It is easy to check that the communicating assumption holds for the transformed MDP $\hat{\mathcal{M}}$. As such, and given Assumptions C.6 and C.7, the optimal average-reward for the MDP $\hat{\mathcal{M}}$, $\hat{\bar{r}}_*$, can be written as follows:

$$\hat{\bar{r}}_* = h(\bar{r}_*). \tag{C.50}$$

Now, because

$$q_*(s,a) = \sum_{s',\tilde{r}} p(s', \tilde{r} \mid s, a)(\tilde{r} + \max_{a'} q_*(s',a') - \bar{r}_*) \quad \text{(from C.39)}$$

$$= \sum_{s',\tilde{r}} p(s', \tilde{r} \mid s, a)(\tilde{r} + \max_{a'} q_*(s',a') - h^{-1}(\hat{\bar{r}}_*)) \quad \text{(from C.50)}$$

$$= \sum_{s',\tilde{r}} p(s', \tilde{r} \mid s, a)(h(\tilde{r}) + \max_{a'} q_*(s',a') - \hat{\bar{r}}_*) \quad \text{(by linearity of } h)$$

$$= \sum_{s',\tilde{r}} \hat{p}(s', \tilde{r} \mid s, a)(\tilde{r} + \max_{a'} q_*(s',a') - \hat{\bar{r}}_*) \quad \text{(from C.49)},$$

we can see that $q_*$ is a solution of not just the state-action value Bellman equation for the MDP $\tilde{\mathcal{M}}$, but also the state-action value Bellman equation for the transformed MDP $\hat{\mathcal{M}}$.

Next, we can write the optimal subtask value for the MDP $\hat{\mathcal{M}}$, $\hat{z}_{i_*}$, as follows:
$$\hat{z}_{i_*} = z_{i_*} + c. \tag{C.51}$$

We can then combine Equations C.41, C.46, and C.51, which yields:
$$\hat{z}_{i_*} = \eta_i \sum q_*. \tag{C.52}$$

Next, we can combine Equation C.45 with the result from Wan et al. (2021) which shows that $Q_n \to q_*$, which yields:
$$Z_{i,n} \to \eta_i \sum q_* - c. \tag{C.53}$$

Moreover, because $\eta_i \sum q_* = \hat{z}_{i_*}$ (Equation C.52), we have:
$$Z_{i,n} \to \hat{z}_{i_*} - c. \tag{C.54}$$

Finally, because $\hat{z}_{i_*} = z_{i_*} + c$ (Equation C.51), we have:
$$Z_{i,n} \to z_{i_*} \quad \text{a.s. as} \quad n \to \infty. \tag{C.55}$$

We conclude by considering $z_{i,\pi_t} \ \forall z_i \in \mathcal{Z}$, where $\pi_t$ is a greedy policy with respect to $Q_t$. Given Definition 5.1, and that $\bar{r}_{\pi_t} \to \bar{r}_*$ a.s., it directly follows that $z_{i,\pi_t} \to z_{i_*} \ \forall z_i \in \mathcal{Z}$ a.s.

This completes the proof of Theorem C.2.1.

## D  LEVERAGING THE RED RL FRAMEWORK FOR CVAR OPTIMIZATION

This appendix contains details regarding the adaptation of the RED RL framework for CVaR optimization. We first derive an appropriate subtask function, then use it to adapt the RED RL algorithms (see Appendix B) for CVaR optimization. In doing so, we arrive at the *RED CVaR algorithms*, which are presented in full at the end of this appendix. These RED CVaR algorithms allow us to optimize CVaR (and VaR) without the use of an augmented state-space or an explicit bi-level optimization. We also provide a convergence proof for the tabular RED CVaR Q-learning algorithm, which shows that the VaR and CVaR estimates converge to the optimal long-run VaR and CVaR, respectively.

### D.1  A SUBTASK-DRIVEN APPROACH FOR CVAR OPTIMIZATION

In this section, we use the RED RL framework to derive a subtask-driven approach for CVaR optimization that does not require an augmented state-space or an explicit bi-level optimization. To begin, let us consider Equation 7, which is displayed below as Equation D.1 for convenience:

$$\text{CVaR}_\tau(R_t) = \sup_{b \in \mathbb{R}} \mathbb{E}[b - \frac{1}{\tau}(b - R_t)^+] \tag{D.1a}$$

$$= \mathbb{E}[\text{VaR}_\tau(R_t) - \frac{1}{\tau}(\text{VaR}_\tau(R_t) - R_t)^+], \tag{D.1b}$$

where the CVaR parameter, $\tau \in (0,1)$, represents the left $\tau$-quantile of the random variable, $R_t$, which corresponds to the observed per-step reward from the MDP.

We can see from Equation D.1 that CVaR can be interpreted as an expectation (or average) of sorts, which suggests that it may be possible to leverage the average-reward MDP to optimize this expectation, by treating the reward CVaR as the $\bar{r}_\pi$ that we want to optimize. However, this requires that we know the optimal value of the scalar $b$, because the expectation in Equation D.1b only holds for this optimal value (which corresponds to the per-step reward VaR). Unfortunately, this optimal value is typically not known beforehand, so in order to optimize CVaR, we also need to optimize $b$.

Importantly, we can utilize RED RL framework to turn the optimization of $b$ into a subtask, such that CVaR is the primary control objective (i.e., the $\bar{r}_\pi$ that we want to optimize), and VaR ($b$ in Equation D.1), is the (single) subtask. This is in contrast to existing MDP-based methods, which typically leverage Equation D.1 when optimizing for CVaR by augmenting the state-space with an estimate of $\text{VaR}_\tau(R)$ (in this case, $b$), and solving the bi-level optimization shown in Equation 8, thereby increasing computational costs.

First, we need to derive a valid subtask function for CVaR that satisfies the requirements of Definition 5.1. As a starting point, let us consider Equation D.1. We can see that if we treat the expression inside the expectation in Equation D.1 as our subtask function, $f$ (see Definition 5.1), then we have a piecewise linear subtask function that is invertible with respect to each input given all other inputs, where the subtask, VaR, is independent of the observed per-step reward. Now, if we directly used this expression as the subtask function and applied the RED RL framework, then we would have an estimate of VaR, in this case $b$, that we would try to optimize in the hopes that we eventually find our desired solution, such that $b = \text{VaR}$. However, there is nothing in this tentative subtask function that incentivizes our algorithm to seek out an estimate of $b$ that is close to the actual VaR value. This means that, hypothetically, our algorithm could find some optimal solution such that $b \neq \text{VaR}$.

Hence, we need to modify Equation D.1 in such a way that incentivizes our algorithm to seek out an estimate of $b$ that is close to the actual VaR value. It turns out that we can make the appropriate modification to Equation D.1 by leveraging a concept from quantile regression (Koenker, 2005). Quantile regression refers to the process of estimating a predetermined quantile of a probability distribution from samples. More specifically, let $\tau \in (0,1)$ be the $\tau$th quantile (or percentile) that we are trying to estimate from probability distribution $w$. Hence, the value that we are interested in estimating is $F_w^{-1}(\tau)$. Quantile regression maintains an estimate, $\theta$, of this value, and updates the estimate based on samples drawn from $w$ (i.e., $y \sim w$) as follows:

$$\theta \leftarrow \theta + \eta_\theta(\tau - \mathbb{1}_{\{y < \theta\}}), \tag{D.2}$$

where $\eta_\theta$ is the step size for the update. The estimate for $\theta$ will continue to adjust until the equilibrium point, $\theta^*$, which corresponds to $F_w^{-1}(\tau)$, is reached. In other words, we have:

$$0 = \mathbb{E}[(\tau - \mathbb{1}_{\{y < \theta^*\}})] \tag{D.3a}$$

$$= \tau - \mathbb{E}[\mathbb{1}_{\{y < \theta^*\}}] \tag{D.3b}$$

$$= \tau - \mathbb{P}(y < \theta^*) \tag{D.3c}$$

$$\implies \theta^* = F_w^{-1}(\tau). \tag{D.3d}$$

Equivalently, we also have:

$$0 = \mathbb{E}[((1 - \tau) - \mathbb{1}_{\{y \geq \theta^*\}})] \tag{D.4a}$$

$$= (1 - \tau) - \mathbb{E}[\mathbb{1}_{\{y \geq \theta^*\}}] \tag{D.4b}$$

$$= (1 - \tau) - \mathbb{P}(y \geq \theta^*) \tag{D.4c}$$

$$\implies \theta^* = F_w^{-1}(\tau). \tag{D.4d}$$

In our case, we are not interested in performing quantile regression as described in Equation D.2 (as we will show later in this section, the RED RL framework allows us to use the TD error to update our estimate of the desired quantile, VaR). However, we can augment Equation D.1 with Equations D.3 and D.4 as follows:

$$\text{CVaR}_\tau(R_t) = \sup_{b \in \mathbb{R}} \mathbb{E}[b - \frac{1}{\tau}(b - R_t)^+] \tag{D.5a}$$

$$= \sup_{b \in \mathbb{R}} \left\{ \mathbb{E}[b - \frac{1}{\tau}(b - R_t)^+] - 0 - 0 \right\} \tag{D.5b}$$

$$= \sup_{b \in \mathbb{R}} \left\{ \mathbb{E}[b - \frac{1}{\tau}(b - R_t)^+] - c_1 0 - c_2 0 \right\} \tag{D.5c}$$

$$= \sup_{b \in \mathbb{R}} \left\{ \mathbb{E}[b - \frac{1}{\tau}(b - R_t)^+] - c_1 \mathbb{E}[(\tau - \mathbb{1}_{\{R_t < b\}})] - c_2 \mathbb{E}[((1 - \tau) - \mathbb{1}_{\{R_t \geq b\}})] \right\} \tag{D.5d}$$

$$= \sup_{b \in \mathbb{R}} \left\{ \mathbb{E}\left[ b - \frac{1}{\tau}(b - R_t)^+ - c_1(\tau - \mathbb{1}_{\{R_t < b\}}) - c_2((1 - \tau) - \mathbb{1}_{\{R_t \geq b\}}) \right] \right\} \tag{D.5e}$$

$$= \mathbb{E}[\text{VaR}_\tau(R_t) - \frac{1}{\tau}(\text{VaR}_\tau(R_t) - R_t)^+ - c_1\left(\tau - \mathbb{1}_{\{R_t < \text{VaR}_\tau(R_t)\}}\right) \cdots$$
$$- c_2\left((1 - \tau) - \mathbb{1}_{\{R_t \geq \text{VaR}_\tau(R_t)\}}\right)], \tag{D.5f}$$

where, $c_1$ and $c_2$ are positive scalars. Here, we have essentially added a 'penalty' into the expectation for having a VaR estimate that does not equal the actual VaR value. With this, we have narrowed the set of possible solutions that maximize the expectation, to those that have an acceptable VaR estimate. Consequently, we can now adapt Equation D.5 as our subtask function, as follows:

$$\tilde{R}_t = \text{VaR} - \frac{1}{\tau}(\text{VaR} - R_t)^+ - c_1\left(\tau - \mathbb{1}_{\{R_t < \text{VaR}\}}\right) - c_2\left((1 - \tau) - \mathbb{1}_{\{R_t \geq \text{VaR}\}}\right), \tag{D.6}$$

where, $R_t$ is the observed per-step reward, $\tilde{R}_t$ is the extended per-step reward, VaR is the value-at-risk of the observed per-step reward, $\tau$ is the CVaR parameter, and $c_1$ and $c_2$ are positive scalars. Empirically, we found that setting $c_1 = 1.0$ and $c_2 = (1 - \tau)$ yielded good results. Importantly, this is a valid subtask function with the following properties: the average (or expected value) of the extended reward corresponds to the CVaR of the observed reward, and the optimal average of the extended reward corresponds to the optimal CVaR of the observed reward. This is formalized as Corollaries D.1 - D.4 below:

**Corollary D.1.** *The function presented in Equation D.6 is a valid subtask function.*

*Proof.* The function presented in Equation D.6 is clearly a piecewise linear function that is invertible with respect to each input given all other inputs. Moreover, the subtask, VaR, is independent of the observed per-step reward. Hence, this function satisfies Definition 5.1 for the subtask, VaR. □

**Corollary D.2.** *If the subtask, VaR (from Equation D.6) is estimated, and such an estimate is equal to the long-run VaR of the observed reward, then the average (or expected value) of the extended reward, $\tilde{R}_t$, from Equation D.6 is equal to the long-run CVaR of the observed reward.*

*Proof.* This follows directly from Equation D.5f. □

**Corollary D.3.** *If the subtask, VaR (from Equation D.6) is estimated, and the resulting average of the extended reward from Equation D.6 is equal to the long-run CVaR of the observed reward, then the VaR estimate is equal to the long-run VaR of the observed reward.*

*Proof.* This follows directly from Equation D.5f. □

**Corollary D.4.** *A policy that yields an optimal long-run average of the extended reward, $\tilde{R}_t$, from Equation D.6 is a CVaR-optimal policy. In other words, the optimal long-run average of the extended reward corresponds to the optimal long-run CVaR of the observed reward.*

*Proof.* For a given policy, we know from Equation D.5e that, across a range of VaR estimates, the best possible long-run average of the extended reward for that policy corresponds to the long-run CVaR of the observed reward for that same policy. Hence, the best possible long-run average of the extended reward that can be achieved across various policies and VaR estimates, corresponds to the optimal long-run CVaR of the observed reward. □

As such, we now have a valid subtask function with a subtask, VaR, and an extended reward average that when optimized, corresponds to the optimal CVaR of the observed reward. We are now ready to apply the RED RL framework. First, we derive the learning update for our subtask, VaR, using the methods shown in Theorem 5.1. In particular, we provide a theorem, which is a CVaR-specific version of Theorem 5.1, which shows that we can optimize our subtask, VaR, using the TD error.

**Theorem D.1.1.** *Given the subtask function presented in Equation D.6, an average-reward MDP can optimize the VaR estimate using the TD error.*

*Proof.* From Corollary D.1, we know that $\tilde{R}_t = \text{VaR} - \frac{1}{\tau}(\text{VaR} - R_t)^+ - c_1\left(\tau - \mathbb{1}_{\{R_t < \text{VaR}\}}\right) - c_2\left((1 - \tau) - \mathbb{1}_{\{R_t \geq \text{VaR}\}}\right)$ is a valid subtask function (as per Definition 5.1), where $R_t$ is the observed per-step reward, $\tilde{R}_t$ is the extended per-step reward whose long-run average, $\bar{r}_\pi$, is the primary prediction or control objective, $\tau$ is the CVaR parameter, and $c_1$ and $c_2$ are positive scalars.

We can write the TD error for the control case as follows:

$$\delta_t = \tilde{R}_t - \bar{R}_t + \max_{a'} Q_t(S_{t+1}, a') - Q_t(S_t, A_t) \tag{D.7a}$$

$$\begin{aligned} = \text{VaR}_t - \frac{1}{\tau}(\text{VaR}_t - R_t)^+ - c_1\left(\tau - \mathbb{1}_{\{R_t < \text{VaR}_t\}}\right) \cdots \\ - c_2\left((1 - \tau) - \mathbb{1}_{\{R_t \geq \text{VaR}_t\}}\right) - \bar{R}_t + \max_{a'} Q_t(S_{t+1}, a') - Q_t(S_t, A_t). \end{aligned} \tag{D.7b}$$

where $Q_t : \mathcal{S} \times \mathcal{A} \to \mathbb{R}$ denotes a table of state-action value function estimates, $\bar{R}_t$ denotes an estimate of the average-reward, $\bar{r}_\pi$, and $\text{VaR}_t$ is the VaR estimate at time $t$.

Similarly, we can incorporate the subtask function into the Bellman equation 4 and solve for VaR as follows:

$$q_\pi(s,a) = \mathbb{E}_\pi[\tilde{R}_t - \bar{r}_\pi + \max_{a'} q_\pi(S_{t+1}, a') \mid S_t = s, A_t = a] \tag{D.8a}$$

$$= \mathbb{E}_\pi[\text{VaR} - \frac{1}{\tau}(\text{VaR} - R_t)^+ - c_1\left(\tau - \mathbb{1}_{\{R_t < \text{VaR}\}}\right) \cdots$$
$$- c_2\left((1-\tau) - \mathbb{1}_{\{R_t \geq \text{VaR}\}}\right) - \bar{r}_\pi + \max_{a'} q_\pi(S_{t+1}, a') \mid S_t = s, A_t = a] \tag{D.8b}$$

$$\implies \text{VaR} = \mathbb{E}[\phi_{\text{VaR},t} \mid S_t = s, A_t = a], \tag{D.8c}$$

where,

$$\phi_{\text{VaR},t} = \begin{cases} -(-c_1\tau + c_2\tau - \bar{r}_\pi + \max_{a'} q_\pi(S_{t+1}, a') - q_\pi(s,a)), R_t \geq \text{VaR} \\\\ (\frac{\tau}{1-\tau})(\frac{1}{\tau}R_t - c_1(\tau - 1) - c_2(1-\tau) - \bar{r}_\pi + \max_{a'} q_\pi(S_{t+1}, a') - q_\pi(s,a)), R_t < \text{VaR}. \end{cases} \tag{D.8d}$$

Here, we use $\phi_{\text{VaR},t}$ to denote the (piecewise) expression inside the expectation in Equation D.8c. Thus, as in Theorem 5.1, we can utilize the common RL update rule to learn VaR from experience, which yields the update:

$$\text{VaR}_{t+1} = \text{VaR}_t + \eta\alpha_t[\phi_{\text{VaR},t} - \text{VaR}_t], \tag{D.9}$$

where, $\text{VaR}_t$ is the VaR estimate at time $t$, and $\eta\alpha_t$ is the step size. With some algebra, the above expression can be re-written in terms of the TD error (see Equation D.7), as follows:

$$\text{VaR}_{t+1} = \begin{cases} \text{VaR}_t - \eta\alpha_t\delta_t, & R_t \geq \text{VaR}_t \\ \text{VaR}_t + \eta\alpha_t(\frac{\tau}{1-\tau})\delta_t, & R_t < \text{VaR}_t \end{cases}, \tag{D.10}$$

where, $R_t$ is the observed reward at time $t$, $\delta_t$ is the TD error, $\eta\alpha_t$ is the step size, and $\tau$ is the CVaR parameter. Importantly, this implies that minimizing the TD error is equivalent to optimizing the VaR estimate. This completes the proof of Theorem D.1.1. □

Hence, we now have an update rule that allows us to optimize VaR using the TD error. Importantly, this means that we now have all the components needed to utilize the RED algorithms in Appendix B to optimize CVaR (where CVaR corresponds to the $\bar{r}_\pi$ that we want to optimize). We call these CVaR-specific algorithms, the *RED CVaR algorithms*. The full algorithms are included at the end of this appendix. We now present the tabular RED CVaR Q-learning algorithm, along with a convergence proof which shows that the VaR and CVaR estimates converge to the optimal long-run VaR and CVaR of the observed reward, respectively.

**RED CVaR Q-learning algorithm (tabular):** We update a table of estimates, $Q_t : \mathcal{S} \times \mathcal{A} \to \mathbb{R}$ as follows:

$$\tilde{R}_t = \text{VaR}_t - \frac{1}{\tau}(\text{VaR}_t - R_t)^+ - c_1\left(\tau - \mathbb{1}_{\{R_t < \text{VaR}_t\}}\right) - c_2\left((1-\tau) - \mathbb{1}_{\{R_t \geq \text{VaR}_t\}}\right) \tag{D.11a}$$

$$\delta_t = \tilde{R}_{t+1} - \text{CVaR}_t + \max_a Q_t(S_{t+1}, a) - Q_t(S_t, A_t) \tag{D.11b}$$

$$Q_{t+1}(S_t, A_t) = Q_t(S_t, A_t) + \alpha_t\delta_t \tag{D.11c}$$

$$Q_{t+1}(s,a) = Q_t(s,a), \quad \forall s, a \neq S_t, A_t \tag{D.11d}$$

$$\text{CVaR}_{t+1} = \text{CVaR}_t + \eta_{\text{CVaR}}\alpha_t\rho_t\delta_t \tag{D.11e}$$

$$\text{VaR}_{t+1} = \begin{cases} \text{VaR}_t - \eta_{\text{VaR}}\alpha_t\delta_t, R_t \geq \text{VaR}_t \\ \text{VaR}_t + \eta_{\text{VaR}}\alpha_t(\frac{\tau}{1-\tau})\delta_t, R_t < \text{VaR}_t \end{cases}, \tag{D.11f}$$

where, $R_t$ is the observed reward, $\text{VaR}_t$ is the VaR estimate, $\alpha_t$ is the step size, $\delta_t$ is the TD error, $\text{CVaR}_t$ is the CVaR estimate, and $\eta_{\text{CVaR}}, \eta_{\text{VaR}}$ are positive scalars.

We now provide a theorem for our tabular RED CVaR Q-learning algorithm, which shows that $\text{CVaR}_t$ converges to the optimal long-run CVaR of the observed reward, $\text{CVaR}^*$, $\text{VaR}_t$ converges to the optimal long-run VaR of the observed reward, $\text{VaR}^*$, and $Q_t$ converges to a solution of $q$ in Equation 4, all up to an additive constant:

**Theorem D.1.2.** *The RED CVaR Q-learning algorithm D.11 converges, almost surely, $\text{CVaR}_t$ to $\text{CVaR}^*$, $\text{VaR}_t$ to $\text{VaR}^*$, $\text{CVaR}_{\pi_t}$ to $\text{CVaR}^*$, $\text{VaR}_{\pi_t}$ to $\text{VaR}^*$, and $Q_t$ to a solution of $q$ in the Bellman Equation 4, up to an additive constant, $c$, where $\pi_t$ is any greedy policy with respect to $Q_t$, if the following assumptions hold: 1) the MDP is communicating, 2) the solution of $q$ in 4 is unique up to a constant, 3) the step sizes are decreased appropriately, 4) all the state–action pairs are updated an infinite number of times, 5) the ratio of the update frequency of the most-updated state–action pair to the least-updated state–action pair is finite, and 6) the subtask function is in accordance with Definition 5.1.*

*Proof.* By definition, the RED CVaR Q-learning algorithm D.11 is of the form of the generic RED Q-learning algorithm 16, where $\text{CVaR}_t$ corresponds to $\bar{R}_t$ and $\text{VaR}_t$ corresponds to $Z_{i,t}$ for a single subtask. We also know from Corollary D.1 that the subtask function used is valid. Hence, Theorem 5.3 applies, such that:

*i)* $\text{CVaR}_t$ and $\text{CVaR}_{\pi_t}$ converge a.s. to the optimal long-run average, $\bar{r}*$, of the extended reward from the subtask function (i.e., the optimal long-run average of $\tilde{R}_t$),

*ii)* $\text{VaR}_t$ and $\text{VaR}_{\pi_t}$ converge a.s. to the corresponding optimal subtask value, $z*$, and

*iii)* $Q_t$ converges to a solution of $q$ in the Bellman Equation 4,

all up to an additive constant, $c$.

Hence, to complete the proof, we need to show that $\bar{r}* = \text{CVaR}^*$ and $z* = \text{VaR}^*$:

From Corollary D.4 we know that the optimal long-run average of the extended reward corresponds to the optimal long-run CVaR of the observed reward, hence we can conclude that $\bar{r}* = \text{CVaR}^*$. Finally, from Corollary D.3 we can deduce that since $\text{CVaR}_t$ converges a.s. to $\text{CVaR}^*$, then $z*$ must correspond to $\text{VaR}^*$. This completes the proof. $\square$

As such, with the RED CVaR Q-learning algorithm, we now have a way to optimize the long-run CVaR (and VaR) of the observed reward without the use of an augmented state-space, or an explicit bi-level optimization.

A natural question to ask would be whether we can extend these convergence results to the prediction case. In other words, can we show that a tabular RED CVaR TD-learning algorithm will converge to the long-run VaR and CVaR of the observed reward induced by following a given policy? It turns out that, because we are not optimizing the expectation in Equation D.5e when doing prediction (we are only learning it), we cannot guarantee that we will eventually find the optimal VaR estimate, which implies that we may not recover the CVaR value (since Equation D.5f only holds to the optimal VaR value). However, this is not to say that a RED CVaR TD-learning algorithm has no use. In fact, we do use such an algorithm as part of an actor-critic architecture for optimizing CVaR in the inverted pendulum experiment (see Appendix E). Empirically, as discussed in Appendix E, we find that this actor-critic approach is able to find the optimal CVaR policy.

We end this section by briefly noting that in the risk measure literature, risk measures are typically classified into two categories: *static* or *dynamic*. This classification is based on the *time consistency* of the risk measure that one aims to optimize (Boda and Filar, 2006). Curiously, in our case the CVaR that we aim to optimize does not fit into either category perfectly. One could make the argument that the CVaR that we aim to optimize most closely matches the *static* category, given that there is some time inconsistency before $t \to \infty$. Conversely, one could make a different argument that the CVaR that we aim to optimize most closely resembles the *dynamic* category since the sum over $t$ for the average-reward is outside of the CVaR operator (see Theorem 1 of Xia et al. (2023)), such that an optimal deterministic stationary policy exists (unlike the static case; see Bäuerle and Ott (2011)). This does not affect the significance of our results, but rather suggests that a third category of risk measures may be needed to capture such nuances that occur in the average-reward setting.

## D.2 RED CVAR ALGORITHMS

Below is the pseudocode for the RED CVaR algorithms. Empirically, we found that setting $c_1 = 1.0$ and $c_2 = (1 - \tau)$ yielded good results.

---

**Algorithm 5** RED CVaR Q-Learning (Tabular)

---

    **Input:** the policy $\pi$ to be used (e.g., $\epsilon$-greedy)
    **Algorithm parameters:** step size parameters $\alpha$, $\eta_{\text{CVaR}}$, $\eta_{\text{VaR}}$, CVaR parameter $\tau$, scalars $c_1, c_2$
    Initialize $Q(s, a) \; \forall s, a$ (e.g. to zero)
    Initialize CVaR arbitrarily (e.g. to zero)
    Initialize VaR arbitrarily (e.g. to zero)
    Obtain initial $S$
    **while** still time to train **do**
        $A \leftarrow$ action given by $\pi$ for $S$
        Take action $A$, observe $R, S'$
        $\tilde{R} = \text{VaR} - \frac{1}{\tau} \max\{\text{VaR} - R, 0\} - c_1(\tau - \mathbb{1}_{\{R < \text{VaR}\}}) - c_2\left((1 - \tau) - \mathbb{1}_{\{R \geq \text{VaR}\}}\right)$
        $\delta = \tilde{R} - \text{CVaR} + \max_a Q(S', a) - Q(S, A)$
        **if** $R \geq \text{VaR}$ **then**
            $\text{VaR} = \text{VaR} - \eta_{\text{VaR}}\alpha\delta$
        **else**
            $\text{VaR} = \text{VaR} + \eta_{\text{VaR}}\alpha(\frac{\tau}{1-\tau})\delta$
        **end if**
        $\text{CVaR} = \text{CVaR} + \eta_{\text{CVaR}}\alpha\delta$
        $Q(S, A) = Q(S, A) + \alpha\delta$
        $S = S'$
    **end while**
    return $Q$

---

**Algorithm 6** RED CVaR Actor-Critic

---

    **Input:** a differentiable state-value function parameterization $\hat{v}(s, \boldsymbol{w})$; a differentiable policy parameterization $\pi(a \mid s, \boldsymbol{\theta})$
    **Algorithm parameters:** step size parameters $\alpha$, $\eta_\pi$, $\eta_{\text{CVaR}}$, $\eta_{\text{VaR}}$, CVaR parameter $\tau$, scalars $c_1, c_2$
    state-value weights $\boldsymbol{w} \in \mathbb{R}^d$ and policy weights $\boldsymbol{\theta} \in \mathbb{R}^{d'}$ (e.g. to $\boldsymbol{0}$)
    Initialize CVaR arbitrarily (e.g. to zero)
    Initialize VaR arbitrarily (e.g. to zero)
    Obtain initial $S$
    **while** still time to train **do**
        $A \sim \pi(\cdot \mid S, \boldsymbol{\theta})$
        Take action $A$, observe $R, S'$
        $\tilde{R} = \text{VaR} - \frac{1}{\tau} \max\{\text{VaR} - R, 0\} - c_1(\tau - \mathbb{1}_{\{R < \text{VaR}\}}) - c_2\left((1 - \tau) - \mathbb{1}_{\{R \geq \text{VaR}\}}\right)$
        $\delta = \tilde{R} - \text{CVaR} + \hat{v}(S', \boldsymbol{w}) - \hat{v}(S, \boldsymbol{w})$
        **if** $R \geq \text{VaR}$ **then**
            $\text{VaR} = \text{VaR} - \eta_{\text{VaR}}\alpha\delta$
        **else**
            $\text{VaR} = \text{VaR} + \eta_{\text{VaR}}\alpha(\frac{\tau}{1-\tau})\delta$
        **end if**
        $\text{CVaR} = \text{CVaR} + \eta_{\text{CVaR}}\alpha\delta$
        $\boldsymbol{w} = \boldsymbol{w} + \alpha\delta\nabla\hat{v}(S, \boldsymbol{w})$
        $\boldsymbol{\theta} = \boldsymbol{\theta} + \eta_\pi\alpha\delta\nabla\ln\pi(A \mid S, \boldsymbol{\theta})$
        $S = S'$
    **end while**
    return $\boldsymbol{w}, \boldsymbol{\theta}$

---

## E NUMERICAL EXPERIMENTS

This appendix contains details regarding the numerical experiments performed as part of this work. We discuss the experiments performed in the *red-pill blue-pill* environment (see Appendix F for more details on the red-pill blue-pill environment), as well as the experiments performed in the *inverted pendulum* environment. The aim of the experiments was to contrast and compare the RED RL algorithms (see Appendix D) with the Differential learning algorithms from Wan et al. (2021) in the context of CVaR optimization. In particular, we aimed to show how the RED RL algorithms could be utilized to optimize for CVaR (without the use of an augmented state-space or an explicit bi-level optimization scheme), and contrast the results to those of the Differential learning algorithms, which served as a sort of 'baseline' to illustrate how our risk-aware approach contrasts a risk-neutral approach. In other words, we aimed to show whether our algorithms could successfully enable a learning agent to act in a risk-aware manner instead of the usual risk-neutral manner.

In terms of the algorithms used, Algorithm 5 corresponds to the RED CVaR Q-learning algorithm used in the red-pill blue-pill experiment, and Algorithm 6 corresponds to the RED CVaR Actor-Critic algorithm used in the inverted pendulum experiment. In terms of the Differential learning algorithms used for comparison (see Appendix E.5 for the full algorithms), Algorithm 7 corresponds to the Differential Q-learning algorithm used in the red-pill blue-pill experiment, and Algorithm 8 corresponds to the Differential Actor-Critic algorithm used in the inverted pendulum experiment. For all experiments, we set $c_1 = 1.0$ and $c_2 = (1 - \tau)$ in the RED CVaR algorithms.

### E.1 RED-PILL BLUE-PILL EXPERIMENT

In the first experiment, we consider a two-state environment that we created for the purposes of testing our algorithms. It is called the *red-pill blue-pill* environment (see Appendix E), where at every time step an agent can take either a red pill, which takes them to the 'red world' state, or a blue pill, which takes them to the 'blue world' state. Each state has its own characteristic reward distribution, and in this case, the red world state has a reward distribution with a lower (worse) mean but higher (better) CVaR compared to the blue world state. Hence, we would expect the regular Differential Q-learning algorithm to learn a policy that prefers to stay in the blue world, and that the RED CVaR Q-learning algorithm learns a policy that prefers to stay in the red world. This task is illustrated in Fig. 1a).

For this experiment, we ran both algorithms using various combinations of step sizes for each algorithm. We used an $\epsilon$-greedy policy with a fixed epsilon of $0.1$, and a CVaR parameter, $\tau$, of $0.25$. We set all initial guesses to zero. We ran the algorithms for 100k time steps.

For the Differential Q-learning algorithm, we tested every combination of the value function step size, $\alpha \in \{$2e-1, 2e-2, 2e-3, 2e-4, $1/n\}$ (where $1/n$ refers to a step size sequence that decreases the step size according to the time step, $n$), with the average-reward step size, $\eta\alpha$, where $\eta \in \{$1e-4, 1e-3, 1e-2, 1e-1, 1.0, 2.0$\}$, for a total of 30 unique combinations. Each combination was run 25 times using different random seeds, and the results were averaged across the runs. The resulting (averaged) average-reward over the last 1,000 time steps is displayed in Fig. E.1. As shown in the figure, a value function step size of 2e-4 and an average-reward $\eta$ of 1.0 resulted in the highest average-reward in the final 1,000 time steps in the red-pill blue-pill task. These are the parameters used to generate the results displayed in Fig. 2a).

For the RED CVaR Q-learning algorithm, we tested every combination of the value function step size, $\alpha \in \{$2e-1, 2e-2, 2e-3, 2e-4, $1/n\}$, with the average-reward (in this case CVaR) $\eta \in \{$1e-4, 1e-3, 1e-2, 1e-1, 1.0, 2.0$\}$, and the VaR $\eta \in \{$1e-4, 1e-3, 1e-2, 1e-1, 1.0, 2.0$\}$, for a total of 180 unique combinations. Each combination was run 25 times using different random seeds, and the results were averaged across the runs. The resulting (averaged) reward CVaR over the last 1,000 time steps is displayed in Fig. E.2. As shown in the figure, combinations with larger step sizes converged to the optimal policy within the 100k time steps, and combinations with smaller step sizes did not (see Section E.3 for more discussion on this point). A value function step size of 2e-2, an average-reward (CVaR) $\eta$ of 1e-2, and a VaR $\eta$ of 1e-2 were used to generate the results displayed in Figs. 2a) and 3.

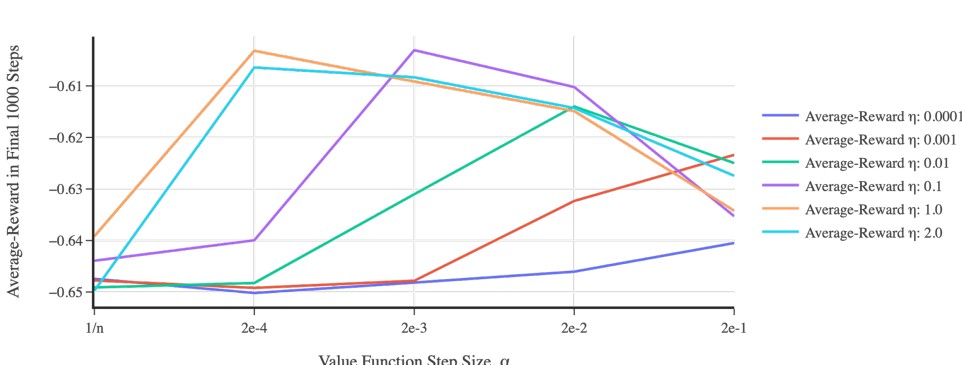

Figure E.1: Step size tuning results for the red-pill blue-pill task when using the Differential Q-learning algorithm. The average-reward in the final 1,000 steps is displayed for various combinations of value function and average-reward step sizes.

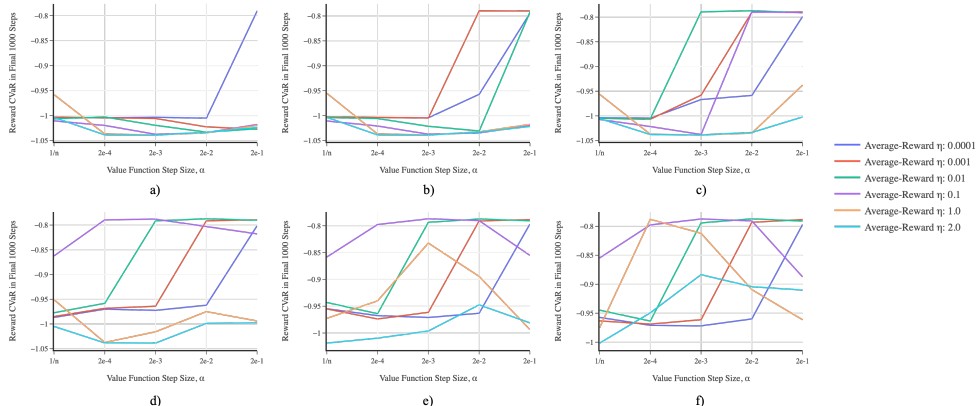

Figure E.2: Step size tuning results for the red-pill blue-pill task when using the RED CVaR Q-learning algorithm. Each plot represents a different $\eta_{VaR}$ used: a) 2e-4; b) 2e-3; c) 2e-2; d) 2e-1; e) 1.0; f) 2.0. Within each plot, the reward CVaR in the final 1,000 steps is displayed for various combinations of value function and average-reward (in this case CVaR) step sizes.

Fig. E.3a) shows the VaR and CVaR estimates as learning progresses when using the RED CVaR Q-learning algorithm with the same step sizes used in Figures 2a) and 3. We see that the resulting VaR and CVaR estimates generally track with what one would expect (similar values, with the VaR value being slightly larger than the CVaR value). We can see however that these estimates do not correspond to the actual VaR and CVaR values induced by the policy (as shown in Fig. 2a)). This is because, as previously mentioned, the solutions to the average-reward MDP Bellman equations (Equations 3, 4), which in this case include the VaR and CVaR estimates, are only correct up to a constant. For comparison, we hard-coded the true VaR value and re-ran the same experiment, and found that the agent still converged to the correct policy, this time with a CVaR estimate that matched the actual CVaR value. Fig. E.3b) shows the results of this hard-coded VaR run.

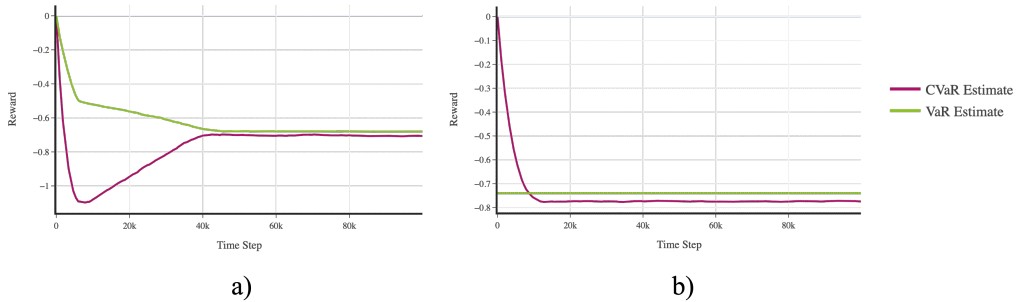

a)                            b)

Figure E.3: The VaR and CVaR estimates as learning progresses when using the RED CVaR Q-learning algorithm: a) as per usual, and b) when hard-coding the VaR estimate to the true VaR.

**Follow-up Experiment: Varying the CVaR Parameter**

Given the results shown in Fig. 2a), we can see that, with proper hyperparameter tuning, the tabular RED CVaR Q-learning algorithm is able to reliably find the optimal CVaR policy for a CVaR parameter, $\tau$, of 0.25. In the context of the red-pill blue-pill environment, this means that the agent learns to stay in the red world state because the state has a characteristic reward distribution with a better (higher) CVaR compared to the blue world state. By contrast, the risk-neutral differential algorithm yields an average-reward optimal policy that dictates that the agent should stay in the blue world state because the state has a better (higher) average reward compared to the red world state.

Now consider what would happen if we used the RED CVaR Q-learning algorithm with a $\tau$ of 0.99. By definition, a CVaR corresponding to a $\tau \approx 1.0$ is equivalent to the average reward. Hence, with a $\tau$ of 0.99, we would expect that the optimal CVaR policy corresponds to staying in the blue world state (since it has the better average reward). This means that for some $\tau$ between 0.25 and 0.99, there is a critical point where the CVaR-optimal policy changes from staying in the red world (let us call this the *red policy*) to staying in the blue world state (let us call this the *blue policy*).

We can estimate this critical point using simple Monte Carlo (MC). We are able to use MC in this case because both policies effectively stay in a single state (the red or blue world state), such that the CVaR of the policies can be estimated by sampling the characteristic reward distribution of each state, while accounting for the exploration $\epsilon$. Fig. E.4 shows the MC estimate of the CVaR of the red and blue policies for a range of CVaR parameters, assuming an exploration $\epsilon$ of 0.1. Note that we used the same distribution parameters listed in Appendix F for the red-pill blue-pill environment. As shown in Fig. E.4, this critical point occurs somewhere around $\tau \approx 0.8$.

Hence, one way that we can further validate the tabular RED CVaR Q-learning algorithm, is by re-running the red-pill blue-pill experiment for different CVaR parameters, and seeing if the optimal CVaR policy indeed changes at a $\tau \approx 0.8$. Importantly, this allows us to empirically validate whether the algorithm actually optimizes at the desired risk level. When running this experiment, we used the same hyperparameters used to generate the results in Fig. 2a), with the exception of using a VaR $\eta$ of 1e-1, as this showed slightly better performance for a broader range of CVaR parameters. We ran the experiment for $\tau \in \{0.1, 0.25, 0.5, 0.75, 0.85, 0.9\}$. For each $\tau$, we performed 10 runs using different random seeds, and the results were averaged across the runs.

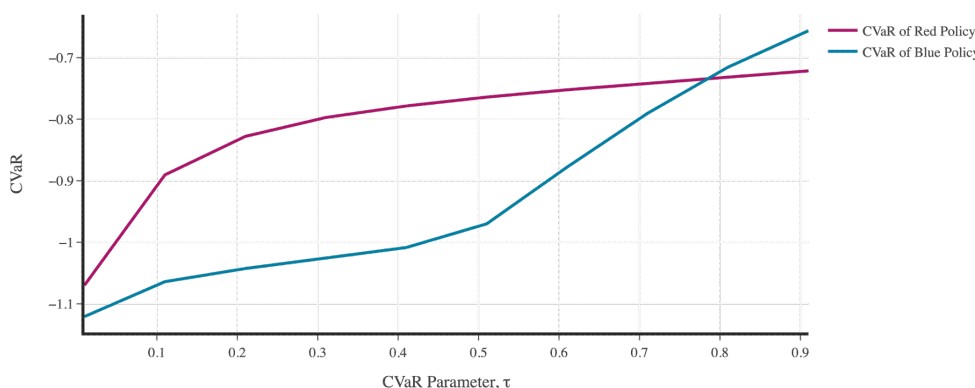

Figure E.4: Monte Carlo estimates of the CVaR of the red and blue policies for a range of CVaR parameters in the red-pill blue-pill environment.

Fig. E.5 shows the results of this experiment. In particular, the figure shows a rolling percent of time that the agent stays in the blue world state as learning progresses (note that we used an exploration $\epsilon$ of 0.1). From the figure, we can see that for $\tau \in \{0.1, 0.25, 0.5, 0.75\}$, the agent learns to stay in the red world state, and for $\tau \in \{0.85, 0.9\}$, the agent learns to stay in the blue world state. This is consistent with what we would expect, given that the critical point is $\tau \approx 0.8$. Hence, these results further validate that our algorithm is able to optimize at the desired risk level. We end by noting that for simplicity, we used the same step sizes across the various $\tau$'s, however, with more robust and $\tau$-specific hyperparameter tuning, more stable results can be obtained, especially for the experiments corresponding to $\tau \in \{0.85, 0.9\}$.

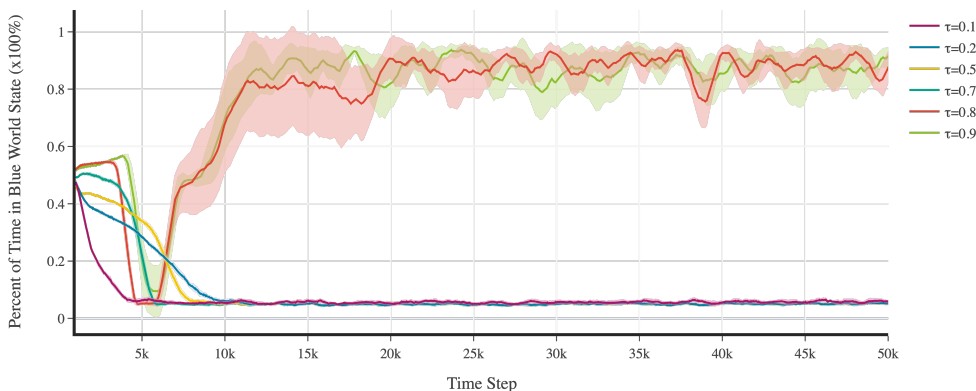

Figure E.5: Rolling percent of time that the agent stays in the blue world state as learning progresses when using the RED CVaR Q-learning algorithm in the red-pill blue-pill tasks for a range of CVaR parameters. A solid line denotes the mean percent of time spent in the blue world state, and the corresponding shaded region denotes the 95% confidence interval over 10 runs.

## E.2 INVERTED PENDULUM EXPERIMENT

In the second experiment, we consider the well-known *inverted pendulum* task, where an agent learns how to optimally balance an inverted pendulum. We chose this task because it provides us with opportunity to test our algorithm in an environment where: 1) we must use function approximation (given the large state and action spaces), and 2) where the policy for the optimal average-reward and the policy for the optimal reward CVaR is the same policy (i.e., the policy that best balances the pendulum will yield a limiting reward distribution with both the optimal average-reward and reward CVaR). This hence allows us to directly compare the performance of our RED algorithms to the regular Differential learning algorithms, as well as to gauge how function approximation affects the performance of our algorithms. For this task, we utilized a simple actor-critic architecture (Barto et al., 1983; Sutton and Barto, 2018) as this allowed us to compare the performance of the (non-tabular) RED TD-learning algorithm with a (non-tabular) Differential TD-learning algorithm. This task is illustrated in Fig. 1b).

For this experiment, we ran both algorithms using various combinations of step sizes for each algorithm. We used a fixed CVaR parameter, $\tau$, of $0.1$. We set all initial guesses to zero. We ran the algorithms for 100k time steps. For simplicity, we used tile coding (Sutton and Barto, 2018) for both the value function and policy parameterizations, where we parameterized a softmax policy. For each parameterization, we used 32 tilings, each with 8 X 8 tiles. By using a linear function approximator (i.e., tile coding), the gradients for the value function and policy parameterizations can be simplified as follows:

$$\nabla \hat{v}(s, \boldsymbol{w}) = \boldsymbol{x}(s), \tag{E.1}$$

$$\nabla \ln \pi(a \mid s, \boldsymbol{\theta}) = \boldsymbol{x}_h(s, a) - \sum_{\xi \in \mathcal{A}} \pi(\xi \mid s, \boldsymbol{\theta}) \boldsymbol{x}_h(s, \xi), \tag{E.2}$$

where $s \in \mathcal{S}$, $a \in \mathcal{A}$, $\boldsymbol{x}(s)$ is the state feature vector, and $\boldsymbol{x}_h(s, a)$ is the softmax preference vector.

For the RED CVaR Actor-Critic algorithm, we tested every combination of the value function step size, $\alpha \in \{$2e-2, 2e-3, 2e-4, $1/n\}$ (where $1/n$ refers to a step size sequence that decreases the step size according to the time step, $n$), with $\eta$'s for the average-reward, VaR, and policy step sizes, $\eta\alpha$, where $\eta \in \{$1e-3, 1e-2, 1e-1, 1.0, 2.0$\}$, for a total of 500 unique combinations. Each combination was run 10 times using different random seeds, and the results were averaged across the runs. The resulting (averaged) reward CVaR over the last 1,000 time steps is displayed in Fig. E.6a) and the resulting (averaged) average-reward over the last 1,000 time steps is displayed in Fig. E.6b). As shown in the figure, most combinations allow the algorithm to converge to the optimal policy that balances the pendulum (as indicated by a reward CVaR and average-reward of zero). A value function step size of 2e-3, a policy $\eta$ of 1.0, an average-reward (CVaR) $\eta$ of 1e-2, and a VaR $\eta$ of 1e-2 were used to generate the results displayed in Fig. 2b).

For the Differential Actor-Critic algorithm, we tested every combination of the value function step size, $\alpha \in \{$2e-2, 2e-3, 2e-4, $1/n\}$, with $\eta$'s for the average-reward and policy step sizes, $\eta\alpha$, where $\eta \in \{$1e-3, 1e-2, 1e-1, 1.0, 2.0$\}$, for a total of 100 unique combinations. Each combination was run 10 times using different random seeds, and the results were averaged across the runs. The resulting (averaged) reward CVaR over the last 1,000 time steps is displayed in Fig. E.6c) and the resulting (averaged) average-reward over the last 1,000 time steps is displayed in Fig. E.6d). As shown in the figure, most combinations allow the algorithm to converge to the optimal policy that balances the pendulum. A value function step size of 2e-3, a policy $\eta$ of 1.0, and an average-reward $\eta$ of 1e-3 were used to generate the results displayed in Fig. 2b).

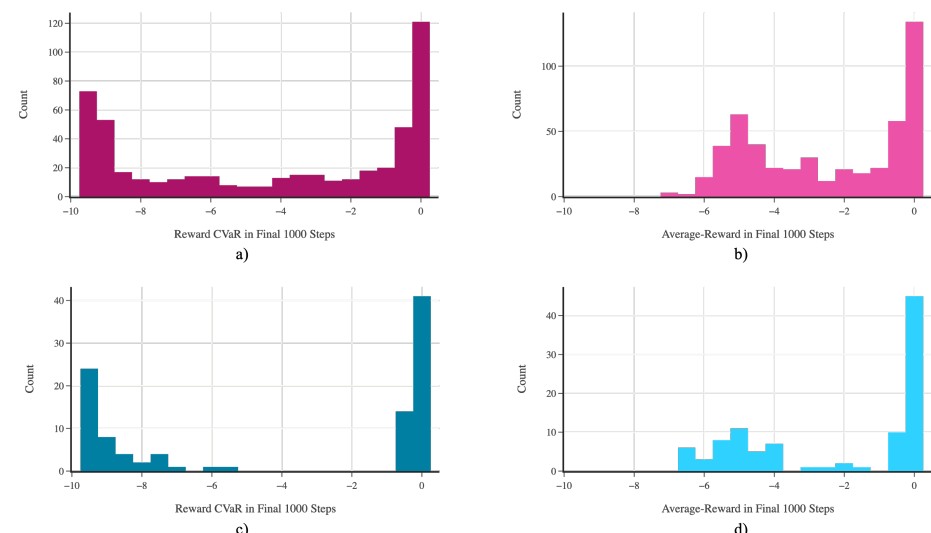

Figure E.6: Step size tuning results for the inverted pendulum task when using the RED CVaR TD-learning and Differential TD-learning algorithms (through an actor-critic architecture). Each plot shows a histogram of either the reward CVaR or average-reward in the last 1,000 steps. More specifically, the histograms show: a) the reward CVaR when using the RED algorithm; b) the average-reward when using the RED algorithm; c) the reward CVaR when using the Differential algorithm; d) the average-reward when using the Differential algorithm.

Fig. E.7a) shows the VaR and CVaR estimates as learning progresses when using the RED CVaR Actor-Critic algorithm with the same step sizes used in Fig. 2b). We see that the resulting VaR and CVaR estimates generally track with what one would expect (similar values, with the VaR value being slightly larger than the CVaR value). We can see however that these estimates do not correspond to the actual VaR and CVaR values induced by the policy (as shown in Fig. 2b)). This is because, as previously mentioned, the solutions to the average-reward MDP Bellman equations (Equations 3, 4), which in this case include the VaR and CVaR estimates, are only correct up to a constant. For comparison, we hard-coded the true VaR value and re-ran the same experiment, and found that the agent still converged to the correct policy, this time with a CVaR estimate that more closely matched the actual CVaR value (note that in the inverted pendulum environment, rewards are capped at zero). Fig. E.7b) shows the results of this hard-coded VaR run.

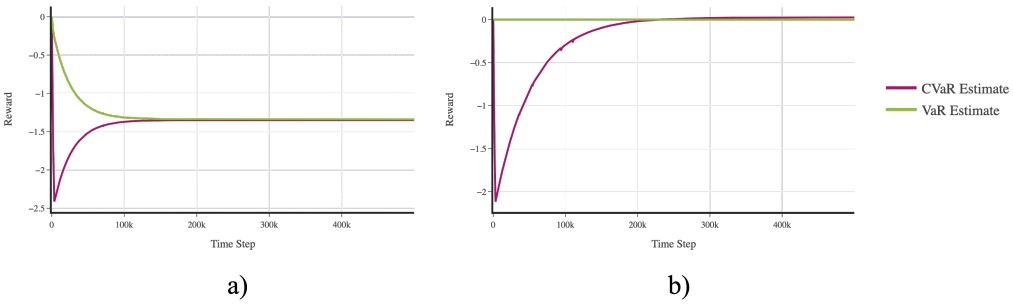

Figure E.7: The VaR and CVaR estimates as learning progresses when using the RED CVaR TD-learning algorithm (through an actor-critic architecture): a) as per usual, and b) when hard-coding the VaR estimate to the true VaR value. Note that in the inverted pendulum environment, rewards are capped at zero.

### E.3 Additional Commentary on Experimental Results

**Red-Pill Blue-Pill**: In the red-pill blue-pill task, we can see from Fig. E.2 that for combinations with large step sizes, the RED CVaR Q-learning algorithm was able to successfully learn a policy, within the 100k time steps, that prioritizes maximizing the reward CVaR over the average-reward, thereby achieving a sort of risk-awareness. However, for combinations with smaller step sizes, particularly for low VaR $\eta$'s, the algorithm did not converge in the allotted training period. We re-ran some of the combinations with constant step sizes for longer training periods, and found that the algorithm eventually converged to the risk-aware policy given enough training time. For combinations with the $1/n$ step size, we found that if the other step sizes in the combination were sufficiently small, the algorithm would not converge to the correct policy (even with more training time). This suggests that a more slowly-decreasing step size sequence should be used instead so that the algorithm has more time to find the correct policy before the step sizes in the sequence become too small.

**Inverted Pendulum**: In the inverted pendulum task, we can see from Fig. E.6 that both algorithms achieved similar performance, as shown by the similar histograms for both the reward CVaR and average-reward during the final 1,000 time steps. These results suggest that both algorithms converged to the same set of (sometimes sub-optimal) policies, as expected.

**Overall**: In both experiments, we can see that with proper hyperparameter tuning, the RED CVaR algorithms were able to consistently and reliably find the optimal CVaR policy. The VaR and CVaR estimates generally tracked with what one would expect (similar values, with the VaR value being slightly larger than the CVaR value). However, these estimates were not always the same as the actual VaR and CVaR values induced by the policy because the solutions to the average-reward MDP Bellman equations are only correct up to a constant. This is typically not a concern, given that the relative ordering of the policies is usually what is of interest.

### E.4 Compute Time and Resources

For the red-pill blue-pill hyperparameter tuning, each case (which encompassed a specific combination of step sizes) took roughly 1 minute (total) to compute all 25 random seed runs for the case on a single CPU, for an approximate total of 4 CPU hours. For the inverted pendulum hyperparameter tuning, each case took roughly 5 minutes (total) to compute all 10 random seed runs for the case on a single CPU, for an approximate total of 50 CPU hours.

## E.5 RISK-NEUTRAL DIFFERENTIAL ALGORITHMS

Below is the pseudocode for the risk-neutral differential algorithms used for comparison in our experiments.

---

**Algorithm 7** Differential Q-Learning (Tabular)

---

**Input:** the policy $\pi$ to be used (e.g., $\epsilon$-greedy)
**Algorithm parameters:** step size parameters $\alpha$, $\eta$
Initialize $Q(s, a)\ \forall s, a$ (e.g. to zero)
Initialize $\bar{R}$ arbitrarily (e.g. to zero)
Obtain initial $S$
**while** still time to train **do**
    $A \leftarrow$ action given by $\pi$ for $S$
    Take action $A$, observe $R, S'$
    $\delta = R - \bar{R} + \max_a Q(S', a) - Q(S, A)$
    $\bar{R} = \bar{R} + \eta\alpha\delta$
    $Q(S, A) = Q(S, A) + \alpha\delta$
    $S = S'$
**end while**
return $Q$

---

**Algorithm 8** Differential Actor-Critic

---

**Input:** a differentiable state-value function parameterization $\hat{v}(s, \boldsymbol{w})$; a differentiable policy parameterization $\pi(a \mid s, \boldsymbol{\theta})$
**Algorithm parameters:** step size parameters $\alpha$, $\eta_\pi$, $\eta_{\bar{R}}$
state-value weights $\boldsymbol{w} \in \mathbb{R}^d$ and policy weights $\boldsymbol{\theta} \in \mathbb{R}^{d'}$ (e.g. to $\boldsymbol{0}$)
Initialize $\bar{R}$ arbitrarily (e.g. to zero)
Obtain initial $S$
**while** still time to train **do**
    $A \sim \pi(\cdot \mid S, \boldsymbol{\theta})$
    Take action $A$, observe $R, S'$
    $\delta = R - \bar{R} + \hat{v}(S', \boldsymbol{w}) - \hat{v}(S, \boldsymbol{w})$
    $\bar{R} = \bar{R} + \eta_{\bar{R}}\alpha\delta$
    $\boldsymbol{w} = \boldsymbol{w} + \alpha\delta\nabla\hat{v}(S, \boldsymbol{w})$
    $\boldsymbol{\theta} = \boldsymbol{\theta} + \eta_\pi\alpha\delta\nabla\ln\pi(A \mid S, \boldsymbol{\theta})$
    $S = S'$
**end while**
return $\boldsymbol{w}, \boldsymbol{\theta}$

---

## F    RED-PILL BLUE-PILL ENVIRONMENT

This appendix contains the code for the *red-pill blue-pill* environment introduced in this work. The environment consists of a two-state MDP, where at every time step an agent can take either a red pill, which takes them to the 'red world' state, or a blue pill, which takes them to the 'blue world' state. Each state has its own characteristic reward distribution, and in this case, the red world state has a reward distribution with a lower (worse) mean but higher (better) CVaR compared to the blue world state. More specifically, the red world state reward distribution is characterized as a gaussian distribution with a mean of $-0.7$ and a standard deviation of $0.05$. The blue world state is characterized by a mixture of two gaussian distributions with means of $-1.0$ and $-0.2$, and standard deviations of $0.05$. We assume all rewards are non-positive.

The Python code for the environment is provided below:

```python
import pandas as pd
import numpy as np

class EnvironmentRedPillBluePill:
  def __init__(self, dist_2_mix_coefficient=0.5):
    # set distribution parameters
    self.dist_1 = {'mean': -0.7, 'stdev': 0.05}
    self.dist_2a = {'mean': -1.0, 'stdev': 0.05}
    self.dist_2b = {'mean': -0.2, 'stdev': 0.05}
    self.dist_2_mix_coefficient = dist_2_mix_coefficient

    # start state
    self.start_state = np.random.choice(
      ['redworld',
       'blueworld']
    )

  def env_start(self, start_state=None):
    # return initial state
    if pd.isnull(start_state):
      return self.start_state
    else:
      return start_state

  def env_step(self, state, action, terminal=False):
    if action == 'red_pill':
      next_state = 'redworld'
    elif action == 'blue_pill':
      next_state = 'blueworld'

    if state == 'redworld':
      reward = np.random.normal(loc=self.dist_1['mean'],
                                scale=self.dist_1['stdev'])
    elif state == 'blueworld':
      dist = np.random.choice(['dist2a', 'dist2b'],
                              p=[self.dist_2_mix_coefficient,
                                 1 - self.dist_2_mix_coefficient])
      if dist == 'dist2a':
        reward = np.random.normal(loc=self.dist_2a['mean'],
                                  scale=self.dist_2a['stdev'])
      elif dist == 'dist2b':
        reward = np.random.normal(loc=self.dist_2b['mean'],
                                  scale=self.dist_2b['stdev'])

    return min(0, reward), next_state, terminal
```

