# OpenReview forum: "Burning RED: Unlocking Subtask-Driven Reinforcement Learning and Risk-Awareness in Average-Reward Markov Decision Processes"
_ICLR.cc/2025/Conference — Submitted to ICLR 2025_

### Official Review · Reviewer_FEZJ · 2024-10-26

**Soundness:** 1
**Presentation:** 2
**Contribution:** 3
**Rating:** 3
**Confidence:** 3

**Summary:**

This paper proposes a way to optimizes the conditional value-at-risk (CVaR) risk measure of the average-reward rate in finite MDPs. The CVaR of a random variable $X$ with a parameter $\tau in (0, 1)$ is the expectation of the lower $\tau$ quantile of X. The key idea is to utilize a property of CVaR by Rockafellar and Uryasev (2000) --- the CVaR of X with a parameter $\tau$ is the expectation of a piece-wise linear function of X and the value-at-risk (VaR; the lower $\tau$ quantile of X). By estimating VaR separately and treating the output of this piece-wise function as a new reward, the paper proposes that CVaR can be estimated using an existing average-reward algorithm. The key advantage of this approach to estimate the CVaR of the reward rate is that it does not perform the bi-level optimization and does not augment the state space, whereas existing algorithms need to do one of them.

**Strengths:**

The claimed contribution has sufficient novelty. However, I am not an expert in this area so I can not confirm if the claim is true.

**Weaknesses:**

I have three concerns about this paper.

First, the writing of this paper is vague, making it hard to understand. For example, it is not clear why subtasks are introduced when the main goal is the CVaR problem, until Section 5. Even in Section 5, the authors didn't explain explicitly how these two ideas are related and how equation (19) was derived. Another example is the discussion about the literature. The paper only mentioned one work for estimating CVaR (Xia et al. 2023) in the average-reward setting. Is that the only work? In addition, was the paper's idea applied to other settings (discounted, episodic) before? If so, what are the differences?

Yet, the major problem is that the derivation of the results in the paper seems to be problematic. Specifically, the step from 13a to 13b does not hold in general for piece-wise linear function f (the proof says that f is linear but Definition 4.1 says that f can be piece-wise linear and in order to be applicable to CVaR, f needs to be piece-wise linear). Similarly, 14a to 14b does not seem to hold.

Third, there are quite a few typos/incorrectness/weird statements of this paper. I list some of them here:
"Average-reward (or average-cost) MDPs were first studied in works such as Puterman (1994)." Puterman's book summarizes previous works. I don't think it's fair to say that these MDPs were "first studied in works such as Puterman (1994)".
At the beginning of Section 3.1, discreet -> discrete, S -> \mathcal{S}, A -> \mathcal{A}.
Equation 1 depends on the start state S_0 while the l.h.s. shows that it is not.
"Such assumptions ensure that, for the induced Markov chain, ...". \mu_\pi here is the limiting distribution, instead of a stationary distribution. In addition, the limiting distribution does not exist for periodic Markov chains.
Max in equation (10) should be Sup. So does several other places in the paper.
Definition 4.1 (ii) should be a property on the function f, instead of a property on the z_i, because z_i is just a scalar input of f, not a random variable.
Lower case letters r, s, s' are sometimes used as random variables and sometimes used as scalars.

**Questions:**

See weaknesses.

---

> ### Author Response · Authors · 2024-11-18
>
> We thank the reviewer for their consideration and review of our paper. Please see our response below:
>
> **It is not clear why subtasks are introduced when the main goal is the CVaR problem, until Section 5. Even in Section 5, the authors didn't explain explicitly how these two ideas are related and how equation (19) was derived.**
>
> We appreciate the reviewer’s feedback. We have added a section (Section 4 in the updated draft) that explains the challenges of CVaR optimization, as well as how the subtask approach can be used to mitigate these challenges.
>
> For clarity, the aim of the paper is to present a framework for solving subtasks simultaneously, with CVaR being an important case study that successfully utilizes this framework. We note that the fundamental approach presented (Theorems 5.1-5.3; previously 4.1-4.3) is not CVaR-specific.
>
> We note that Equation 19 (now equation 17 in the updated draft) is derived in Appendix D of the updated draft.
>
> **The paper only mentioned one work for estimating CVaR (Xia et al. 2023) in the average-reward setting. Is that the only work?**
>
> Yes, to the best of our knowledge, Xia et al. (2023) is the only other work that has looked at optimizing CVaR in the average-reward setting. We note that Xia et al.’s foundational work on the subject is more of a direct adaptation of CVaR optimization methods from the discounted case. By contrast, our work proposes a fundamentally-different approach that does not require the augmented state-space or explicit bilevel optimization that is used by Xia et al.
>
> **Was the paper's idea applied to other settings (discounted, episodic) before? If so, what are the differences?**
>
> There is a lot to unpack here, but we will keep it brief. To answer the reviewer’s question: no, to the best of our knowledge, the paper’s idea has not been applied to episodic and discounted settings. The reason is that our paper’s idea critically relies on the stochastic approximation theory that the average-reward MDP is built upon. By contrast, episodic and discounted MDPs rely on the more typical contraction mapping theory. Hence, applying our idea in the discounted/episodic case would require careful consideration of the differences between the theoretical underpinnings of the various methods.
>
> **The step from 13a to 13b does not hold in general for piece-wise linear function f. Similarly, 14a to 14b does not seem to hold.**
>
> We thank the reviewer for pointing this out, as we should have been more explicit with our explanation. The proof for Theorem 4.1 (now Theorem 5.1) only shows the case for a linear function (we have clarified this in the updated draft), however the results can trivially be extended to the piecewise-linear case, by considering each piecewise segment individually. This is what is done with CVaR, where the resulting update (i.e., Equation 19 in the old draft; Equation 17 in the updated draft) is also piecewise. We have updated the proof of Theorem 4.1 (now Theorem 5.1) to mention this.
>
> Note that we have simplified the proof for Theorem 4.1 (now Theorem 5.1) based on another reviewer’s comments, but the same logic still applies for the updated proof.
>
> **There are quite a few typos/incorrectness/weird statements of this paper.**
>
> We thank the reviewer for identifying these typos and minor errors. The updated draft has corrected these typos/errors, as well as other typos identified by us after the submission deadline. Most notably, we have updated our wording from ‘stationary’ to ‘limiting’ where appropriate, as well as tweaked Definition 4.1 (now Definition 5.1), as per your recommendation.
>
> **Overall**:
>
> We hope that we have addressed the reviewer’s concerns regarding the piecewise-linear function. In the updated draft, we have fixed the typos/errors mentioned by the reviewer. We are happy to engage with the reviewer to provide additional clarifications.

---

> > ### Author Response · Authors · 2024-12-02
> >
> > Dear Reviewer FEZJ,
> >
> > We hope that our updated draft and response to your comments have addressed your concerns and provided the necessary clarifications.
> >
> > We are happy to further engage with you to address any remaining concerns and/or answer any more questions.
> >
> > Best regards,
> >
> > Authors

---

> ### Comment · Reviewer_FEZJ · 2024-12-03
> **Follow-up questions**
>
> I am still confused about the meaning of subtasks. According to your definition in Def 5.1, a subtask is a constant value and a subtask function weighted sum up the reward and all subtasks. Then what does it mean when you say "An average-reward MDP can simultaneously predict or control any arbitrary number of subtasks ". Also, are all the zs given? Could you write down clearly what the agent observes and what the underlying process is (like in the standard MDP setting, the agent observes a state and reward according to the transition probability of the MDP)?

---

> > ### Author Response · Authors · 2024-12-03
> >
> > We thank the reviewer for their most recent comments. We are happy to provide the requested clarifications. Due to the character limit, we split our response across two comments:
> >
> > **(1 / 2)**
> >
> > **I am still confused about the meaning of subtasks. According to your definition in Def 5.1, a subtask is a constant value and a subtask function is a weighted sum of the reward and all subtasks.**
> >
> > Your interpretation is correct. A subtask, $z_i$, is indeed a constant. Definition 5.1 states that in order to satisfy the definition of a “subtask” (in the context of our work), the constant $z_i$ must belong to a suitable function (a “subtask function”) that satisfies the criteria listed in Definition 5.1. In essence, the subtask function must be of the form:
> >
> > $$\tilde{R}_t = R_t + a_0 + a_1z_1 + a_2z_2 + \ldots + a_iz_i + \ldots + a_nz_n.$$
> >
> > (Where we have $n$ subtasks).
> >
> > A natural question to ask might be: *Why must the subtask function be of this form?* Well, as we will discuss below, a subtask function of this form makes it possible to estimate and/or optimize any given subtask using only a modified version of the TD error.
> >
> > **Aren’t all the zs given?**
> >
> > Although the subtasks (**zs**) are constant, they are not given, and so we need a way to estimate them. To this end, the primary contribution of our paper is a general-purpose framework that allows us to estimate and/or optimize the subtasks using only a modified version of the TD error. In other words, we start with an initial guess for a given subtask $z_i$ (i.e., $Z_{i, t=0}$), then, through our framework, we can estimate/optimize this estimate using only a modified version of the TD error (this happens in parallel to the regular value function learning/optimizing that occurs in the MDP). We note that Appendix C includes convergence proofs for the subtask estimates for the tabular case (i.e. $Z_{i, t} \to z_i$ as $t \to \infty$). Also see Theorems 5.2, 5.3, and D.1.2.
> >
> > A natural follow-up question to ask might be: *What is the point of estimating or optimizing a subtask?* In our work, we assume that there is some underlying motivation or benefit to estimating and/or optimizing the subtask $z_i$. For example, we might want to know what the constant $z_i$ is for a given policy (hence, we want to ‘estimate’ the subtask), or, we may want to know what the constant $z_i$ is for the optimal policy (hence, we want to ‘optimize’ the subtask). In the case of CVaR, we want to know what the value-at-risk (VaR) of the optimal policy is, because if we know what this (constant) value is, then we can turn the computationally-expensive process of optimizing CVaR (which can involve solving multiple MDPs) into a trivial one (that only requires solving a single MDP). See lines 210-240 for a thorough discussion on this. Importantly, our framework is able to simultaneously estimate/optimize any arbitrary number of subtasks simultaneously in a fully-online manner.
> >
> > We note that the ‘weights’ $a_0, …, a_n$ are given (they are problem-specific). For example, in the CVaR case study we have single subtask, VaR, with the following ‘weight’:
> >
> > $a_{\text{VaR}, t} = 1$ if $R_t >= \text{VaR}_t$,  and
> >
> > $a_{\text{VaR}, t} = \frac{\tau - 1}{\tau}$ if $R_t < \text{VaR}_t$ (where $\tau$ is the known CVaR parameter).
> >
> > We note that a full derivation of these ‘weights’ for CVaR can be found in Appendix D.
> >
> > **...(Continued in next comment)**

---

> ### Author Response · Authors · 2024-12-03
>
> **(2 / 2)**
>
> **What does it mean when you say "An average-reward MDP can simultaneously predict or control any arbitrary number of subtasks "**
>
> This means that in the average-reward setting, we can develop a learning/update rule for a given subtask (that belongs to a subtask function that satisfies the criteria listed in Defn 5.1) based solely on a modified version of the TD error:
>
> $$Z_{i, t+1} = Z_{i, t} + \eta\alpha_{t}(-1/a_i)\delta_t$$
>
> Where, $Z_{i, t}$ is the estimate of the subtask, $z_i$, at time $t$, $\eta\alpha_{t}$ is the step size, and $\delta_t$ is the TD error.
>
> Importantly, the $(-1/a_i)\delta_t$ term satisfies a TD error-dependent property: it goes to zero as the TD error, $\delta_t$, goes to zero. This implies that the arbitrary subtask update is dependent on the TD error, such that the subtask estimate will only cease to update once the TD error is zero. Hence, minimizing the TD error allows us to solve (i.e., estimate or optimize) the arbitrary subtask, $z_i$, simultaneously using the TD error.
>
> See Theorem 5.1 for a more thorough discussion.
>
> **Could you write down clearly what the agent observes and what the underlying process is (like in the standard MDP setting, the agent observes a state and reward according to the transition probability of the MDP)?**
>
> Let $\mathcal{M} \doteq \langle \mathcal{S}, \mathcal{A}, \mathcal{R}, p \rangle$ denote the standard (average-reward) MDP, where $\mathcal{S}$ is a finite set of states, $\mathcal{A}$ is a finite set of actions, $\mathcal{R} \subset \mathbb{R}$ is a finite set of rewards, and $p: \mathcal{S}\, \times\, \mathcal{A}\, \times\, \mathcal{R}\, \times\,  \mathcal{S} \rightarrow{} [0, 1]$ is a probabilistic transition function that describes the dynamics of the environment. At each discrete time step, $t = 0, 1, 2, \ldots$, the agent chooses an action, $A_t \in \mathcal{A}$, based on its current state, $S_t \in \mathcal{S}$, and receives a reward, $R_{t+1} \in \mathcal{R}$, while transitioning to a (potentially) new state, $S_{t+1}$, such that $p(s', r \mid s, a) = \mathbb{P}(S_{t+1} = s', R_{t+1} = r \mid S_t = s, A_t = a)$.
>
> In our proposed framework, we only modify the reward via the subtask function, such that:
>
> $$\tilde{R}_t = R_t + a_0 + a_1z_1 + a_2z_2 + \ldots + a_iz_i + \ldots + a_nz_n.$$
>
> This results in the modified MDP, $\mathcal{\tilde{M}} \doteq \langle \mathcal{S}, \mathcal{A}, \tilde{\mathcal{R}}, \tilde{p} \rangle$, where the states and actions are the same as in the standard MDP, $\tilde{R_t} \in \tilde{\mathcal{R}}$, and the transition probabilities are essentially identical to that of the standard MDP, such that the probability of obtaining the modified reward, $\tilde{R_t}$, is the same probability as obtaining the corresponding regular reward (from the standard MDP), $R_t$, such that:
>
> $\tilde{p}(s', \tilde{r} \mid s, a) = \mathbb{P}(S_{t+1} = s', \tilde{R}_{t+1} = \tilde{r} \mid S_t = s, A_t = a) \ldots$
>
> $\quad = \mathbb{P}(S_{t+1} = s', R_{t+1} = r \mid S_t = s, A_t = a) = p(s', r \mid s, a)$
>
> (This follows from the subtasks being independent of the states and actions; see Definition 5.1)
>
> Hence, at each time step:
>
> 1) The agent chooses an action, $A_t$, based on the current state, $S_t$.
>
> 2) We observe the reward, $R_{t+1}$, and new state, $S_{t+1}$, from the environment.
>
> 3) We calculate the modified reward, $\tilde{R}_{t+1}$, using the subtask function
>
> (i.e., using $R_{t+1}$ and the subtask estimates $Z_{0,t}, …, Z_{i,t}, …, Z_{n,t}$).
>
> 4) Proceed with the usual learning updates (TD error, value functions, etc.) using the modified reward.
>
> 5) Update the subtask estimates using a modified version of the TD error (see our response to the previous questions).
>
> 6) Set $S_t = S_{t+1}$. Go back to step 1.
>
> We note that full pseudocode is provided in Appendix B (line 702).
>
> As such, this process allows us to estimate/optimize the long-run average of the modified reward, as well as the subtasks. In the case of CVaR, optimizing the long-run average of the modified reward corresponds to optimizing the long-run CVaR of the reward from the standard MDP (which is our actual objective).
>
> We note that through this subtask-driven approach, we have provided the first algorithm in any MDP-based setting to optimize CVaR without the use of an augmented state-space, or an explicit bi-level optimization, thereby reducing significant computational costs. We also note that this subtask-driven approach is not specific to CVaR and can be applied to other learning problems in the future.
>
> We appreciate the reviewer's time and dedication, and hope that we have provided sufficient clarifications.

---

> > ### Comment · Reviewer_FEZJ · 2024-12-03
> >
> > Thanks. So the agent maintains subtask estimates Z_i and wants to estimate z_i. But there must be some signals that are grounded to z_i and these signals are observed by the agent, right? What are these signals?

---

> ### Author Response · Authors · 2024-12-03
>
> We thank the reviewer for their quick response and most recent comments. We are happy to provide the additional clarification.
>
> **The agent maintains subtask estimates Z_i and wants to estimate z_i. But there must be some signals that are grounded to z_i and these signals are observed by the agent, right? What are these signals?**
>
> Consider a subtask function of the form:
>
> $$\tilde{R}_t = R_t + a_0 + a_1z_1 + a_2z_2 + \ldots + a_iz_i + \ldots + a_nz_n$$
>
> where the modified reward, $\tilde{R}_t$, is the output of the subtask function.
>
> As per Theorem 5.1, this yields an update rule for an arbitrary subtask, $z_i$, of the form:
>
> $$Z_{i, t+1} = Z_{i, t} + \eta\alpha_{t}(-1/a_i)\delta_t.$$
>
> As previously mentioned, our framework assumes that the constants (or ‘weights’) $a_0, a_1, ..., a_i, ..., a_n$ are known.
>
> Hence, we can enforce a particular update that is grounded to $z_i$ based on our choice of $a_i$.
>
> For example, in the CVaR case study, we can enforce the desired update for the subtask, VaR, by choosing the following $a_{\text{VaR}}$:
>
> $a_{\text{VaR}, t} = 1$ if $R_t >= \text{VaR}_t$,  and
>
> $a_{\text{VaR}, t} = \frac{\tau - 1}{\tau}$ if $R_t < \text{VaR}_t$.
>
> This choice comes directly from Equation 7 in the update draft. This equation defines CVaR in terms of VaR. In our work, we adapt this equation into the following subtask function:
>
> $$\tilde{R}_t = \text{VaR} - \frac{1}{\tau}(\text{VaR} - R_t)^{+} + \text{other terms}$$
>
> (See Equation D.6 for the full function; the other terms are not relevant to this specific response)
>
> Hence, we get $a_{\text{VaR}}$ by grouping the VaR terms in the subtask function (for both cases: $R_t >= \text{VaR}$ and $R_t < \text{VaR}$).
>
> In other words, we use the definition of CVaR to define the subtask function, thereby yielding an $a_{\text{VaR}}$ that allows us to estimate the desired subtask, VaR.
>
> This highlights the appeal of our approach and the core contribution of our paper: Instead of having to rely on some other gradient (e.g. quantile regression) to estimate our subtask (in this case, VaR), our approach allows us to estimate the subtask in a theoretically-sound way using a modified version of the TD error.
>
> Conversely, it also highlights a limitation: it may not always be straightforward to derive a useful subtask function (and the corresponding $a_0, a_1, ..., a_i, ..., a_n$). However, as evidenced by the significant CVaR result in our paper, given an appropriate subtask function, our framework can be quite useful and powerful.
>
> Hence, to summarize: The ‘signal’ that makes this all possible is the modified reward, $\tilde{R}_t$, which, through our framework, yields the update rules for the subtasks, such that the update for a given subtask $z_i$ is grounded by the choice of $a_i$.
>
> We thank the reviewer for their consideration of our paper. We hope that we have provided an adequate clarification.

---

### Official Review · Reviewer_NkZR · 2024-10-29

**Soundness:** 2
**Presentation:** 2
**Contribution:** 2
**Rating:** 3
**Confidence:** 4

**Summary:**

This paper focuses on the infinite horizon average criterion, to learn CVaR return without bi-level optimization. Indeed, CVaR-MDPs i.e., MDPs under CVaR objective, require solving an optimization problem at each policy evaluation step. By switching to the CVaR of the average reward (instead of discounted), the authors introduce RED-CVaR, a TD-type algorithm that avoids the inner optimization problem. Convergence results are provided under standard assumption. The approach is validated on a two-state MDP and inverted pendulum.

**Strengths:**

- The paper is easy to read, and the writing skills are good.
- I am unaware of previous work that proposed CVaR optimization for the average reward criterion, so this is original (as far as I can tell).

**Weaknesses:**

I reviewed this paper for another venue where reviewers voted for rejection unanimously.
There have not been any substantial updates since that submission, so my concerns apply to this one. I copy below the most critical concerns I had then:

- Even in the risk-neutral case, the average reward criterion has some analytical advantages. Notably, [2] focuses on that same criterion rather than the discounted one. As a side comment, I am unaware of any provably convergent AC algorithms for the risk-neutral discounted return. In that respect, it does not surprise me that the same holds for risk-sensitive MDPs, which questions the significance of this work.
- Another missing related work is [5], which considers infinite horizon average reward but with entropic risk instead of CVaR
A discussion on the nature of the risk considered in this work is missing: is it nested or static? It looks static, i.e., the objective is $\text{CVaR}(\bar{r}_{\pi})$, not nested, see [3, 4]. Therefore, time-consistency issues may arise and if not, they should be discussed. On the other hand, the nested formulation enables doing DP but lacks interpretability.
- The initial claims in Sec 2.1 are incorrect: average criteria have been extensively studied already in the 60-s with Howard and Blackwell. In particular, the Blackwell optimality criterion bridges the gap between discounted and average returns. See Chaps 8-9 of [1].
Eqs (4)-(5) are called Poisson equations, see [2].
- In the risk-neutral case, [2] do function approximation on the average reward setting. I think the same could be done for CVaR + function approximation.
- Def 4.1 is unclear. How does this definition translate to the max in Eq. (10) ?
- The statements of Thms. 4.1 and 4.2 are vague and should be formalized: what do they show? Why not focus on just one subtask as this is the case of CVaR optimization?
- Formal algorithm pseudo-codes should appear instead of a list of equations (17)
- The learning rates $\eta$, $\alpha$ are sometimes constant, sometimes time or even state-dependent.
- The convergence plots seem to show one run per experiment. How is the seed chosen? Is it random? Have the algorithms been run on more seeds? The authors are encouraged to plot error curves with mean±std.


Broadly speaking, the following concerns led to my grading:

- Some claims are inaccurate, including related works.
- The experiments are somewhat unclear and not very convincing.
- Although optimizing a CVaR-average return criterion is new, the theoretical contribution seems to be a simple adaptation of risk-neutral TD to CVaR-TD. In particular, explaining the analytical challenges encountered under risk-sensitive criteria would be helpful. In particular, why is the unichain assumption still necessary? I think this comes from the static nature of the CVaR -- I don't think the same assumption would be required/enough if CVaR here were nested.


*[1] Puterman, Martin L. Markov decision processes: discrete stochastic dynamic programming. John Wiley & Sons, 2014. [2] Bhatnagar, Shalabh, et al. "Incremental natural actor-critic algorithms." Advances in neural information processing systems 20 (2007). [3] Hau, Jia Lin, Marek Petrik, and Mohammad Ghavamzadeh. "Entropic risk optimization in discounted MDPs." International Conference on Artificial Intelligence and Statistics. PMLR, 2023. [4] Shen, Yun, Wilhelm Stannat, and Klaus Obermayer. "Risk-sensitive Markov control processes." SIAM Journal on Control and Optimization 51.5 (2013): 3652-3672. [5] Murthy, Yashaswini, Mehrdad Moharrami, and R. Srikant. "Modified Policy Iteration for Exponential Cost Risk Sensitive MDPs." Learning for Dynamics and Control Conference. PMLR, 2023.*

**Questions:**

I encourage the authors to account for previous reviews' comments and suggestions and update their paper accordingly.

---

> ### Author Response · Authors · 2024-11-18
>
> We thank the reviewer for their consideration and review of our paper. Please see our response below. Due to the character limit, we split our response across three comments:
>
> **(1 / 3)**
>
> **The statements of Thms. 4.1 and 4.2 are vague and should be formalized: what do they show? Why not focus on just one subtask as this is the case of CVaR optimization?**:
>
> Based on the reviewer’s comments, we have updated the proof for Theorem 4.1 (now Theorem 5.1), such that the concept is presented in a clearer manner (it no longer uses the vague function inverse notation). Moreover, the updated draft now includes CVaR-specific proofs of Theorems 4.1 and 4.3 (now Theorems 5.1 and 5.3) in Appendix D (see Theorems D.1.1, D.1.2).
>
> The generic theorems (5.1-5.3) show that we can solve any subtask that meets definition 4.1 (now defn 5.1) using the TD error. This is significant because it means that we do not have to rely on other types of gradients (for example, the gradient of the quantile loss) to solve the subtasks. The CVaR specific theorems (D.1.1, D.1.2) show that, by only using the TD error, our algorithms converge to the optimal long-run VaR and CVaR of the observed reward.
>
> We note that the underlying methods presented in our work are not CVaR-specific. Hence, we present them in a generalized way in the main text, so that we do not give the impression that the methods presented can only be used in the context of CVaR.
>
> **How does definition 4.1 translate to the max in Eq. (10) ?**:
>
> Conceptually, we can think of the subtask function as the expression inside the expectation in Equation 10 (Equation 7 in the updated draft), such that $X$ corresponds to the observed per-step reward, $b$ corresponds to the subtask that we want to solve, and the output of this expression corresponds to a modified per-step reward, $\tilde{r}$, who’s long-run average we can optimize using the average-reward MDP.
>
> The basic idea is that: the max in Equation 10 (Equation 7 in the updated draft) motivates having to optimize the subtask $b$ because if we optimize $b$, as well as the average of the modified reward, then we will have optimized the CVaR of the observed reward (see Corollaries D.1-D.4). As such, the max does not directly correspond to defn 4.1 (now defn 5.1), but motivates our framework for solving subtasks (as defined in defn 5.1).
>
> Now, in practice Equation 10 (Equation 7 in the updated draft) needs to be modified because directly using it as the subtask function may result in multiple solutions (see lines 1383-1450), so in Appendix D.1 we augment it to narrow the set of possible solutions to those such that $b=$VaR.
>
> **Notably, [2] focuses on that same criterion rather than the discounted one.**
>
> We now mention [2] in Section 2.1 of the updated draft.
>
> The key difference between our work and [2] lies in the average-reward (i.e. $\bar{R}_t$) update. In [2], the average-reward update is as follows (step 5 in Table 1 of Bhatnagar, et. al., where we align the notation with that of our draft):
>
> $$\bar{R}_{t+1} = (1 - \alpha) \bar{R}_t + \alpha R_t$$
>
> where, $R_t$ is the observed reward at time t, and $\alpha$ is the step size.
>
> In contrast, the algorithms in our work estimate the average-reward using the TD error, $\delta$ (e.g. Eqn 6d in our draft):
>
> $$\bar{R}_{t+1} = \bar{R}_t + \alpha \delta_t$$
>
> Hence, the analytical results of Bhatnagar, et. al. are not directly applicable.
>
> As a side note, the update used in Bhatnagar, et. al. is restricted to the on-policy case, whereas our update is applicable to both the on-policy and off-policy cases.
>
> **In the risk-neutral case, [2] do function approximation on the average reward setting. I think the same could be done for CVaR + function approximation.**:
>
> We provide algorithms with function approximation in Appendix B and D.2 of the updated draft (where algorithm 6 in Appendix D.2 does function approximation + CVaR). Moreover, we use CVaR algorithms with function approximation in the inverted pendulum experiment. We also note that as previously mentioned, the work that the reviewer mentioned does not use the TD error to estimate the average reward, and hence the results are not directly applicable.
>
> **As a side comment, I am unaware of any provably convergent AC algorithms for the risk-neutral discounted return. In that respect, it does not surprise me that the same holds for risk-sensitive MDPs, which questions the significance of this work.**
>
> We would be grateful if the reviewer could elaborate on this point, as it is not clear to us how the convergence of AC algorithms impacts the significance of our work. We would respectfully argue that the significance of our work lies in the ability to solve various subtasks simultaneously in a fully-online manner, with theoretical guarantees in the tabular case. The CVaR case study is an important result, however our approach is not specific to CVaR, and can be applied beyond the risk-aware domain.
>
> **...(Continued in next comment)**

---

> ### Author Response · Authors · 2024-11-18
>
> **(2 / 3)**
>
> **The experiments are somewhat unclear and not very convincing.**:
>
> While our experiments are limited, they, in combination with the theoretical work and impactful case study presented, address several critical questions and potential concerns about the capabilities of our algorithms. In particular, they show that:
>
> - Our algorithms are able to successfully optimize CVaR even if it results in a lower average-reward.
>
> - Our algorithms have comparable (if not better) performance to the baseline risk-neutral algorithms, as shown in the inverted pendulum experiment where both methods share the same optimal solution.
>
> - Even with function approximation and an actor-critic architecture, the algorithms are still able to find the optimal CVaR policy.
>
> - Our algorithms are robust to the initial guesses for the VaR/CVaR estimates.
>
> We have also included an additional experiment in Appendix E (line 1806) of the update draft, which further validates that our algorithms can optimize at the desired risk level.
>
> As such, we believe that our experiments, in conjunction with the theoretical results, sufficiently demonstrate the capabilities of our algorithms.
>
> **The theoretical contribution seems to be a simple adaptation of risk-neutral TD to CVaR-TD.**:
>
> There are two points that we would like to make.
>
> The first is that the theoretical work presented in our work goes beyond CVaR. In particular, we present a general-purpose framework that makes it possible to solve various learning objectives simultaneously and in a fully-online manner in the average-reward setting using only (a potentially modified version of) the TD error. This includes convergence proofs for tabular algorithms derived from this framework. For clarity, the key theoretical contributions of this work are not CVaR-specific, and can be applied beyond the risk-aware domain.
>
> The second point is that we are able to leverage this general-purpose framework to achieve an important result: optimizing CVaR without augmenting the state-space or needing an explicit bilevel optimization scheme. To our knowledge, our algorithm is the first to achieve this in an MDP-based setting. By contrast, a simple adaptation of risk-neutral TD to CVaR-TD, such as the methods proposed by Xia et al. (2023), would need to have an augmented state-space and a bilevel optimization to optimize CVaR. This can potentially mean having to solve multiple MDPs or a standalone optimization at every step.
>
> **Explaining the analytical challenges encountered under risk-sensitive criteria would be helpful.**:
>
> We have included a discussion of this in the updated draft in Section 4.
>
> The primary non-triviality lies in that we need to know what the optimal VaR is in order to calculate the optimal CVaR. However, one does not typically know this value beforehand, so existing methods have to perform some version of the optimization presented in Equation 11 of our paper (Equation 8 in the updated draft).
>
> In a standard/naive implementation of Equation 11 (Equation 8 in the updated draft), we need to augment the state-space with VaR, which can be any real (potentially-bounded) number. Moreover, a naive implementation often implies solving multiple MDPs (each with a different guess for VaR), which compounds the computational costs induced by a larger state-space.
>
> Now consider more clever methods that attempt to mitigate the computational costs. One of the most well-known, computationally-efficient examples is Chow et al. (2015), who utilized a clever but cumbersome decomposition technique that made it possible to only need to augment the state-space with a value between 0 and 1, as well as only needing to solve a single MDP. However, even this clever method requires the use of linear interpolation, as well as having to solve a standalone optimization at every iteration.
>
> By contrast, the average-reward formulation, in combination with our proposed approach, allows us to circumvent these issues altogether, such that we can optimize both VaR and CVaR simultaneously in a fully-online manner.
>
> **...(Continued in next comment)**

---

> ### Author Response · Authors · 2024-11-18
>
> **(3 / 3)**
>
>  **Why is the unichain assumption still necessary? I think this comes from the static nature of the CVaR -- I don't think the same assumption would be required/enough if CVaR here were nested.**:
>
> As per our previous discussions, the unichain requirement may vary depending on whether we consider a nested or a static risk measure. Here the CVaR that we aim to optimize is the CVaR of the limiting reward distribution, which is a stationary measure.
>
> **The convergence plots seem to show one run per experiment. How is the seed chosen? Is it random? Have the algorithms been run on more seeds? The authors are encouraged to plot error curves with mean±std.**:
>
> We kindly point out that convergence plots show the mean and 95% confidence interval for 50 random seed runs.
>
> **A discussion on the nature of the risk considered in this work is missing**:
>
> We included a discussion on the nature of the risk in Appendix C of the submitted draft (lines 1844-1847). We have expanded upon this section in the updated draft (now Appendix D, lines 1608-1619). We recognize that reviewers are not required to look at appendices, however we felt that this section was best suited to be in the Appendix, where we discuss the CVaR-specific approach in great detail.
>
> **The learning rates are sometimes constant, sometimes time or even state-dependent.** , **Another missing related work is [5], which considers infinite horizon average reward but with entropic risk instead of CVaR**, and **The initial claims in Sec 2.1 are incorrect**:
>
> We have rectified these points in the updated draft.
>
> **Formal algorithm pseudo-codes should appear instead of a list of equations (17)**:
>
> We include full formal algorithm pseudo-codes in Appendix B (and D). We recognize that reviewers are not required to look at appendices, however we felt that the space in the main body was better used for explaining other concepts, than including a detailed algorithm that may not be of immediate interest to the reader.
>
> **Eqs (4)-(5) are called Poisson equations**:
>
> Correct. We note that Eqs (4)-(5) are also commonly referred to as Bellman equations in the RL literature. We have added Poisson in brackets in the updated draft.
>
> **Even in the risk-neutral case, the average reward criterion has some analytical advantages.**
>
> Agreed! We hope that our work will encourage more exploration of methods in the average-reward setting, beyond the risk-aware domain.
>
> **References**:
>
> Xia, Li, Luyao Zhang, and Peter W. Glynn. "Risk‐sensitive Markov decision processes with long‐run CVaR criterion." Production and Operations Management 32.12 (2023): 4049-4067.
>
> Yinlam Chow, Aviv Tamar, Shie Mannor, and Marco Pavone. Risk-sensitive and robust decision making: a CVaR optimization approach. In Advances in neural information processing systems 28, 2015.

---

> > ### Author Response · Authors · 2024-12-02
> >
> > Dear Reviewer NkZR,
> >
> > We hope that our updated draft and response to your comments have addressed your concerns and provided the necessary clarifications.
> >
> > We are happy to further engage with you to address any remaining concerns and/or answer any more questions.
> >
> > Best regards,
> >
> > Authors

---

### Official Review · Reviewer_dBpb · 2024-10-29

**Soundness:** 2
**Presentation:** 2
**Contribution:** 2
**Rating:** 6
**Confidence:** 3

**Summary:**

This paper extends the risk-averse average-reward MDP framework from [1] and introduces a new approach called "Reward Extended Differential (RED)" for solving various subtasks (e.g., scalar prediction or control objectives) concurrently. Instead of using the observed reward $R$ directly, the TD error is defined using a modified reward $\tilde{R} = f(R,Z_1,Z_2 ...,Z_n)$ where $f$ is an invertible function mapping the observed reward and all subtasks to a modified reward $\tilde{R}$. The authors demonstrate their algorithm’s application to risk-averse (CVaR) decision-making in a fully online setting.

References:

[1] Wan, Yi, Abhishek Naik, and Richard S. Sutton. "Learning and planning in average-reward markov decision processes." International Conference on Machine Learning. PMLR, 2021.

**Strengths:**

(a) The abstract, introduction, and preliminaries on average reward reinforcement learning are well-written and clearly presented.

(b) The TD and Q-learning with stochastic approximation algorithms, along with Theorems 4.1–4.3, appear to be rigorously verified with proofs in Appendix B. These proofs effectively extend the results from [1, 2, 3] to the multi-subtasks setting proposed in this work.

References:

[1] Wan, Yi, Abhishek Naik, and Richard S. Sutton. "Learning and planning in average-reward markov decision processes." International Conference on Machine Learning. PMLR, 2021.

[2] Vivek S Borkar. Asynchronous stochastic approximations. SIAM Journal on Control snd Optimization, 36(3):840–851, 1998.

[3] Vivek S Borkar. Stochastic Approximation: A Dynamical Systems Viewpoint. Springer, 2009.

**Weaknesses:**

Despite the authors' in-depth understanding of stochastic approximation and model-free Q-learning proofs, the paper lacks sufficient validation regarding the extension to risk awareness in average reward MDPs.

(a) The paper demonstrates a limited engagement with prior work and foundational concepts in risk-averse CVaR MDPs. The authors inaccurately claim that “our work is the first to propose an MDP-based CVaR optimization algorithm that does not require an explicit bi-level optimization scheme or an augmented state-space.” However, several existing approaches such as dynamic risk-averse MDPs [1], risk-averse distributional RL [2, 3,11] and average-criteria CVaR [9] also avoid state-space augmentation and employ stationary Markov policies, similar to this work. Furthermore, the proposed algorithm still seems to be bilevel as it aim to optimize for CVaR but update the VaR estimate at every level. Moreover the author mentioned that "the CVaR that we aim to optimize most closely matches the static category", restricting to stationary Markov policies can impair both the optimality and interpretability of static CVaR MDPs (see [4, 5, 6, 7]), since the sum over $t \in 1:n$ for average criteria is outside of the CVaR operator, this is closer to the dynamic category where optimal deterministic stationary policy exist (see Theorem 1 of [9]). Additionally, the authors overlook related works [8] applies a similar TD update and [10] consider time-consistent policies set. It should be noted that "notable works such as [6]" describe in related work section, are known to be sub-optimal for policy optimization (see [7]). For this reason, augmented state-space primal methods with bi-level optimization, as in static CVaR MDP algorithms [4, 5, 9], are generally preferred.

(b) The CVaR analysis in Appendix C.1 is focused solely on evaluation, leaving out an analysis for policy optimization claim "We can now optimize the expectation in Equation C.5f using the RED RL framework". Additionally, the average criterion CVaR objective function itself is not explicitly presented in the paper. Sections 4 and 5 feel somewhat disconnected; providing a clearer explanation to link these sections, along with an explicit proof that the proposed algorithm can optimize the CVaR objective, would significantly strengthen the paper’s claims regarding risk-aware reinforcement learning in average reward MDPs.

(c) Limited empirical results: The results in Section 5 do not demonstrate that the proposed algorithm effectively optimizes the desired CVaR risk level. The evaluation would be more convincing if the authors trained the algorithm across multiple distinct CVaR risk levels (e.g. $\tau \in [0.01,0.05,0.1,0.5,1]$), and subsequently assessed performance by calculating the CVaR of the average reward over the final $n$ steps for each risk level $\tau' \in [0.01,0.05,0.1,0.5,1]$. Ideally, the maximum performance at each evaluated risk level should correspond to the training run specifically conducted at that CVaR risk level, reinforcing that the algorithm correctly optimizes for the specified risk. Furthermore, comparing the proposed algorithm’s performance with other approaches [4,9,11] under an average reward criterion could also provide a clearer benchmark for its effectiveness.

(d) The claim to “learn a policy that optimized the CVaR value without using an explicit bi-level optimization scheme or an augmented state-space, thereby alleviating some of the computational challenges” is not substantiated. This claim would be more convincing if the authors compared the computational complexity or running time of the proposed method with that of the algorithm proposed in [9].

References:

[1] Ruszczyński, Andrzej. "Risk-averse dynamic programming for Markov decision processes." Mathematical programming 125 (2010): 235-261.

[2] Keramati, Ramtin, et al. "Being optimistic to be conservative: Quickly learning a CVaR policy." Proceedings of the AAAI conference on artificial intelligence. Vol. 34. No. 04. 2020.

[3] Dabney, Will, et al. "Implicit quantile networks for distributional reinforcement learning." International conference on machine learning. PMLR, 2018.

[4] Nicole Bauerle and Jonathan Ott. Markov Decision Processes with Average-Value-at-Risk criteria. Mathematical Methods of Operations Research, 74(3):361–379, 2011.

[5] Nicole Bauerle and Alexander Glauner. Minimizing spectral risk measures applied to Markov decision processes. Mathematical Methods of Operations Research, 94(1):35–69, 2021.

[6] Yinlam Chow, Aviv Tamar, Shie Mannor, and Marco Pavone. Risk-sensitive and robust decision making: a CVaR optimization approach. In Advances in neural information processing systems 28, 2015.

[7] Hau, Jia Lin, et al. "On dynamic programming decompositions of static risk measures in Markov decision processes." Advances in Neural Information Processing Systems 36 (2024).

[8] Stanko, Silvestr, and Karel Macek. "Risk-averse Distributional Reinforcement Learning: A CVaR Optimization Approach." IJCCI. 2019.

[9] Xia, Li, Luyao Zhang, and Peter W. Glynn. "Risk‐sensitive Markov decision processes with long‐run CVaR criterion." Production and Operations Management 32.12 (2023): 4049-4067.

[10] Miller, Christopher W., and Insoon Yang. "Optimal control of conditional value-at-risk in continuous time." SIAM Journal on Control and Optimization 55.2 (2017): 856-884.

[11] Lim, Shiau Hong, and Ilyas Malik. "Distributional reinforcement learning for risk-sensitive policies." Advances in Neural Information Processing Systems 35 (2022): 30977-30989.

**Questions:**

Do we know the proposed algorithm updating VaR and CVaR simultaneously would converge to the optimal fixed point, not any other fixed point?

(a) The quantile regression stochastic approximation from equation (C.2) provides a quantile estimate which may not be unique for discrete random variable, VaR is only an element of quantile which is not an elicitable risk measure (see [1]). Therefore, quantile regression may not converge to VaR, perhaps VaR is not necessary and any quantile estimate is sufficient? However, CVaR is also not elicitable which makes it unclear how stochastic approximation can approximate these values accurately. There may be an assumption missing for the subtask function $f$ to handle the nuances of the problem discussing here.

(b) It is unclear why the VaR approximation in algorithm 7 is update with $\delta$ instead of the gradient of quantile (L1) loss update (C.2) (see [2]). Note that the gradient of L1 loss is a piecewise constant.

(c) In Appendix C, the claim that "We can see that when the VaR estimate is equal to the actual VaR value, the quantile regression-inspired terms in Equation C.5f become zero" holds only for continuous distributions during policy evaluation. Furthermore, this is insufficient, the authors may need to demonstrate that, starting from any initial estimate, the VaR estimate converges to the actual VaR, and similarly, that the CVaR estimate converges to the actual CVaR. Even if convergence is achieved in policy evaluation, there is no proof validating this statement for the discrete case or for policy optimization.

References:

[1] Bellini, Fabio, and Valeria Bignozzi. "On elicitable risk measures." Quantitative Finance 15.5 (2015): 725-733.

[2] Shen, Yun, et al. "Risk-sensitive reinforcement learning." Neural computation 26.7 (2014): 1298-1328.

---

> ### Author Response · Authors · 2024-11-18
>
> We thank the reviewer for their thorough and insightful review of our paper. We appreciate the amount of detail and consideration given. Please see our response below. Due to the character limit, we split our response across three comments:
>
> **(1 / 3)**
>
> **Several existing approaches such as dynamic risk-averse MDPs [1], risk-averse distributional RL [2, 3,11] and average-criteria CVaR [9] also avoid state-space augmentation and employ stationary Markov policies, similar to this work.**:
>
> We are happy to engage with the reviewer further on this topic. However, we would argue that our claim stands based on the following:
>
> *Dynamic risk-averse MDPs [1]*: It was shown by Boda (2006) (see below for our rebuttal references) that CVaR is not a time consistent risk measure, hence any time-consistent interpretation of CVaR is only an approximation (note that our claim is only specific to CVaR).
>
> *Risk-averse distributional RL [2, 3, 11]*: It was shown in [11] that the CVaR optimization approach utilized in [2, 3] (which avoided the state-space augmentation) converges to neither the optimal dynamic-CVaR nor the optimal static-CVaR policies. The authors of [11] then proposed a valid approach that utilizes an augmented state-space.
>
> *Average-criteria CVaR [9]*: We kindly point out to the reviewer that this work indeed used an augmented state-space as well as an explicit (sensitivity-based) bi-level optimization (note that this was mentioned in our paper).
>
> Note that we have included these points in Section 2.2 of the updated draft.
>
> **The proposed algorithm still seems to be bilevel as it aim to optimize for CVaR but update the VaR estimate at every level.**:
>
> Correct, our algorithm is an *implicit* bilevel optimization, where both VaR and CVaR are updated in a fully-online manner in a single MDP. By contrast, methods such as [4, 5, 9] require an *explicit* bilevel optimization, where, for example, multiple MDPs with different VaR guesses must be solved in order to find the optimal policy. Our claim, as stated in the paper, is that our method does not require an explicit bilevel optimization.
>
> **Restricting to stationary Markov policies can impair both the optimality and interpretability of static CVaR MDPs (see [4, 5, 6, 7]), since the sum over for average criteria is outside of the CVaR operator, this is closer to the dynamic category where optimal deterministic stationary policy exist (see Theorem 1 of [9])**:
>
> We agree that the CVaR being optimized in our paper has properties of both static and dynamic risk measures (while not perfectly fitting either definition). In the updated draft, we have updated the wording in the paper to be more neutral, such that it includes arguments for each (including the insightful point made by the reviewer). See lines 1608-1619 of the update draft.
>
> **Additionally, the authors overlook related works [8] applies a similar TD update and [10] consider time-consistent policies set.**:
>
> We thank the reviewer for bringing [8] and [10] to our attention as they do have enough resemblance to our work to merit a discussion in our paper. We note that while one of the methods in [8] does use a vaguely similar TD update, all of the methods proposed in [8] require either an augmented state-space or an explicit bi-level optimization. Similarly, while [10] does not use an augmented state-space, they also require an explicit bi-level optimization. In both cases, these works, while relevant, do not impact the novelty or claims made in our paper.
>
> **The CVaR analysis in Appendix C.1 is focused solely on evaluation, leaving out an analysis for policy optimization claim "We can now optimize the expectation in Equation C.5f using the RED RL framework".**:
>
> The updated draft now includes a CVaR-specific proof in Appendix D (see Theorem D.1.2; line 1570).
>
> **The average criterion CVaR objective function itself is not explicitly presented in the paper.**:
>
> The updated draft now explicitly presents the CVaR objective as Equation D.6 (line 1450)
>
> **Sections 4 and 5 feel somewhat disconnected; providing a clearer explanation to link these sections, along with an explicit proof that the proposed algorithm can optimize the CVaR objective, would significantly strengthen the paper’s claims regarding risk-aware reinforcement learning in average reward MDPs.**:
>
> We have updated the wording at the start of Section 4 and 5 to better communicate how they are related to each other. Moreover, the updated draft now includes a CVaR-specific proof in Appendix D (see Theorem D.1.2; line 1570).
>
> **...(Continued in next comment)**

---

> ### Author Response · Authors · 2024-11-18
>
> **(2 / 3)**
>
> **The claim to “learn a policy that optimized the CVaR value without using an explicit bi-level optimization scheme or an augmented state-space, thereby alleviating some of the computational challenges” is not substantiated.**:
>
> We would respectfully argue that reducing the size of the state-space, or only having to, for example, solve a single MDP instead of multiple MDPs (due to an explicit bilevel optimization) can be reasonably interpreted as alleviating computational challenges. We do not claim any specific numerical advantage.
>
> **Do we know the proposed algorithm updating VaR and CVaR simultaneously would converge to the optimal fixed point, not any other fixed point?**:
>
> The updated draft now includes a CVaR-specific proof in Appendix D (see Theorem D.1.2; line 1570) that addresses this. In short, we know that (as is the case with the standard average-reward formulation) the proposed (tabular) algorithm converges to the optimal fixed point, up to an additive constant. We note that this does not affect our ability to find the CVaR optimal policy, given that the relative ordering of policies is what is of interest.
>
> **It is unclear why the VaR approximation in algorithm 7 is update with the TD error instead of the gradient of quantile (L1) loss update (C.2)**
>
> This highlights the appeal of our approach and the core contribution of our paper. Instead of having to rely on the gradient of the quantile loss to estimate our subtask (in this case, VaR), our approach allows us to estimate VaR in a theoretically-sound way using a modified version of the TD error.  The updated draft now includes CVaR-specific proofs in Appendix D (see Theorems D.1.1 and D.1.2) that formalize this logic.
>
> We have also included a clarifying statement in Appendix D (lines 1419-1422) to make it clear that we do not use the quantile loss directly. The reason that we bring up quantile regression in our paper, is to motivate the terms that get added to the expression inside the expectation in Equation 10 (Equation 7 in the updated draft), which yields the final subtask function for CVaR (see Appendix D.1 in the updated draft). These quantile regression-inspired terms are needed because they narrow the set of possible solutions, to those with a reasonable VaR estimate.
>
> **The quantile regression stochastic approximation from equation (C.2) provides a quantile estimate which may not be unique for discrete random variable, VaR is only an element of quantile which is not an elicitable risk measure (see [1]). Therefore, quantile regression may not converge to VaR, perhaps VaR is not necessary and any quantile estimate is sufficient? However, CVaR is also not elicitable which makes it unclear how stochastic approximation can approximate these values accurately. There may be an assumption missing for the subtask function to handle the nuances of the problem discussed here.**
>
> As stated in the above response, the key appeal of our approach is that we do not need to rely on the quantile loss to estimate our subtask (in this case, VaR). In particular, our approach allows us to estimate VaR (and consequently CVaR) in a theoretically-sound way using a modified version of the TD error. The updated draft now includes CVaR-specific proofs in Appendix D (see Theorems D.1.1 and D.1.2) that formalize this logic.
>
> The reason that we bring up quantile regression in our paper, is to motivate the terms that get added to the expression inside the expectation in Equation 10 (Equation 7 in the updated draft), which yields the final subtask function for CVaR (see Appendix D.1 in the updated draft). These quantile regression-inspired terms are needed because they narrow the set of possible solutions, to those with a reasonable VaR estimate.
>
> **In Appendix C, the claim that "We can see that when the VaR estimate is equal to the actual VaR value, the quantile regression-inspired terms in Equation C.5f become zero" holds only for continuous distributions during policy evaluation. Furthermore, this is insufficient, the authors may need to demonstrate that, starting from any initial estimate, the VaR estimate converges to the actual VaR, and similarly, that the CVaR estimate converges to the actual CVaR. Even if convergence is achieved in policy evaluation, there is no proof validating this statement for the discrete case or for policy optimization.**
>
> The updated draft now includes a CVaR-specific proof in Appendix D (see Theorem D.1.2; line 1570).
>
> **...(Continued in next comment)**

---

> ### Author Response · Authors · 2024-11-18
>
> **(3 / 3)**
>
> **The results in Section 5 do not demonstrate that the proposed algorithm effectively optimizes the desired CVaR risk level. The evaluation would be more convincing if the authors trained the algorithm across multiple distinct CVaR risk levels.**:
>
> The updated draft now includes an additional experiment (see Appendix E; line 1806) that shows that our CVaR algorithm optimizes at the desired risk level, such that it reliably finds the CVaR optimal policy across various CVaR risk levels.
>
> We would also like to note that, as is the case with the standard average-reward formulation, the optimal solution that the algorithm converges to is correct up to an additive constant. Note that this was mentioned several times in our paper (e.g. lines 147-149, 465-468 in the original draft). As such, comparing the CVaR estimates of algorithms trained using different $\tau$’s may not be productive, given that comparing the values of the estimates themselves is not meaningful.
> What is meaningful is seeing whether the algorithm converges to the optimal CVaR policy at the desired risk level. This is precisely what the new experiment shows.
>
> **Comparing the proposed algorithm’s performance with other approaches [4,9,11] under an average reward criterion could also provide a clearer benchmark for its effectiveness.**:
>
> The notion of comparing results from an average-reward MDP (as in our method) to those of episodic and discounted MDPs, such as [4, 11], is an interesting one. However, there are several intricacies that would need to be addressed to make that comparison, and we fear that it would distract from the main purpose of this paper. For instance, a discounted approach would optimize the CVaR of the discounted sum of rewards, vs. our approach, which optimizes the CVaR of the reward received per time-step, thereby making it challenging to interpret a comparison of such methods. While it is possible to quantify the performance of discounted algorithms via an average reward criterion, doing such a quantification would only answer questions related to the discounted approach, such as, “how does optimizing the CVaR of the discounted sum of rewards affect the long-run reward CVaR?”. By contrast, such a quantification would not reveal any insights related to our approach.
>
> In the case of [9], their approach requires the use of an explicit bilevel optimization, which cannot be directly compared to our approach, which optimizes CVaR in a fully-online manner. The only motivation to compare our approach to [9] would be if we did not know what the optimal CVaR policy is (such that we could check whether both methods converge to the same solution). However, in our experiments, we know what the optimal policies are, so there is no need to compare our approach to [9].
>
> As such, while we recognize that our experiments are limited, we hope that the reviewer will take into account that the novelty of our work makes it difficult to directly compare our algorithm to existing works. For instance, even being able to plot (fully-online) learning curves, such as the ones in our paper, cannot be done for most of the (CVaR) algorithms referenced by the reviewer, given the explicit bilevel optimization that is required by many of these methods. The only direct comparison that can be made is to the risk-neutral differential algorithms, which we did compare our algorithms against, and found that in the case where both approaches shared a common solution (i.e., the inverted pendulum experiment), our approach had better performance than the risk-neutral algorithm.
>
> **Overall**:
>
> We hope that we have clarified and justified the following to the reviewer:
>
> - That our paper’s claims are accurate in the context of the previous works mentioned by the reviewer.
>
> - That our CVaR optimization approach is valid, as shown by the CVaR-specific proof in Appendix D of the updated draft (see Theorem D.1.2; line 1570).
>
> - That while our experiments are limited, they answer key questions related to our approach, and show that our algorithms can indeed find the optimal CVaR policy. We have conducted a follow-up experiment which shows that the RED CVaR algorithm converges to the CVaR optimal policy for a range of $\tau$'s, thereby showing that our algorithm is able to optimize at the desired risk level. These new results are presented in Appendix E of the updated draft (line 1806).
>
> We are happy to provide additional clarifications and discussion as needed by the reviewer.
>
> **References:**
> K. Boda and J. Filar. Time consistent dynamic risk measures. Mathematical Methods of Operations Research, 63(1):169–186, 2006

---

> ### Comment · Reviewer_dBpb · 2024-11-20
> **Good rebuttal with possible improvement**
>
> I have revised my rating based on the authors' response. Despite, the organization of the main paper remains difficult to follow and the clarity of their proofs could be further improved. I thank the authors for their detailed rebuttal, which strongly highlights the significance of their contributions and has largely convinced me of their merit.
>
> - I would still advise the author to explicitly include their average reward CVaR objective in the main body i.e. $\lim_{n \to \infty} \frac{1}{n} \sum_{t=1}^n CVaR_\tau[R_t | S_0=s,A_{0:t-1} \sim \pi]$.
>
> - Since this paper focus on long term average reward, I also agree with Reviewer NkZR that the Blackwell optimality criterion should be analyze or discussed to strengthen the paper.

---

> ### Author Response · Authors · 2024-11-20
>
> We thank the reviewer for their quick response, their willingness to reconsider their initial score, as well as their most recent comments.
>
> We have now included the average-reward CVaR objective in the main body of the latest draft (Equation 9).
>
> We have also added a discussion on Blackwell-optimality, as suggested by the reviewer (lines 165-171). In short, the Blackwell optimality criterion is related to discounted MDPs, and serves as a metric that indicates whether a discounted-optimal policy is also average-optimal (i.e., whether it satisifies the long-run average-reward optimality criteria). In the context of our paper, we employ methods that utilize the standard average-reward MDP, which only aims to optimize the long-run behavior (hence the solution is not blackwell-optimal). We note that it is the unique properties of the standard average-reward MDP that enable our subtask-driven approach. As such, although out methods do not yield Blackwell-optimal policies, they do yield our subtask-driven approach, along with the key CVaR result.
>
> Finally, we are more than happy to work with the reviewer to improve the ordering of the paper. In particular, we would be grateful if the reviewer could specify what aspect of the orderning is confusing, so that we can address it.
>
> We look forward to further discussions with the reviewer to address any remaining concerns and answer any more questions.

---

> ### Comment · Reviewer_dBpb · 2024-11-20
> **Reply to Authors comment**
>
> We thank the author for their quick response and made changes to their draft.
>
> - In terms of readability of the paper, I do agree with Reviewer FEZJ that **hard to understand: It is not clear why subtasks are introduced when the main goal is the CVaR problem, until Section 5. Even in Section 5, the authors didn't explain explicitly how these two ideas are related and how equation (19) was derived.** Despite the fact the author has made clarification and **the aim of the paper is to present a framework for solving subtasks simultaneously, with CVaR being an important case study that successfully utilizes this framework. We note that the fundamental approach presented (Theorems 4.1-4.3) is not CVaR-specific.** This paper aims to address general subtask-driven RL, but it provides limited motivation and applications beyond the CVaR objective. Focusing on a clear emphasis either on general subtask-driven RL or the CVaR average criterion, could enhance the readability. If the emphasis is on general subtask-driven RL, providing motivation and demonstrating applications beyond CVaR would strengthen the narrative. Alternatively, if the focus is on the CVaR average criterion, then introducing the CVaR problem early on and framing the subtasks as a methodology for addressing the CVaR objective may make the paper easier to follow.
>
> - It seems to me that Lemma D.1-4 should be Corollary since their result have already been proven but worded for more intuitive interest.

---

> > ### Author Response · Authors · 2024-11-21
> >
> > We thank the reviewer for their continued engagement and quick responses.
> >
> > As suggested by the reviewer, we have changed Lemmas D.1-D.4 to Corollaries.
> >
> > We have also added a section to the main body (Section 4 of the latest draft), that explains the challenges of CVaR optimization, as well as how the subtask approach can be used to mitigate these challenges. We believe that this new section will better explain to the reader how our subtask approach fits into the goal of optimizing CVaR, and thereby make the paper easier to follow.
> >
> > We look forward to further discussions with the reviewer to address any remaining concerns and answer any more questions.

---

### Official Review · Reviewer_zhPt · 2024-11-09

**Soundness:** 3
**Presentation:** 3
**Contribution:** 3
**Rating:** 6
**Confidence:** 3

**Summary:**

This paper studies a class of average-reward reinforcement learning problems, which includes risk-sensitive RL as special case.  The main contribution is a new framework called Reward-Extended Differential (RED) RL, which leverages structural properties of average-reward RL. RED RL can be used to devise RL algorithms for a rather broad range of objectives admitting some notion called “subtask”. In particular, the paper showcases the efficacy of this framework in the design of risk-averse RL algorithms under the CVaR risk measure, and this appears to be main motivation behind RED RL. The key benefit of the new framework here would be to avoid bi-level optimization or state-space augmentation that appear in the existing RL algorithms under CVaR criterion.

**Strengths:**

The paper studies an interesting problem in average-reward RL, which leverages a structural property that is specific to average-reward MDPs. The introduced framework appears interesting in its generic form, although its presentation in the paper is done in a rather high and abstract level. I found its application to CVaR RL quite interesting. In addition, that it removes the need to solve bi-level optimization problems explicitly is definitely a plus.

The paper is mostly well-organized and well-written. I have a couple of minor comments about writing and organization that I defer until the next section. Whenever applicable, the paper uses some figures to illustrate some concept, which proved quite helpful.

The paper includes numerical experiments, which is a positive aspect. The two domains used for the experiments sound interesting and relevant to showcase the framework.

**Weaknesses:**

Main Comments:
-
- One main comment is regarding the assumption. In view of statements in line 123-124, it appears to me that effectively a unichain assumption is made both for prediction and control.

- As a weak aspect, the presented framework only is shown to enjoy asymptotic convergence (in the tabular case).

- Regarding CVaR RL, use of an augmented state-space is mentioned as a standard technique. Of course, it is clear that we lack interest in extending state-space – especially if there is some workaround – for the classical performance bounds that deteriorate as the size of state-space grows. However, it is worth remarking that an “augmented but highly structured” state-space is not necessarily a weak aspect if one could leverage the underlying structure. Could you explain whether this is the case for CVaR RL?

- In Section 5, Equation 19: could you clarify what the choice of function $f$ is.

- As a general comment, I wonder whether RED performs simultaneous learning of multiple subtasks without any sacrifice? If not, it is not highlighted enough in the paper (or maybe I miss something).

- Subtask may bring some confusion because of its use as a standard terminology in hierarchical RL terminology. Also, I do not think that this choice of naming effectively reflects what it actually serves. Other candidates?

- I found the literature review part rather week. Admittedly, there is a rarity of prior work dealing with learning multiple goals/objectives in average-reward MDPs. However, in other MDPs settings and bandits – that are obviously more straightforward to analyze – there might exist a relatively richer literature. Further, one key contribution of the paper falls into the realm of risk-sensitive RL. It is therefore expected to see a better coverage of the related literature (and for discounted and episodic settings).

- The preliminary on average-reward MDPs and RL is rather long. Despite less work on them comparatively, they are standard settings and notions for a venue such as ICLR. I suggest Section 3.1 to be compressed to that the space in the main text could be used for more novel aspects.

Minor Comments:
-
- Figures are not readable.

Typos:
-
- Line 84: builds off of Wan et al. ==> Did you mean “builds on Wan et al.”?
- Line 61: in the Appendix ==> I think it is more correct to use “in Appendix” or “in the appendix”.
- Line 105: $S$ is a finite set of states, $A$ is … ==> $\mathcal S$ is …, $\mathcal A$ is …

**Questions:**

See above.

---

> ### Author Response · Authors · 2024-11-18
>
> We thank the reviewer for their consideration and review of our paper. The updated draft has corrected the typos identified by the reviewer. We are also happy to engage with the reviewer to see what about the figures makes them hard to read. Please see our response to the main comments below. Due to the character limit, we split our response across two comments:
>
> **(1 / 2)**
>
> **In view of statements in line 123-124, it appears to me that effectively a unichain assumption is made both for prediction and control.**:
>
> We thank the reviewer for pointing this out. We have updated the wording to highlight that the communicating assumption only guarantees the existence of a unique optimal average-reward. For clarity, our methods only require a communicating assumption for control.
>
> **As a weak aspect, the presented framework only is shown to enjoy asymptotic convergence.**
>
> We note that this is also the case for the standard average-reward formulation. We also note that while we do not offer convergence proofs for the non-tabular case, we utilized function approximation in the inverted pendulum experiment, and our algorithm still converged (in fact, it showed faster convergence compared to the risk-neutral differential algorithm).
>
> **Regarding CVaR RL, use of an augmented state-space is mentioned as a standard technique. Of course, it is clear that we lack interest in extending state-space – especially if there is some workaround – for the classical performance bounds that deteriorate as the size of state-space grows. However, it is worth remarking that an “augmented but highly structured” state-space is not necessarily a weak aspect if one could leverage the underlying structure. Could you explain whether this is the case for CVaR RL?**:
>
> We are happy to provide an explanation. In short: this is not the case for CVaR RL.
>
> The primary non-triviality lies in that we need to know what the optimal VaR is in order to calculate the optimal CVaR. However, one does not typically know this value beforehand, so existing methods have to perform some version of the optimization presented in Equation 11 of our paper (Equation 8 in the updated draft).
>
> In a standard/naive implementation of Equation 11 (Equation 8 in the updated draft), we need to augment the state-space with VaR, which can be any real (potentially-bounded) number. Moreover, a naive implementation often implies solving multiple MDPs (each with a different guess for VaR), which compounds the computational costs induced by a larger state-space.
>
> Now consider more clever methods that attempt to mitigate the computational costs. One of the most well-known, computationally-efficient examples is [1] (see below for our rebuttal references), who utilized a clever but cumbersome decomposition technique that made it possible to only need to augment the state-space with a value between 0 and 1, as well as only needing to solve a single MDP. However, even this clever method requires the use of linear interpolation, as well as having to solve a standalone optimization at every iteration.
>
> By contrast, the average-reward formulation, in combination with our proposed approach, allows us to circumvent these issues altogether, such that we can optimize both VaR and CVaR simultaneously in a fully-online manner. As such, we believe that our method is more appealing than even the more clever methods used in the discounted case.
>
> **In Section 5, Equation 19: could you clarify what the choice of the subtask function is:**
>
> The subtask function used is presented explicitly as Equation D.6 (line 1450) in the updated draft. In short, the function is a modified version of Equation 10 (Equation 7 in the updated draft). The reason Equation 10  (Equation 7 in the updated draft) needs to be modified is that directly using it as the subtask function may result in multiple solutions (see lines 1383-1450), so we need a way to reduce the set of possible solutions to only solutions that have realistic VaR estimates. To accomplish this, we employ techniques from quantile regression. Note that we explain this in great detail in Appendix D.1.
>
> **...(Continued in next comment)**

---

> > ### Author Response · Authors · 2024-11-18
> >
> > **(2 / 2)**
> >
> > **As a general comment, I wonder whether RED performs simultaneous learning of multiple subtasks without any sacrifice? If not, it is not highlighted enough in the paper (or maybe I miss something).**:
> >
> > From our perspective, there are only two sacrifices that need to be made in order to perform the simultaneous learning of multiple subtasks. The first is adhering to the assumptions about the induced Markov chain (unichain or communicating). The second is crafting the subtask function, which may not be straightforward. However, given a valid subtask function, and the willingness to adhere to the Markov chain assumptions, our algorithms are able to learn an arbitrary number of subtasks simultaneously without additional sacrifice (other than having to perform an extra learning update per subtask). In the case of CVaR, this has the added benefit of removing the need to augment the state-space and having to solve multiple MDPs. Empirically, we saw that our algorithm could reliably learn the subtask, and even outperformed the risk-neutral differential algorithm (which does not have to estimate a subtask) in the inverted pendulum experiment (where both methods shared a common optimal solution).
> >
> > **Subtask may bring some confusion because of its use as a standard terminology in hierarchical RL terminology. Also, I do not think that this choice of naming effectively reflects what it actually serves. Other candidates?**
> >
> > We agree that there may be some confusion, however the term ‘subtask’ or ‘auxiliary task’ is used in various contexts in RL (e.g. [2]). In this regard, we have updated Section 2 to contrast our definition of subtask, to that of other subfields, such as hierarchical RL.
> >
> > **I found the literature review part rather weak. Admittedly, there is a rarity of prior work dealing with learning multiple goals/objectives in average-reward MDPs. However, in other MDPs settings and bandits – that are obviously more straightforward to analyze – there might exist a relatively richer literature. Further, one key contribution of the paper falls into the realm of risk-sensitive RL. It is therefore expected to see a better coverage of the related literature (and for discounted and episodic settings).**:
> >
> > Agreed, we have added some non-average-reward MDP references in the updated draft related to learning multiple goals/objectives in the literature review section (such as [2]). In terms of risk-sensitive RL, we have added a few more additional key (CVaR-related) references for the discounted and episodic settings.
> >
> > **The preliminary on average-reward MDPs and RL is rather long. Despite less work on them comparatively, they are standard settings and notions for a venue such as ICLR. I suggest Section 3.1 to be compressed to that the space in the main text could be used for more novel aspects.**:
> >
> > We thank the reviewer for their recommendation and we have shortened Section 3.1 to expand the literature review.
> >
> > **Overall**:
> >
> > We hope that we have addressed the reviewer’s comments. We are happy to engage with the reviewer to provide additional clarifications.
> >
> > **References**:
> >
> > [1] Chow, Yinlam, Aviv Tamar, Shie Mannor, and Marco Pavone. 2015. “Risk-Sensitive and Robust Decision-Making: A CVaR Optimization Approach.” In Advances in Neural Information Processing Systems 28.
> >
> > [2] McLeod, Matthew, et al. "Continual auxiliary task learning."  In Advances in Neural Information Processing Systems 34.

---

> > > ### Author Response · Authors · 2024-12-02
> > >
> > > Dear Reviewer zhPt,
> > >
> > > We hope that our updated draft and response to your comments have addressed your concerns and provided the necessary clarifications.
> > >
> > > We are happy to further engage with you to address any remaining concerns and/or answer any more questions.
> > >
> > > Best regards,
> > >
> > > Authors

---

> > > > ### Comment · Reviewer_zhPt · 2024-12-03
> > > >
> > > > Thanks for detailed responses and revising the paper.
> > > >
> > > > - Regarding figures, the legends are small. Consider using larger font sizes.
> > > >
> > > > - Regarding the comment about asymptotic convergence: I am not sure if I was clear enough. I am not sure either if I fully understood your response. Even though the average-reward notion is an asymptotic one, non-asymptotic bounds could still be relevant, and this is done in the context of many learning algorithms derived for this setting (e.g., non-asymptotic regret bounds for regret minimization in average-reward RL, and many others). In essence, the notion of gain $g^\pi$ of a policy $\pi$ is asymptotic, but it can be related to the running average $\sum_{t=1}^T r_t/T$ for $r_t \sim R(s_t,a_t)$ with $a_t\sim \pi(s_t)$ for any finite $T$ and corresponding deviation bounds could be derived. It is fine if the current version only derives asymptotic bounds. Yet, I think deriving non-asymptotic ones is of relevance as future direction.

---

> > > > > ### Author Response · Authors · 2024-12-03
> > > > >
> > > > > We thank the reviewer for their most recent comments.
> > > > >
> > > > > We will increase the font size in our plots to make them more readable (including the legends).
> > > > >
> > > > > We also thank the reviewer for the clarification on the non-asymptotic bounds. We agree that deriving non-asymptotic bounds is a fruitful direction to pursue in future work.
> > > > >
> > > > > We again thank the reviewer for their time and consideration.

---

### Author Response · Authors · 2024-11-18
**Rebuttal Revision**

We would like to thank the reviewers for their detailed and considerate reviews of our paper. We have uploaded a modified version of our paper that incorporates the reviewer’s comments. Namely, in the updated draft:

- We fixed the typos/errors identified by the reviewers,
- We expanded the literature review section (Section 2) and incorporated the relevant works identified by the reviewers,
- We added a section (Section 4 in the updated draft) that explains the challenges of CVaR optimization, as well as how the subtask approach can be used to mitigate these challenges,
- We simplified the proof for Theorem 4.1 (now Theorem 5.1) to make it less vague,
- We included CVaR-specific proofs in Appendix D (see Theorems D.1.1 and D.1.2) that show that our CVaR optimization approach is valid, and
- We ran an additional experiment and included results in Appendix E (line 1806). This new experiment shows that our approach is able to successfully optimize CVaR at the desired risk level.

All in all, we believe that these changes address the reviewers' concerns. We look forward to further engaging with the reviewers to address any remaining concerns, and answer any remaining questions.

---

### Meta-Review · Area_Chair_ktcw · 2024-12-08

**Metareview:**

This paper introduces a Reward Extended Differential (RED) approach for risk-averse AMDP that aims to handle multiple subtasks concurrently, by defining the TD error through a modified reward generated by the observed rewards and subtasks through an invertible function. They leverage a property of CVaR from Rockafellar and Uryasev and treat CVaR as the expectation of a piecewise linear function of a random variable and its Value-at-Risk and hence apply the RED method to optimize the risk-averse CVaR objective function.

However, apart from the relatively minor concerns in the related work discussion, numerical experiments, the paper has a major technical issue in that the derivation 13a-13b and 14a-14b, which does not hold for nonlinear functions. Although after the revision, the authors modified this proof and make it specific to linear $f$. However, linear $f$ does not contain their main target (CVaR). They authors claim in a hand-waving style that the proof can be trivially extended to piece-wise linear function by applying it on each piece. However, this claim is wrong. In fact, piece-wise linear functions provide uniform approximation of continuous functions over compact set, the authors' claim would suggest their result to hold for arbitrary continuous functions, which is impossible.

Based on this technical issue, we decide to reject this paper.

**Additional Comments On Reviewer Discussion:**

The reviewers have major concerns in related works, writing and clarity of presentation, whether the considered risk is nested or not, and technical validity of the derivations.

The authors have cleared several of the above concerns by making many changes in the writing and derivations. However, their response to technical validity does not convince the AC and we believe the proof is not correct.

---

### Decision · Program_Chairs · 2025-01-22

Reject